**ARTICLES**

# Antibody stabilization for thermally accelerated deep immunostaining

Hei Ming Lai [1,2,3,4,5,16] ✉, Yumi Tang [2,3,16], Zachary Y. H. Lau [2,3,16], Robert A. A. Campbell[6],
Juno C. N. Yau[1,2,3], Caleb C. Y. Chan[1,3], Danny C. W. Chan [2,3,4,5,7,8], Tin Yan Wong [9],
Harriet K. T. Wong[1,3], Leo Y. C. Yan[2,3,10], William K. K. Wu [7,8], Sunny H. Wong[2,11,12], Ka-Wai Kwok [13],
Yun-Kwok Wing [1,4,5], Henry H. N. Lam [9], Ho-Keung Ng[14], Thomas D. Mrsic-Flogel[6],
Vincent C. T. Mok [2,4,5], Jason Y. K. Chan [10] and Ho Ko [1,2,3,4,5,8,15] ✉

Antibodies have diverse applications due to their high reaction specificities but are sensitive to denaturation when a higher working temperature is required. We have developed a simple, highly scalable and generalizable chemical approach for stabilizing off-the-shelf antibodies against thermal and chemical denaturation. We demonstrate that the stabilized antibodies (termed SPEARs) can withstand up to 4 weeks of continuous heating at 55 °C and harsh denaturants, and apply our method to 33 tested antibodies. SPEARs enable flexible applications of thermocycling and denaturants to dynamically modulate their binding kinetics, reaction equilibrium, macromolecular diffusivity and aggregation propensity. In particular, we show that SPEARs permit the use of a thermally facilitated three-dimensional immunolabeling strategy (termed ThICK staining), achieving whole mouse brain immunolabeling within 72 h, as well as nearly fourfold deeper penetration with threefold less antibodies in human brain tissue. With faster deep-tissue immunolabeling and broad compatibility with tissue processing and clearing methods without the need for any specialized equipment, we anticipate the wide applicability of ThICK staining with SPEARs for deep immunostaining.

ntibodies bind diverse molecules with high affinity and specificity, and play critical roles in many biomedical, chemical and industrial applications. However, as protein molecules, they are prone to irreversible denaturation by thermal energy and chemical denaturants, which limit the scope of their applications. Given that temperature is a critical determinant in many physical, chemical and biological processes, such as diffusion, reaction equilibrium and kinetics, engineering antibodies to withstand high temperatures is of considerable interest. Although recombinant technologies and other protein engineering approaches have been successful in improving antibody thermostability[1–6], they are difficult to generalize to the vast majority of commercially available antibodies due to the work involved and the unpredictable results.

Immunostaining takes advantage of the affinity and specificity of antibodies to precisely localize their molecular targets in tissues. This has become even more important in recent years with advancements in tissue clearing, which involves chemical techniques that render intact tissues transparent and enables them to be imaged using three-dimensional (3D) optical microscopy[7]. Tissue clearing facilitates high-throughput spatial mapping of tissue proteomes,

which is important to provide a holistic view of tissue biology and pathology. However, the depth of conventional immunostaining is typically limited to tens of micrometers, despite the penetration of light in cleared tissues being in the order of millimeters to centimeters. Such discrepancies restrict the use of deep imaging to tissues expressing endogenous fluorescent proteins and limit the use of immunostaining with tissue clearing techniques in both animal research and clinicopathological investigations. For example, in systems-level interrogations of neural circuits, high-quality brain-wide neurotransmitter-specific neuronal fiber tract tracing and cellular quantification typically rely on genetic labeling in transgenic animals or viral vector transfection[8–11]. To address these limitations, we developed SPEARs, chemically engineered antibodies that enable the use of high temperatures in immunostaining to facilitate their deep penetration.

## Results

**Conceptualization of a deep immunostaining strategy based on stabilized antibodies.** The main barriers to antibody penetration in tissues can be understood in terms of the reaction–diffusion

[1]Department of Psychiatry, The Chinese University of Hong Kong, Shatin, Hong Kong. [2]Department of Medicine and Therapeutics, The Chinese University of Hong Kong, Shatin, Hong Kong. [3]Li Ka Shing Institute of Health Sciences, The Chinese University of Hong Kong, Shatin, Hong Kong. [4]Margaret K. L. Cheung Research Centre for Management of Parkinsonism, The Chinese University of Hong Kong, Shatin, Hong Kong. [5]Gerald Choa Neuroscience Centre, The Chinese University of Hong Kong, Shatin, Hong Kong. [6]Sainsbury Wellcome Centre for Neural Circuits and Behaviour, University College London, London, UK. [7]Department of Anaesthesia and Intensive Care, The Chinese University of Hong Kong, Shatin, Hong Kong. [8]Peter Hung Pain Research Institute, The Chinese University of Hong Kong, Shatin, Hong Kong. [9]Department of Chemical and Biological Engineering, The Hong Kong University of Science and Technology, Clear Water Bay, Hong Kong. [10]Department of Otorhinolaryngology, Head and Neck Surgery, The Chinese University of Hong Kong, Shatin, Hong Kong. [11]Institute of Digestive Disease, The Chinese University of Hong Kong, Shatin, Hong Kong. [12]Lee Kong Chian School of Medicine, Nanyang Technological University, Nanyang Avenue, Singapore. [13]Department of Mechanical Engineering, The University of Hong Kong, Pok Fu Lam, Hong Kong. [14]Department of Anatomical and Cellular Pathology, The Chinese University of Hong Kong, Shatin, Hong Kong. [15]School of Biomedical Sciences, Faculty of Medicine, The Chinese University of Hong Kong, Shatin, Hong Kong. [16]These authors contributed equally: Hei Ming Lai, Yumi Tang, Zachary Y. H. Lau. ✉e-mail: hmlai@cuhk.edu.hk; ho.ko@cuhk.edu.hk

1137

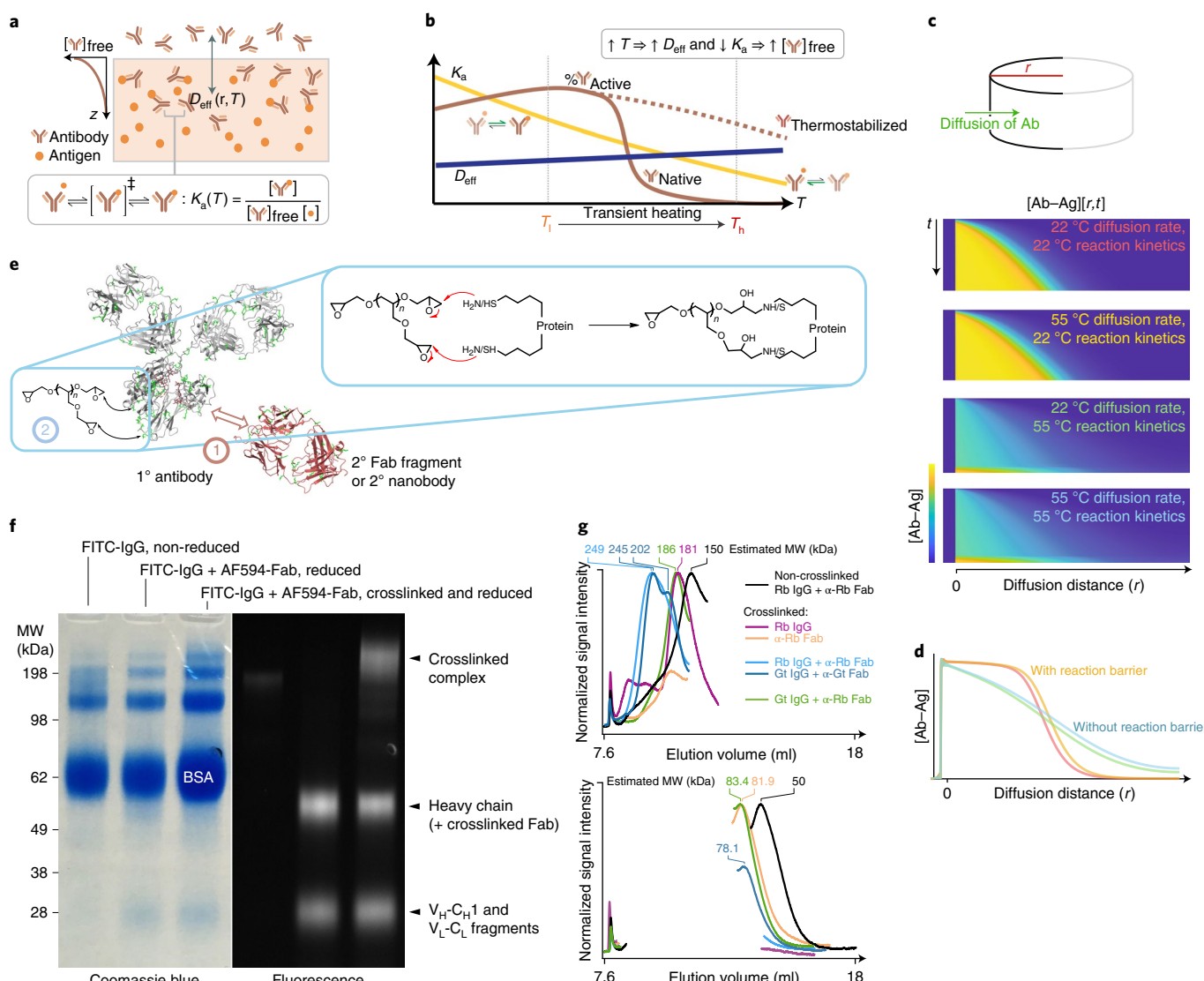

**Fig. 1 | Conception and development of heat-stable antibodies via chemical engineering for high-temperature deep immunostaining. a**, Schematic diagram of antibody (Ab) diffusion to reach the deep tissue antigen (Ag) target. $K_a(T)$ is the association constant of the Ab–Ag binding reactions at a given temperature $T$, and $D_{eff}$ is the effective diffusion coefficient of free Ab as a function of the antibody spatial location ($r$) and $T$. ‡ denotes transition state. **b**, The general relationship between $D_{eff}$, $K_a$, the percentage of active Abs and $T$. In a hypothetical heat-facilitated strategy, the Ab–Ag binding reaction is not favored at higher $T$ (that is, it lowers $K_a$), but is also irreversibly denatured at sufficiently high $T$ (brown solid line). Therefore, raising $T$ to increase the free Ab proportion is viable only if the Abs can be protected from denaturation (brown dotted line). **c**, Reaction–diffusion simulation of Ab–Ag binding and Ab diffusion in a cylindrical arena. The time ($t$)-dependent concentration profiles of the Ab–Ag complex ([Ab–Ag]) along the diffusion distance ($r$) with different combinations of $T$-dependent Ab–Ag binding kinetics and Ab diffusivity are visualized (lower panels). **d**, [Ab–Ag] versus $r$ at the end of the simulations in **c**. **e**, Strategies for stabilizing Abs against permanent heat denaturation: stage 1, complexation with a secondary Fab fragment to stabilize protein conformation; stage 2, multifunctional crosslinkers are used to crosslink the complex (insets). **f**, SDS–PAGE analysis on crosslinking primary Ab–Fab fragment complexes. AF594, Alexa Fluor 594; $C_H$, heavy chain constant domain; $C_L$, light chain constant domain; FITC, fluorescein isothiocyanate; MW, molecular weight; $V_H$, heavy chain variable domain; $V_L$, light chain variable domain. **g**, Gel filtration analysis of the optimized IgG–Fab complex crosslinking reaction mixture, including a pair of IgG and Fab with mismatched Fab host specificity. Gt, goat; Rb, rabbit. Albumin peaks have been removed for clarity. The full traces are shown in Extended Data Fig. 2.

process, in which free antibodies are depleted by antibody–antigen binding reactions and have limited penetration into deep tissue regions (Fig. 1a). To be specific, as the antibodies diffuse through the tissue, there would be intermolecular reactions between an antibody (Ab) and antigen (Ag):

$$\text{Ab (free)} + \text{Ag (immobilized)} \quad (1)$$
$$\rightleftharpoons [\text{Ab} - \text{Ag}] \text{ (bound and immobilized)}$$

This reaction is exothermic and the equilibrium is biased towards the right side of Eq. 1 at ambient or physiological temperatures, converting the free antibody into an immobile form bound to the fixed antigen. To increase the proportion of mobile antibodies, the temperature can be increased during immunostaining to temporarily shift the equilibrium to the left side of Eq. 1: a strategy we named ThICK (for thermo-immunohistochemistry with optimized kinetics) staining (Fig. 1b–d). However, at temperatures higher than ambient or physiological levels, the

denaturation of antibodies (a multistep, intramolecular reaction) becomes significant:

$$Ab_f \rightleftharpoons Ab_i \rightarrow Ab_d \qquad (2)$$

where $Ab_f$, $Ab_i$ and $Ab_d$ denote the functional, functionally deficient intermediate and permanently denatured forms of antibodies, respectively, and the transitions between the different forms are limited by kinetic barriers. To avoid a large proportion of antibodies transitioning into the $Ab_d$ form, the equilibrium between $Ab_f$ and $Ab_i$ should lie towards the $Ab_f$ side by increasing the leftward reaction rate (that is, by thermodynamic stabilization of $Ab_f$) or decreasing the rightward reaction rate (that is, by increasing the kinetic barrier to the $Ab_f \rightarrow Ab_i$ transition). Such thermodynamic stabilization of the antibody can in theory permit a strategy in which an initial high staining temperature (towards the right side of the x axis in Fig. 1b at $T_h$) reduces the antibody–antigen reaction and frees up antibodies for deep penetration. After homogeneously distributing the antibody throughout the tissue, subsequent lowering of the temperature enables the thermostable antibody to bind antigens deep within the tissue (back to the left side of the x axis in Fig. 1b at $T_1$). Consistent with our hypothesis, simulations (Fig. 1c) suggest that immobilization of antibody due to reaction with fixed tissue antigens is the main obstacle to deep penetration by antibodies. By staining at an initially high temperature (as shown by the 55 °C reaction kinetics in Fig. 1c), the reduced antibody–antigen reaction (as evident by the lower values of [Ab-Ag] at any time t) leads to a deeper and more homogeneous staining across the diffusion distance (Fig. 1c,d). The deeper penetration by antibodies is largely not accounted for by increases in diffusion rates due to higher temperatures (for example, compared with 22 °C reaction kinetics, Fig. 1c).

To achieve thermostabilization of antibodies, given that protein engineering methods are complex and inaccessible for commercially available antibodies, we chose to use chemical methods to achieve these goals. Inspired by the approach of using crystallization chaperones (for example, the antigen-binding fragment (Fab fragment) of antibodies or nanobodies) to stabilize protein conformations for crystallography studies[12] (Fig. 1e), as well as a recently reported multifunctional crosslinker, polyglycerol 3-polyglycidyl ether (P3PE) for fluorescent protein protection[13] (Fig. 1e), we developed a simple and scalable chemical strategy for the production of stabilized antibodies. Combining these two strategies enables the covalent crosslinking of an antibody–Fab fragment complex to achieve synergistic thermostabilization of the antibody (Fig. 1e). We termed the crosslinked Ab–Fab complex produced by our method 'synergistically protected polyepoxide-crosslinked Fab-complexed antibody reagents' (SPEARs). We show that by combining the use of stabilized antibodies with high-temperature immunostaining, we can bias the intramolecular and intermolecular reaction equilibria by thermocycling to hinder or facilitate antibody–antigen binding throughout the tissue. An initial high incubation temperature can thus increase the proportion of mobile, functional antibodies (that

is, $Ab_f$) for deep penetration, followed by a lowered temperature to facilitate antibody–antigen binding after they have been homogeneously distributed throughout the tissue (Fig. 1b–d).

**Chemical stabilization of antibodies against denaturation.** The first aim was to develop the ideal SPEARs for our ThICK-staining strategy. We began our investigation by complexing non-specific immunoglobulin G (IgG), as a model for primary antibodies, with anti-IgG Fab fragments (as the choice for the stabilizing chaperone) and crosslinking the so-formed complex. Using fluorescently labeled Fab fragments of secondary antibodies (hereafter referred to as Fabs), we first identified and optimized a reaction condition that leads to the reliable formation of a crosslinked IgG–Fab complex within a reasonable time (<24 h) and reaction scale (10 μl reaction per 0.1–1 μg antibody) (Fig. 1f and Extended Data Fig. 1a–c). We tested and confirmed that the P3PE-crosslinking reaction is compatible with most additives, buffer components and preservatives in commercially available antibody liquors, except Tris base due to its primary amine group (Extended Data Fig. 1d–f). The choice of conjugated fluorophore on Fabs does not affect the efficiency of the P3PE-crosslinking reaction (Extended Data Fig. 1g). Gel filtration chromatography analysis showed that most complexes are bound by 1–2 Fab fragments, and multi-complex crosslinking rarely occurs under our optimized reaction conditions (Fig. 1g and Extended Data Fig. 2).

Although P3PE crosslinking has been demonstrated to retain protein fluorescence upon heating[13], preserving the antigen-binding capability of antibodies after any kind of chemical modification is not guaranteed. We thus next designed and utilized an enzyme-linked immunosorbent assay (ELISA) variant that can functionally assess and optimize both the antigen-binding capability and the heat stability of the SPEARs in a high-throughput manner (Fig. 2a and Extended Data Fig. 3). In particular, non-specific rabbit IgGs served as a model for antigen, which would be bound by goat anti-rabbit secondary SPEARs using biotinylated donkey anti-goat Fab fragments and immobilized on avidin-coated plates. We replaced fluorescent dyes with biotinylation to mimic the protected fluorescent Ab–Fab complex in immunostaining. The amount of bound rabbit IgGs was quantified with colorimetry using horseradish peroxidase-conjugated anti-rabbit antibodies. We found that higher Ab–Fab complex:P3PE molar ratios (Fig. 2b), lower temperatures during crosslinking (Fig. 2c), optimal duration (16–24 h) of crosslinking and the presence of 0.3% w/v Triton X-100 during heat treatment (Fig. 2d) resulted in better antigen-binding capability and thermostability of the SPEARs.

After optimization, the antigen-binding capability of the SPEARs improved from 43% to 98% compared with the uncrosslinked control (Fig. 2e), and 15.9% of crosslinked SPEARs still remained functional after heating at 55 °C for 16 h (Fig. 2f). In a proof-of-concept immunostaining test using rat anti-glial fibrillary acidic protein (GFAP) antibody and Alexa Fluor 594-labeled donkey anti-rat Fab fragments, we found that P3PE crosslinking

**Fig. 2 | Functional evaluation of SPEARs under thermal and chemical denaturing conditions. a**, Schematic diagram of the custom-designed ELISA variant for the functional optimization of SPEAR antigen-binding capability and heat resistance. ABTS, 2,2′-azino-bis[3-ethylbenzothiazoline-6-sulfonic acid; Dn, donkey. **b**, Optimization of Ab–Fab complex:crosslinker molar ratio in heat protection using the ELISA variant (data given as mean ± s.d., $n = 4$ experimental replicates per group, $P = 0.02$, two-sided Mann–Whitney U-test). **c**, Optimization of crosslinking reaction temperature (data given as mean ± s.d., $n = 4$ experimental replicates per group, $P = 0.001$, one-way ANOVA). **d**, Optimization of SPEARs heating buffer composition, with or without the addition of 0.3% Triton X-100 to PBS (data given as mean ± s.d., $n = 4$ experimental replicates per group, $P = 0.03$, two-sided Mann–Whitney U-test). **e**, Antigen-binding capability of the SPEARs before (left panel, 43.2 ± 7.5%) and after (right panel, 98.0 ± 12.9%) functional optimization (data given as mean ± s.d., two-sided Mann–Whitney U-test). **f**, After optimization, 15.9 ± 0.8% of SPEARs retained their antigen-binding capability after heating for 16 h at 55 °C (data given as mean ± s.d., $n = 4$ experimental replicates per group). **g**, Immunostaining using primary anti-GFAP Ab–Fab fragment complex without crosslinking (left column), separate crosslinking followed by complex formation (middle panels), and crosslinking after complex formation (right panels). The lower panels show the results of heating the above stained tissues at 55 °C in 4% w/v SDS in 1× PBS, pH 7.4, for 1 h, for comparison with the upper panels without heating. Scale bars, 100 μm.

and Fab complex formation can synergize to stabilize antibodies against heat and denaturant destruction, while preserving staining quality (Fig. 2g), given that crosslinking after complex formation (Fig. 2g, right panels) is more effective in the protection from denaturation than separately crosslinking Ab and Fab fragments (Fig. 2g, middle panels).

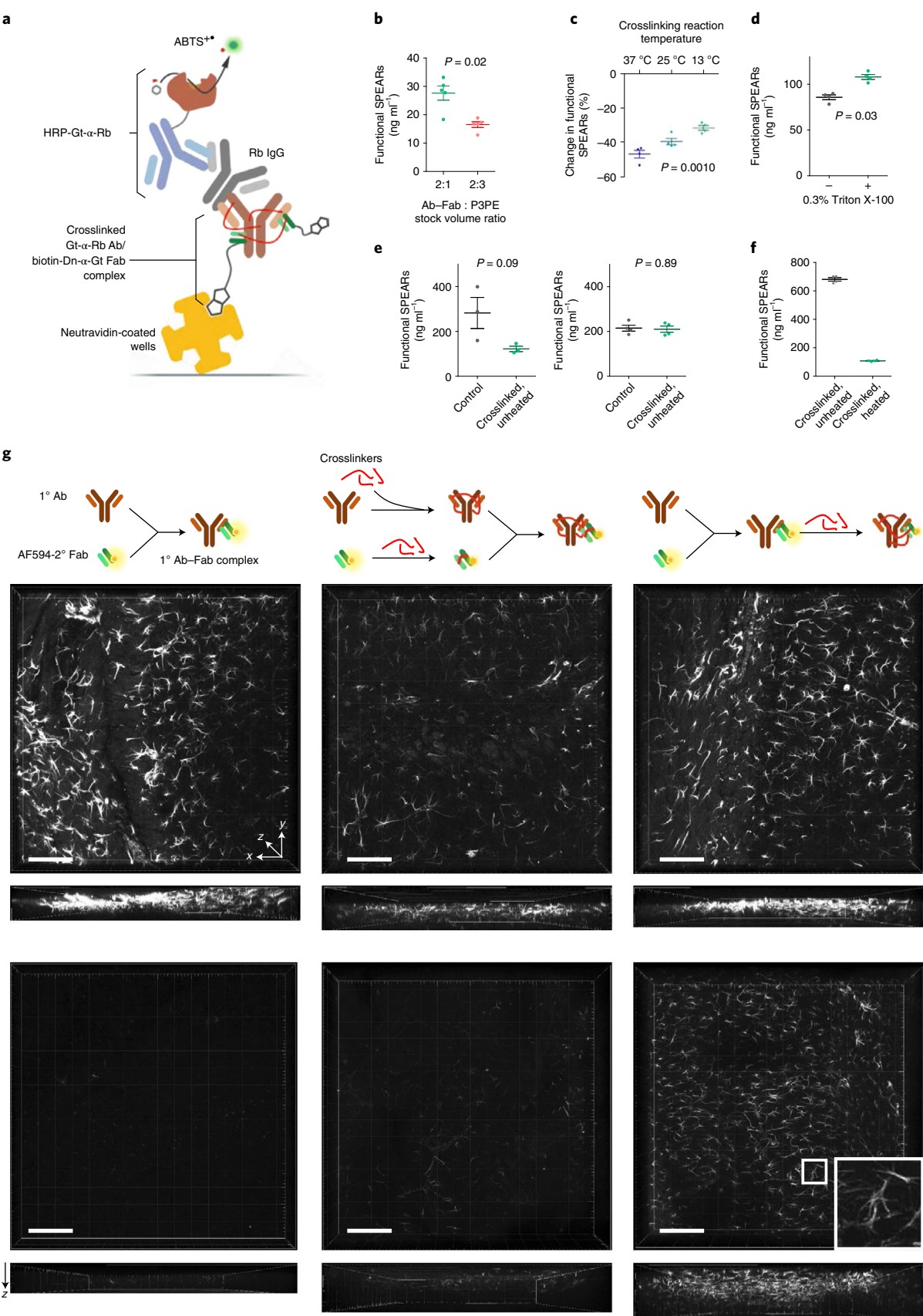

**Thermo-immunohistochemistry.** After obtaining optimally heat-resistant SPEARs, we then moved to the development of ThICK staining. In formalin-fixed mouse spinal cord tissues we found that choline acetyltransferase (ChAT) SPEAR staining can tolerate heating at 55 °C in phosphate-buffered saline with 0.3% Triton X-100 (PBST) for at least 72 h (Fig. 3a), with the signal homogeneity across depth positively correlating with the duration of heating (Fig. 3b). ThICK staining is compatible with endogenous fluorescence preserved by both conventional formalin fixation (Fig. 3c, left, endogenous GCaMP6f with tyrosine hydroxylase SPEAR staining) and SHIELD (stabilization under harsh conditions via intramolecular epoxide linkages to prevent degradation) protection[13] (Fig. 3c, right, endogenous GCaMP6f with vasoactive intestinal peptide SPEAR staining). To expand the scope of applicability, we tested and verified that SPEARs can be readily produced from 33 commercially available primary antibodies (including various neuronal subtype, activity-dependent, synaptic and glial markers, and kidney cell type markers, Supplementary Table 1) for ThICK staining at 55 °C in PBST for at least 16 h (Fig. 3d). Apart from Fab, we also tested and showed that SPEARs can be made from P3PE crosslinking of antibodies complexed with nanobodies (termed NanoSPEARs) (Fig. 3e). However, for some tested antibodies (for example, anti-IBA1, anti-Na⁺-K⁺-ATPase, anti-NeuN and anti-S100b) we failed to obtain functional SPEARs after at least three attempts (Supplementary Table 1), despite performing buffer exchange for the anti-IBA1 antibody (stored in Tris-containing buffer).

For some SPEARs (and antibodies in general) we observed precipitation in vessels (Fig. 3c,d). We speculate that this is because vessels are low-resistance diffusion channels, leading to a high concentration of antibodies for which locally inhomogeneous heating can lead to protein denaturation–refolding cycles that promote aggregation. Alternatively, the use of P3PE may also be a contributing factor. Although it is difficult to fully elucidate the precise mechanisms, we attempted to optimize the staining conditions and buffer composition, and found that the addition of certain denaturants, notably 1 M guanidinium chloride (GnCl), can mitigate the precipitation (Fig. 3f and Extended Data Fig. 4a). Image processing can also effectively remove these precipitates due to their distinct morphology (Extended Data Fig. 4b,c).

To further streamline the ThICK-staining protocol we explored whether a catalyst can improve the crosslinking reaction speed or the yield. Among many candidates (Extended Data Fig. 5a–c) we tested pyridine, which is a moderately strong nucleophile that can form a good pyridinium leaving group when attacked by primary amines (Fig. 3g). Consistent with our hypothesis, pyridine modestly catalyzed the reaction in a concentration-dependent manner (Fig. 3h) and increased the efficiency of precursor-to-product conversion during 4–8 h of reaction time (Fig. 3i and Extended Data Fig. 5d–i). These catalytically formed SPEARs (denoted as SPEAR^py) can be directly used in ThICK staining without additional purification steps (Extended Data Fig. 5j), and had higher heat resistance in a custom-designed hot-start polymerase chain reaction assay (Fig. 3j,k, see Methods). At 16 h of reaction time, ChAT SPEAR^py also improved the quality of ThICK staining compared with that produced by the non-catalyzed reaction (Fig. 3l) and is compatible with SHIELD-protected samples with endogenous fluorescent proteins (Fig. 3m).

**Volumetric immunolabeling using SPEARs and ThICK staining.** As discussed above (and illustrated in Fig. 1), the use of SPEARs in ThICK staining can in theory help overcome the antibody–antigen reaction barrier in deep 3D immunostaining, which is unmatched by simple optimizations in conventional immunostaining, with or without various tissue clearing methods (Extended Data Fig. 6). Experimentally, with optimized conventional immunostaining we observed significant rims of superficial staining that can occur throughout the tissue surface (Extended Data Fig. 6a–d) or locally (Extended Data Fig. 6e–i), regardless of the antibody–antigen pair used if the antigens are sufficiently densely distributed. ThICK staining effectively mitigates these problems, confirming the advantages of globally increasing the incubation temperature to overcome the reaction barrier, a strategy enabled by the heat stability of SPEARs.

To explore the scope of our method's applicability, we applied SPEARs to large-scale 3D imaging and benchmarking experiments using human and mouse brain tissues. We obtained a 5-mm-thick postmortem human pons transverse section inclusive of the locus coeruleus region that had been formalin-fixed for 3 weeks. After 3 days of delipidation and 48 h of ThICK staining with 30 μl tyrosine hydroxylase SPEAR^py (Fig. 4a), we were able to visualize tyrosine hydroxylase-expressing noradrenergic neurons located as far as ~800 μm from the tissue surface (mean depth of segmentable cells, 359 μm, Fig. 4b–d and Extended Data Fig. 7). In comparison, 2 weeks of conventional immunostaining using 100 μl tyrosine hydroxylase antibody on a 1.5-mm-thick human pons section from the same region[14] resulted in a mean depth of segmentable cell of 95 μm (Fig. 4d and Extended Data Fig. 7). The depth distributions of the segmentable tyrosine hydroxylase-positive cells were significantly different ($P = 5.6 \times 10^{-8}$, Kolmogorov–Smirnov test) and much

---

**Fig. 3 | Development and applications of deep immunostaining using thermostabilized antibody–Fab complex. a**, Tolerance of SPEARs to the duration and condition of ThICK staining at 55 °C. Upper panels: x–z view of mouse spinal cords that have been ThICK-stained with ChAT SPEARs. Scale bars, 50 μm. Lower panels, example cells from different depths. Scale bars, 10 μm. Color bar, pixel intensity. **b**, From left to right: homogeneity of pixel intensity mean, variability (that is, s.d.), and signal to noise ratio (SNR) by depth for various durations of ThICK staining (in hours). **c**, ThICK staining of formaldehyde-fixed (left panel, 55 °C for 1 h) and SHIELD-protected (right panel, 55 °C for 16 h) tissues with endogenous fluorescence. Scale bars, 50 μm. **d**, Antibodies compatible with SPEAR synthesis and ThICK staining. Green, Alexa Fluor 488; red, Alexa Fluor 594; cyan, Alexa Fluor 647. MIP, maximum intensity projection; PV, parvalbumin; TH, tyrosine hydroxylase; VIP, vasoactive intestinal peptide. Scale bars, 20 μm. **e**, ThICK staining with NanoSPEARs. Scale bars, 20 μm. **f**, Optimization of ThICK-staining buffer composition with respect to SPEARs intravascular precipitates per unit imaged tissue volume. The experiment was repeated two more times for control (0.3% Triton X-100) and 1 M GnCl. The error bars represent s.d. $P = 0.0216$, two-sided unpaired t-test with Welch's correction. **g**, Pyridine (py)-catalyzed P3PE-crosslinking reaction. **h**, A higher concentration of py is associated with more conversion of precursor to product. **i**, A total of 61.8 mM py accelerates crosslinking when compared with non-catalyzed control by SDS–PAGE. Inset: results of 1–8 h reaction time. ***$P \leq 0.01$ at 4 and 8 h reaction (multiple two-sided t-test with multiple comparisons adjustment), $n = 3$ replicates; data are given as mean ± s.d. **j**, Functional assay based on hot-start PCR for testing py-catalyzed synthesized SPEARs (SPEAR^py) and agarose gel analysis of the PCR product in the lower panel. **k**, Functional activity of Taq SPEAR versus Taq SPEAR^py in inhibiting PCR product formation, comparing SPEARs used directly after synthesis with those pre-heated at 55 °C for 16 h. The experiment was repeated six times and data are given as mean ± s.d. n.s., not significant. ***$P \leq 0.001$ (Tukey's multiple comparison test, two-sided). **l**, Comparison of staining qualities of ChAT SPEAR and ChAT SPEAR^py. Left panels, representative images of staining with ChAT SPEARs. Scale bars, 20 μm. Right panel, signal to background ratio along the axes of representative cells (white rectangles). Data are given as mean (solid lines) ± s.d. (shaded regions). Lighter lines represent normalized intensity profiles of individual cells. **m**, Optimized ThICK staining with ChAT SPEAR^py (red) in a SHIELD-protected sample with endogenous neuronal GCaMP6f (green). Precipitates were identified and highlighted in white. Scale bars, 50 μm.

more positively skewed and heavy-tailed for conventional immunostaining with anti-tyrosine hydroxylase antibody (skewness 1.50, kurtosis 2.78) than for ThICK staining with tyrosine hydroxylase SPEAR^py (skewness −0.17, kurtosis −0.87) (Fig. 4e). These results show that ThICK staining can overcome the antibody–antigen reaction barrier that is prominent in conventional immunostaining, to achieve deeper probe penetration.

We next performed 72 h of ThICK staining with ChAT SPEARs in SHIELD-protected mouse whole brain and hemibrain samples, imaged the samples with serial two-photon tomography and compared the staining quality with that obtained by optimized 10 day ChAT antibody staining (Fig. 4f–l, Extended Data Fig. 6e–i and

Supplementary Video 1). At the mesoscopic scale, cholinergic fiber bundles originating from the nucleus basalis of Meynert and traversing through the striatum and amygdala were well visualized in the ThICK-stained whole brain (Fig. 4g,h, Extended Data Fig. 8a,b and Supplementary Video 1). The staining also penetrated throughout the sample, labeling ChAT-expressing neurons in various deep brain regions (for example, in the hippocampus and the pedunculopontine nucleus; Fig. 4i,j and Supplementary Video 1). Notably, fine axonal fibers (for example, in the horizontal diagonal band of Broca and the cortical layer I; Fig. 4k, Extended Data Fig. 8c and Supplementary Video 1) could also be delineated. With such thorough staining, neurotransmitter-specific tractography can

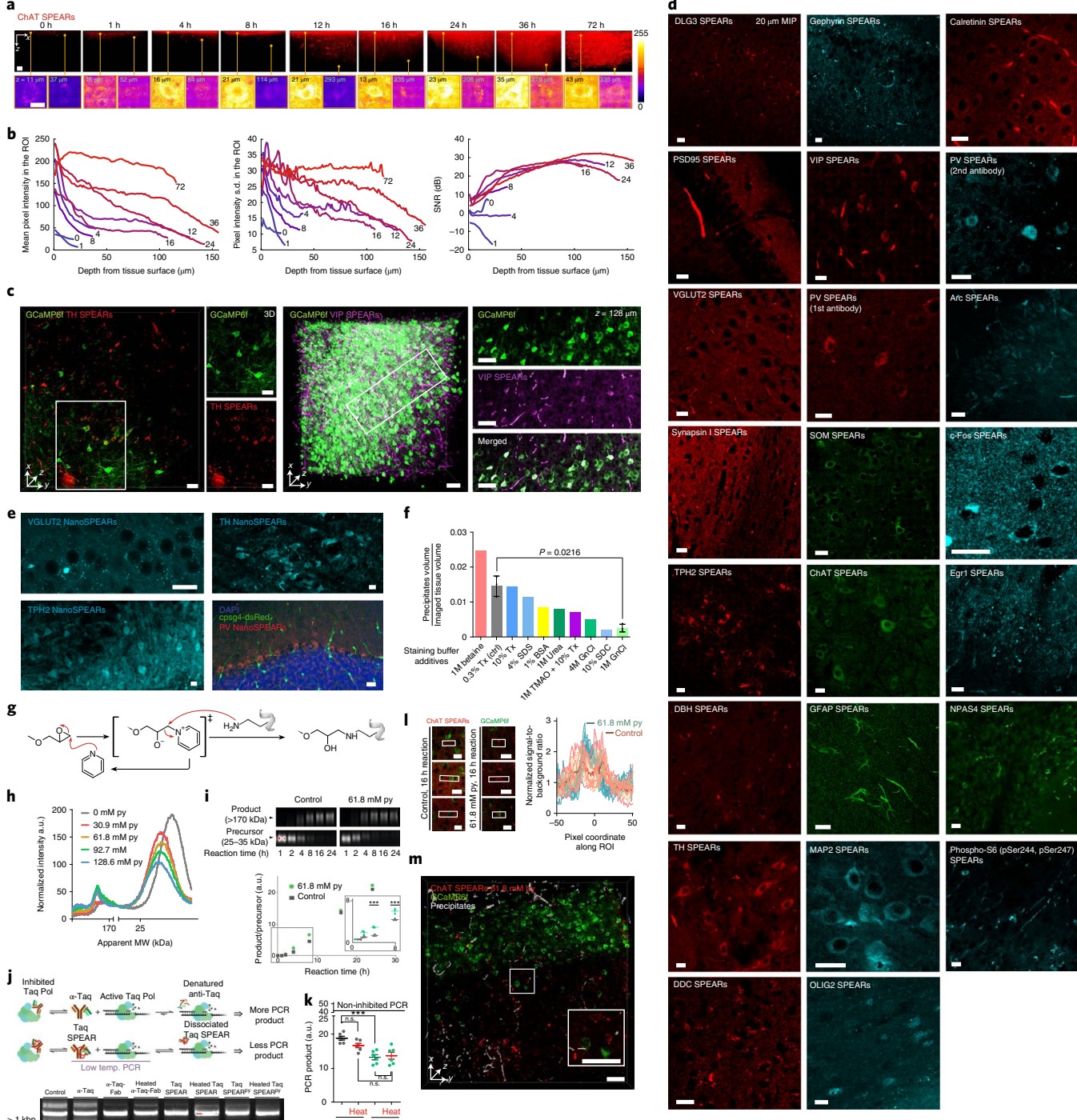

be readily performed across the whole brain (Fig. 4l and Extended Data Fig. 8d). Importantly, such tractography analysis would have been impossible without the ThICK-staining method due to locally restricted penetration of standard anti-ChAT antibodies, leading to inhomogeneous labeling of ChAT-positive fibers even with prolonged incubation and high antibody concentrations (Extended Data Fig. 6f–i).

SPEARs are also compatible with solvent-based clearing methods. Here, we demonstrate a formalin-fixed mouse whole-brain sample processed using the iDISCO method[15], ThICK-stained with parvalbumin SPEAR[py] for 72 h, cleared in benzyl alcohol–benzyl benzoate (BABB) and imaged with mesoscale selective plane illumination microscopy[16] (Fig. 4m,n, Extended Data Fig. 9a and Supplementary Video 2). Brainwide parvalbumin staining was obtained with identifiable somas in regions from the cortex, to deep structures such as thalamic and brainstem nuclei (Fig. 4o–q, Extended Data Fig. 9b–d and Supplementary Video 2). The thorough staining enabled us to quantify parvalbumin-positive cells in the whole cerebellar hemisphere (Fig. 4r, Extended Data Fig. 9b and Supplementary Video 2). We also benchmarked this iDISCO-ThICK-BABB method with tyrosine hydroxylase SPEARs in adult mouse hemibrains (Extended Data Fig. 6j) and noted the homogeneous staining of tyrosine hydroxylase-positive neurons throughout the sample (Extended Data Fig. 6k, l), in stark contrast to conventional immunostaining, in which the anti-tyrosine hydroxylase antibodies were mostly deposited at the tissue surface (Extended Data Fig. 6k,l and Supplementary Video 3). Note that due to tissue shrinkage with post-staining BABB clearing, the actual staining depths achieved with ThICK staining were in fact larger than that apparent on the images shown.

**Stringent benchmarking of immunolabeling depths.** Finally, we ThICK-stained mouse coronal brain slices and hemibrain samples against MAP2 and parvalbumin, and compared the signal intensity and homogeneity across tissue depth, against that obtained with iDISCO or CUBIC-HistoVIsion using stringent benchmarking experiments (Fig. 5, Methods).

In a benchmarking experiment we stained a 2-mm-thick tissue against the diffusely localized dendritic marker MAP2 and performed low-magnification imaging, such that all voxels can be analyzed for signal intensity with respect to distance from the nearest tissue surface. iDISCO staining with anti-MAP2 antibody resulted in higher signal intensities near the tissue surface, whereas ThICK staining with MAP2 SPEARs achieved stronger staining deep in the tissue and better signal homogeneity across depth (Fig. 5a,b).

In a separate experiment we first bulk-stained a mouse hemibrain against parvalbumin with various deep immunostaining protocols, and then repeated this with conventional counter-staining after cutting the tissue near its centroid (cut-staining) (Fig. 5c,d). We found that CUBIC-HistoVIsion achieved the highest penetration depths while signal homogeneity across depth was best with SPEARs and ThICK staining (Fig. 5e,f, right). Meanwhile, as expected, the cut-staining produced higher signal intensities near the cut surface, where the antibodies diffused in, for both the iDISCO and SPEARs experimental sets (Fig. 5f, left). Despite repeated attempts we were unable to obtain cut-staining-positive signals with CUBIC-HistoVIsion-processed samples, perhaps due to post-fixation and residual denaturants in the tissue. On a separate note, the overall background of bulk staining was higher for ThICK staining with SPEARs than for other staining techniques, which may explain the weaker signal to background ratios for ThICK staining (Fig. 5e).

To test the scalability of ThICK staining with SPEARs, we tested and verified that SPEARs can withstand prolonged (4 week) heat-accelerated immunostaining (Extended Data Fig. 10a), a timeframe sufficient for staining decimeter-scale tissues, although with some decrease in signal intensity. This should be noted for mega-scale staining experiments, in which the amount of SPEARs required should be increased to compensate for heat-induced degradation, ideally in a graded manner to avoid overloading the blood vessels with antibodies that may cause precipitates. ThICK staining can also be applied to other organs, which we demonstrated using human kidney tissues with SPEARs made from seven primary antibodies against renal cell type markers. These SPEARs had the same antigen-binding specificity as their antibody counterparts, and 16 h ThICK staining showed the expected staining patterns based on the Human Protein Atlas[17] (Extended Data Fig. 10b).

## Discussion

To date, four other approaches have convincingly demonstrated deep immunostaining: the first one is the delivery of antibodies by perfusion through vessels (for example, vDISCO[18]); the second is the use of gradual changes in detergent micelle concentrations that enable antibodies to diffuse into the tissue, after which they bind deeply located antigens with buffer acidification over time (that is, eFLASH[19]); the third is the conversion of tissues into elastic gels that can be stretched into a thin film to reduce diffusion distance (that is, ELAST[20]); and the last one is the use of non-specific protein denaturants to make antibody–antigen binding less favorable, thereby facilitating deeper antibody diffusion into the tissue (that is, CUBIC-HistoVIsion[21]). Although each of these approaches

---

**Fig. 4 | Application of SPEARs in ThICK staining of human brain and mouse whole-brain tissues with different tissue clearing techniques. a**, Protocol for a 5-mm-thick human brainstem block ThICK-stained with tyrosine hydroxylase (TH) SPEAR[py]. **b**, Image stack overview with segmented neurons color coded by cell volume. Scale bars, 500 μm. **c**, Magnified x–z view of the white boxed area in **b**. Scale bar, 100 μm. **d**, TH-positive neurons in optimized conventional immunostaining with 100 μl anti-TH antibody for 2 weeks (left) versus ThICK staining with 30 μl TH SPEAR[py] for 2 days (right). Scale bars, 500 μm. **e**, Distance of segmented neurons in **d** from the nearest tissue surface in conventional versus ThICK staining. Upper panel: each data point represents one segmented neuron (n = 707 and 2,828 for anti-TH antibody and TH SPEAR[py], respectively). The black lines represent the median and the error bars represent the interquartile range. Lower panel: cumulative distribution of the distance. **f**, Protocol for a SHIELD-protected whole adult mouse brain ThICK-stained with ChAT SPEAR. **g**, Overview of the rendered image volume. Excerpted cross-sectional and volume rendering views are shown in **h–k**, with insets showing magnified views. Scale bar, 2 mm. **h**, Volume rendering of cholinergic fibers traversing the basolateral amygdala (Am) with the internal capsule (IC) and external capsule (EC) included. Scale bar, 200 μm. **i**, Cholinergic neurons (arrowheads) in hippocampal CA1. Scale bar, 100 μm. Inset: magnified view of the white boxed area. Scale bar, 20 μm. **j**, Pedunculopontine nucleus (highlighted with dotted line). Scale bar, 200 μm; inset, 20 μm. **k**, Horizontal diagonal band of Broca (HDB). Upper panel, 100 μm z-MIP; lower panel, excerpted single section. Scale bars: 200 μm (upper), 100 μm (lower). acom, anterior commissure. **l**, Whole-brain tractography of cholinergic fibers, color coded for passage through major anatomical structures (upper panel), origins from major cholinergic cell groups or passage through major tracts (lower panel). **m**, Protocol for whole adult mouse brain ThICK-stained with parvalbumin SPEAR[py] with iDISCO tissue processing and BABB clearing. **n**, Overview of the rendered image volume. Scale bar, 1 mm. **o–r**, Excerpted views, including reticular thalamic nucleus (RT, **o**), cortex and hippocampus (Ctx and Hp, **p**), periaqueductal gray region (PAG, **q**), and cerebellum (inset, **r**). Scale bars: **o–q**, 200 μm; **r**, 1,000 μm. In **r**, the spatial locations of all automatically identified PV-positive interneurons in a cerebellar hemisphere volume are shown, with each neuron represented by a spot. Inset, 3D-rendered image.

is successful, the first method is not applicable to tissues without intact attached vasculature (for example, many ex vivo samples), the second method, eFLASH, requires specialized equipment and a specialized set-up for each sample[19], the third method, ELAST, involves substantial tissue property modifications and expansion[20], and the last method, CUBIC-HistoVIsion, requires a long time

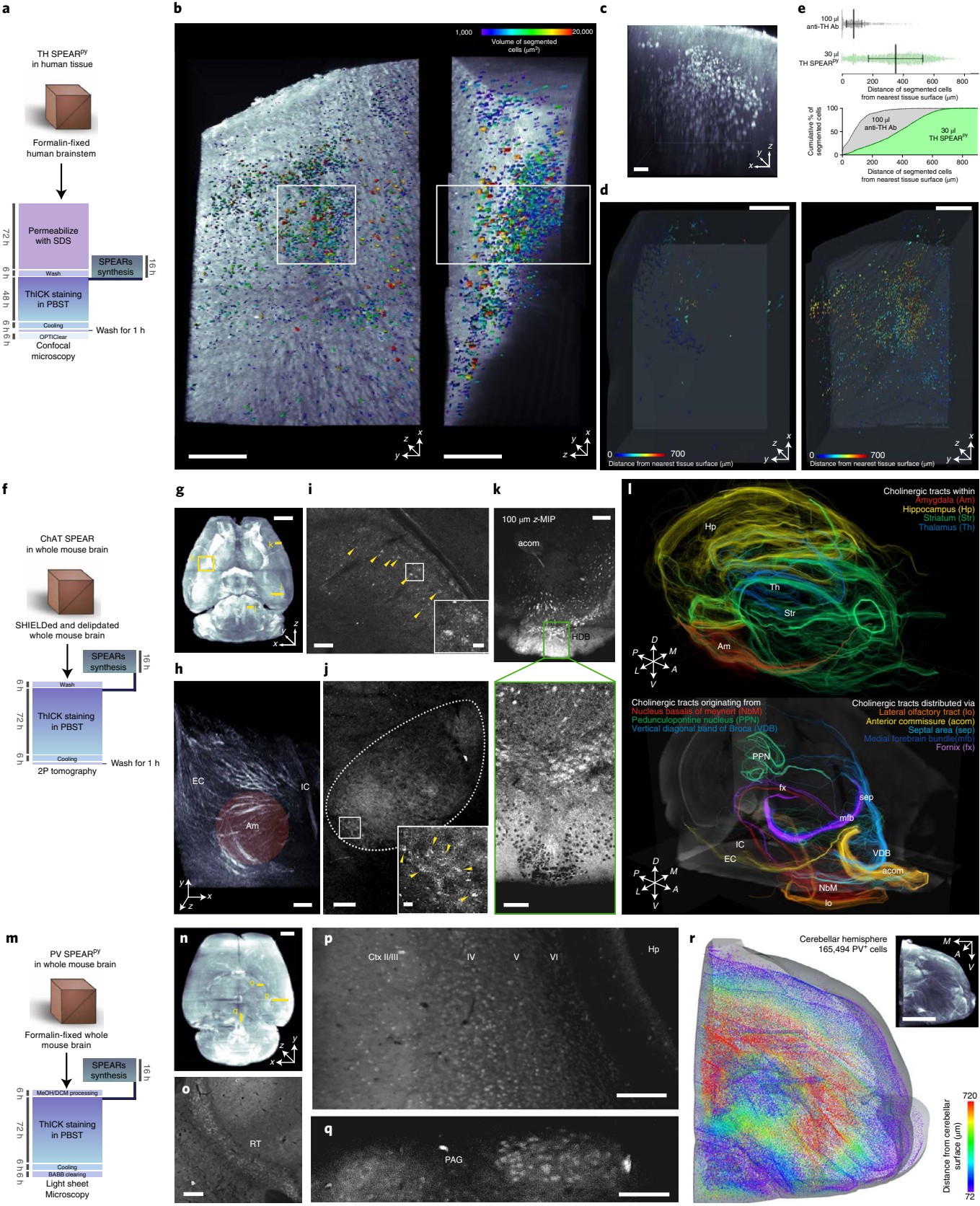

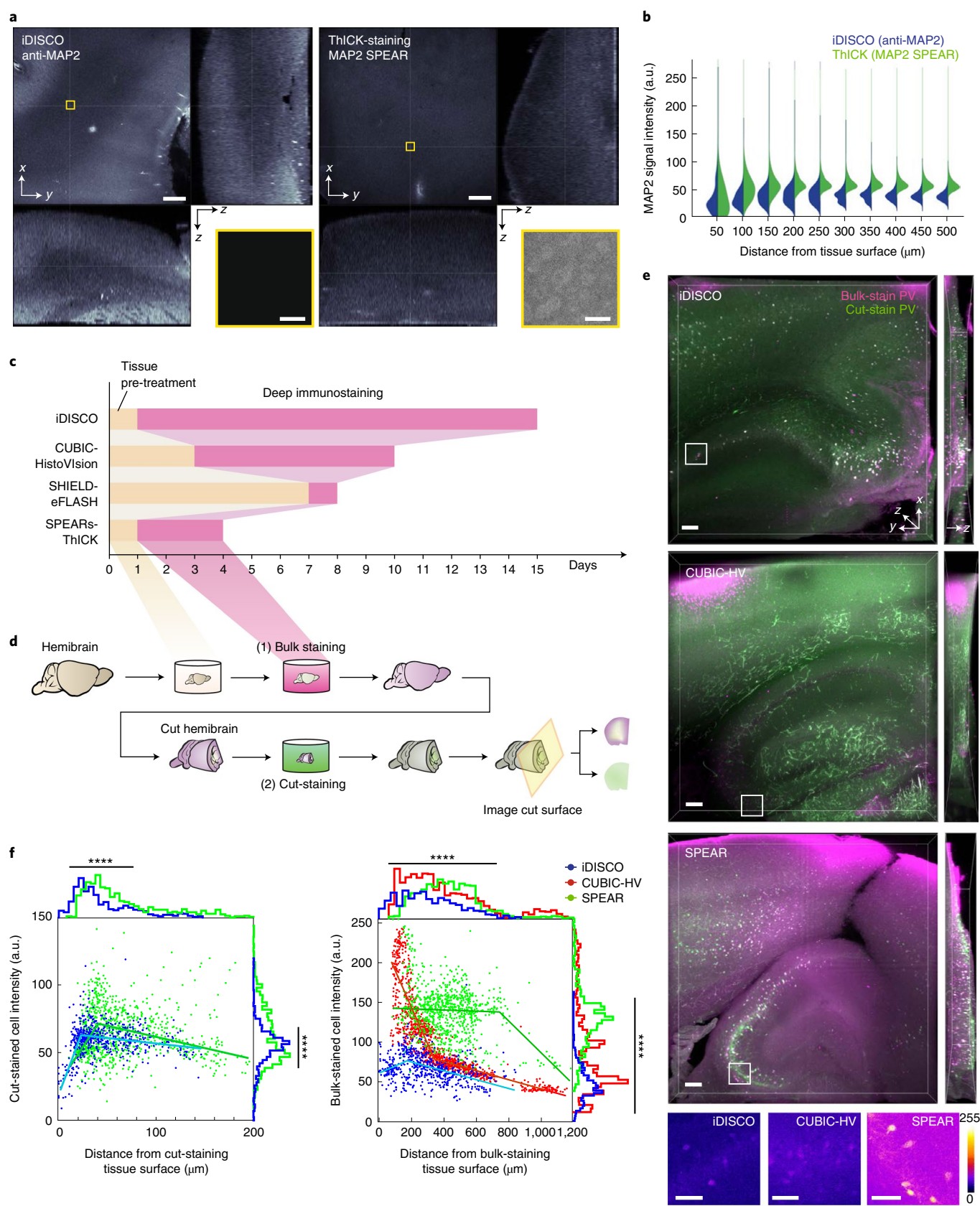

(1–8 weeks) for each round of antibody staining[21]. SPEAR-enabled ThICK staining provides an alternative approach that is rapid, does not require antibody perfusion or any specialized equipment, and complements diverse tissue preprocessing and/or clearing techniques (including conventional paraformaldehyde fixation, SHIELD, iDISCO, OPTIClear and BABB). In principle, SPEARs and ThICK staining can also be applied synergistically with all of these deep immunolabeling methods[18–21] to provide the benefits

**Fig. 5 | Benchmarking of ThICK staining. a**, Orthogonal views of a benchmarking experiment using anti-MAP2 immunostaining with confocal imaging under ×10 magnification. Insets shown are enlarged views of the centermost regions of the 3D image volumes. Scale bars: main panels, 200 μm; insets, 25 μm. **b**, Voxel-wise MAP2 signal intensity distribution versus distance from the nearest tissue surface. **c**, Timelines of deep immunostaining methods that had achieved whole adult mouse brain immunostaining, starting from a perfusion-fixed mouse brain. **d**, Schematic diagram of a stringent benchmarking experiment workflow. A fixed mouse hemibrain was processed through various deep immunostaining protocols with bulk staining (magenta), and then cut and re-stained with the same antibody with a different fluorophore (cut-staining, green) using conventional immunostaining. See Methods for details. Of the methods shown in **c**, we compared ThICK staining with iDISCO and CUBIC-HistoVIsion (CUBIC-HV). **e**, 3D imaging results obtained from the benchmarking experiment described in **d**. The cut surface is shown as an *x–y* plane (left square images), with the imaging depths of 200 μm shown as an *x–z* plane (rightmost rectangular pane). The imaging parameters were kept constant to ensure comparability. To ensure fairness in quantifying tissue diffusion distance, all tissues were cleared using the BABB method to obtain approximately the same degree of tissue shrinkage. Insets show the enlarged views of the white boxed areas in the overview images, which corresponds to a similar distance from the nearest bulk-staining tissue surface. Scale bars: main panels, 100 μm; insets, 50 μm. Color bar, pixel intensity. **f**, The mean signal intensity and distance of segmented cells from **e** from their respective staining tissue surfaces. Each dot represents a segmented cell and is color coded based on the bulk-staining protocol used. Superimposed lines on the scatter plots are segmental linear regression lines. Histograms of the distributions are shown alongside the scatter plots. \*\*\*\*two-tailed $P < 0.0001$ (unpaired *t*-test with Welch's correction, left; Brown–Forsythe ANOVA test, right).

of a more stable antibody and a higher macromolecule diffusivity. Although ThICK staining achieves whole-organ immunolabeling faster than other state-of-the-art methods, it still requires further optimization given the comparatively lower signal to noise ratio of staining. We performed ThICK staining for 1–3 days to demonstrate its main advantage with regard to speed, although the extended thermostability of SPEARs would in principle permit extended heat-accelerated immunostaining (for example, in centimeter-sized samples), albeit at the cost of using more antibodies.

In conclusion, we have established a fast, user-friendly deep immunostaining method that is readily implementable in most laboratories, and compatible with a wide range of commercially available antibodies and different classes of tissue preservation and clearing methods. This is based on a general method for thermostabilizing antibodies, which enables their applicability to heat-accelerated deep immunostaining. Compared with some previously reported attempts at antibody stabilization that involve more complex protein engineering[1–6], the production of SPEARs is simple, scalable and requires only the chemical modification of off-the-shelf antibodies. This enables immediate application to a large number of existing validated antibodies. Nonetheless, there is potential for improvement in antibody stabilization using such a chemical engineering approach, most notably in the relatively poor signal to noise ratio in immunostaining, as well as the incompatibility with some antibody–antigen pairs. Further work will also be necessary to investigate the intercompatibility of SPEARs for multiplexed immunostaining, other tissue clearing and immunostaining protocols, and molecular labeling techniques such as fluorescent in situ hybridization.

## Online content

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

## Methods

**Chemicals and reagents.** All chemicals were stored at temperatures as recommended by their vendors, protected from light, and used without further purification. The Fab fragments of secondary antibodies used were Alexa Fluor 594-conjugated donkey anti-goat IgG Fab fragments (cat. no. 705-547-003, Jackson ImmunoResearch), unconjugated donkey anti-mouse IgG Fab fragments (cat. no. 715-007-003, Jackson ImmunoResearch), Alexa Fluor 488-conjugated donkey anti-mouse IgG Fab fragments (cat. no. 715-547-003, Jackson ImmunoResearch), Alexa Fluor 594-conjugated donkey anti-mouse IgG Fab fragments (cat. no. 715-587-003, Jackson ImmunoResearch), Alexa Fluor 647-conjugated donkey anti-mouse IgG Fab fragments (cat. no. 715-607-003, Jackson ImmunoResearch), unconjugated donkey anti-rabbit IgG Fab fragments (cat. no. 711-007-003 Jackson ImmunoResearch), Alexa Fluor 488-conjugated donkey anti-rabbit IgG Fab fragments (cat. no. 711-547-003, Jackson ImmunoResearch), Alexa Fluor 594-conjugated donkey anti-rabbit IgG Fab fragments (cat. no. 711-587-003, Jackson ImmunoResearch), Alexa Fluor 647-conjugated donkey anti-rabbit IgG Fab fragments (cat. no. 711-607-003, Jackson ImmunoResearch), and Alexa Fluor 488-conjugated goat anti-rat IgG Fab fragments (cat. no. 112-547-003, Jackson ImmunoResearch). The lyophilized Fab fragments were reconstituted using distilled water to a concentration of $1\,\mathrm{mg\,ml^{-1}}$ and stored at $4\,°\mathrm{C}$ in aliquots. *Lycopersicon esculentum* lectin, Dylight 649 was purchased from Vector Laboratories (cat. no. DL-11781-1).

**Mouse brain tissues.** All experimental procedures were approved in advance by the Animal Research Ethics Committee of the Chinese University of Hong Kong (CUHK) and were carried out in accordance with the Guide for the Care and Use of Laboratory Animals. Male C57BL/6 and Thy1-GCaMP6f transgenic adult mice at least 2 months old were used. The mice were provided by the Laboratory Animal Service Center of CUHK and maintained at a controlled temperature ($22–23\,°\mathrm{C}$) with an alternating 12 h light–dark cycle with free access to standard mouse diet and water. The ambient humidity was maintained at <70% relative humidity. Formaldehyde-fixed and SHIELD-protected brain tissues were collected as previously described[13] (we recommend using SHIELD protection for samples labeled with fluorescent proteins). After adequate washing with PBST, tissues were stored at $4\,°\mathrm{C}$ in 1× PBS until use. A total of 12 animals were used to obtain the experimental results in Figs. 1, 2 and 3.

**Chemical stabilization of antibody–Fab fragment complex.** IgGs and their corresponding secondary Fab fragments were first reconstituted or diluted to a stock solution of $1\,\mathrm{mg\,ml^{-1}}$ with distilled water. For a typical 10-μl-scale synthesis, 1 μl stock IgG solution was thoroughly mixed with 1 μl stock Fab fragment solution and incubated at room temperature in a 0.2 ml polymerase chain reaction (PCR) tube for 10 min for complex formation. During this time, 200 μl P3PE (Erisys GE-38, Huntsman) was pipetted into a 1.5 ml Eppendorf tube using cut tips and the reverse pipetting technique. A total of 800 μl distilled water was then added and the tube was tightly capped and vigorously vortexed for 1 min to produce a homogeneous milky emulsion. The tube was then centrifuged at $15,000×g$ for 3 min at room temperature and allowed to sit at room temperature for not longer than 1 h. A total of 1 μl 1 M sodium carbonate pH 10 buffer followed by 5 μl water were then added to the formed IgG–Fab complex and thoroughly mixed. This was followed by the addition of 2 μl prepared P3PE supernatant and immediate vortexing of the tube. Using a thermocycler, the 10 μl reaction mixture was then reacted at $37\,°\mathrm{C}$ for up to 24 h followed by cooling to $4\,°\mathrm{C}$ and kept for not more than 24 h until further use. The reaction can be scaled up to 100 μl each time per PCR tube.

**SPEARs synthesis from commercially available primary antibodies.** Primary antibodies were reconstituted to 1× PBS with 0.1% w/v sodium azide to $1\,\mathrm{μg\,μl^{-1}}$ if lyophilized. The constituents of the storage buffer were reviewed for the presence of any additives containing primary amine groups (note, however, that having BSA as an additive is not an issue). If the storage buffer contains >0.1 M Tris, the antibodies were buffer exchanged to 1× PBS using ultracentrifugal filters with a molecular weight cut-off of 50 kDa (Amicon Ultra-0.5 centrifugal filter unit, cat. no. UFC505008, Millipore). We recommend against the use of non-purified antibodies in serum because non-specific IgGs would consume the Fab fragments. We do not routinely measure the concentration of antibodies in the commercial samples (given that it can be difficult to determine when protein preservatives were used), unless the signal intensities were suboptimal. In our study, 2 μl $0.05\,\mathrm{μg\,μl^{-1}}$ antibody complexed with 2 μl $1\,\mathrm{μg\,μl^{-1}}$ Fab fragment (that is, in molar excess) performed as satisfactorily as 2 μl $1\,\mathrm{μg\,μl^{-1}}$ antibody. Using a larger amount of antibody may help to further boost the signal but it may also increase the propensity to produce intravascular SPEAR precipitates.

SPEARs were freshly synthesized 1 day prior to staining. For a typical 10-μl-scale synthesis, 2 μl primary antibody and 1 μl corresponding Fab fragment at $2\,\mathrm{μg\,μl^{-1}}$ were thoroughly mixed and incubated at room temperature for 10 min to form the Ab–Fab complex. During this time, 200 μl P3PE was pipetted into a 1.5 ml Eppendorf tube using cut 1,000 μl tips and the reverse pipetting technique. A total of 800 μl distilled water was then added and the tube was tightly capped and vigorously vortexed for 1 min to produce a homogeneous milky emulsion.

A smaller scale preparation of P3PE would not be possible because it cannot be vortexed into a milky emulsion for solubilizing the crosslinker. The tube was then centrifuged at $15,000×g$ for 3 min at room temperature and allowed to sit at room temperature for not longer than 1 h. To the formed IgG–Fab complex, 1 μl 1 M sodium carbonate pH 10 buffer followed by 4 μl water were then added and thoroughly mixed. If SPEAR$^{py}$ is intended, 61.8 mM pyridine can be added to the mixture at this step. This can be conveniently done by making a stock solution of 1:9 v/v pyridine : water mixture, followed by adding 0.5 μl pyridine stock in place of water in the water addition step. This is followed by 2 μl freshly prepared P3PE supernatant and then the tube is rigorously vortexed. Using a thermocycler, the 10 μl reaction mixture was then reacted at $25\,°\mathrm{C}$ for 16 h followed by cooling to $4\,°\mathrm{C}$ and kept for not more than 24 h until further use. The reaction can be scaled up to 100 μl each time per PCR tube. The amount of SPEARs to be prepared depends on the density of the antigen and the size of the tissue, and should be optimized individually.

**ThICK staining protocol.** Fresh brain tissues were obtained and stored as described above. Tissues <300 μm thick were permeabilized for 1 day in PBST at $37\,°\mathrm{C}$, while larger samples were treated with 4% w/v SDS in 0.2 M borate buffer, pH 8.5, at $37\,°\mathrm{C}$ until optically transparent. The permeabilized sample was then washed thoroughly in PBST at $37\,°\mathrm{C}$ for 1 h three times. This is essential because any residual SDS will precipitate with the GnCl used in the next step. The washed sample was then equilibrated in approximately fivefold tissue volume of PBST with 1 M GnCl at $55\,°\mathrm{C}$ for 30 min, after which 10 μl SPEAR reaction mixture per 100 μl staining buffer was added to the staining solution and incubated at $55\,°\mathrm{C}$ for 16–72 h, depending on the sample thickness. For some antibodies the GnCl may be omitted if the staining did not produce much intravascular precipitates. As a general rule, we recommend increasing the staining duration by 8 h for every 200 μm staining depth, although it is likely that optimization of antibody concentration and staining duration will be required for individual antibody–antigen pairs. After incubating at $55\,°\mathrm{C}$, the sample was cooled to $25\,°\mathrm{C}$ and incubated further for 1 h. The sample was then washed in PBST to remove any residual GnCl and incubated in OPTIClear2 for 2 h (or OPTIClear for 6 h) at $37\,°\mathrm{C}$. The optically cleared sample can then be imaged.

**OPTIClear2.** OPTIClear2 is an improved version of the original hydrophilic optical clearing solution OPTIClear[14] featuring easier preparation and faster and better optical clearing (although OPTIClear is also compatible with SPEARs and ThICK staining). OPTIClear2 consists of 20% v/v 1-(3-aminopropyl)imidazole (cat. no. A14169, Alfa Aesar), 25% w/v 2,2′-thiodiethanol (cat. no. 166782, Sigma-Aldrich) and 32% w/v iopromide (Ultravist 370, Bayer) without further pH adjustments. OPTIClear consists of 20% w/v $N$-methylglucamine (cat. no. M2004, Sigma-Aldrich), 25% w/v 2,2′-thiodiethanol and 32% w/v iohexol (Nycodenz, cat. no. 1002424, Progen), with the pH adjusted to 7–8 using concentrated hydrochloric acid.

**Confocal microscopy.** Unless otherwise specified, confocal microscopy was performed using a Leica TCS SP8 confocal microscope. The excitation laser wavelengths used were 488 nm, 514 nm, 561 nm and 649 nm. Detection was done using gallium-arsenide-phosphide photomultiplier tubes through an HC PL APO ×10/0.40 CS2 (free working distance (FWD) 2.2 mm) or an HC PL APO ×20/0.75 CS2 (FWD 0.62 mm) objective. All imaging parameters were controlled for each set of experiments.

**Image processing and digital removal of intravascular SPEAR precipitates.** To digitally remove precipitate signals, an acquired multi-channel confocal z stack image in .lif format was first imported into Fiji (ImageJ)[22] and exported in .tiff format. The tiff image was then imported into Imaris (v9.0, Bitplane). A surface was then created based on the SPEARs staining channel, with local contrast settings, surface detail at 1.0 μm and maximum object diameter at 10.0 μm. The created surfaces were then filtered based on their specificities with regards to the vasculature and further edited manually. The created surfaces were then used to mask and set the intra-surface voxel intensity to zero.

**Image analysis for assessing staining homogeneity across z-depth.** Regions of interest (ROIs) of positive staining and background regions of ChAT SPEAR-stained mouse spinal cord sections were manually inspected and defined. The pixel intensities for the ROIs of each image slice were then profiled through the z-depth, with their means, standard deviations and signal to noise ratio (SNR) calculated using a custom-written MATLAB program (R2018b, MathWorks). For each image the SNR is defined as the ratio of the summed squared pixel intensity of ROIs of positive staining to that of the background regions ($r$), expressed in decibel (dB) (that is, $\mathrm{SNR}=10\log_{10}(r)$).

**Image analysis for quantification of intravascular SPEAR precipitates.** Definition and surface masking of intravascular SPEAR precipitates was performed as above for their digital removal using Imaris (v9.0, Bitplane). The total tissue volume was similarly measured with surface rendering and masking except that the surface detail was set at 5.0 μm and there was no subtraction of local contrast

or background. Intravascular SPEAR precipitate volumes and total tissue volumes were semi-automatically quantified based on the generated surfaces in Imaris and exported for analysis.

**Polyacrylamide gel electrophoresis and densitometry.** IgGs were complexed with their respective fluorescently labeled secondary antibody Fab fragments and crosslinked under various conditions as described in Fig. 1e and Extended Data Fig. 1 at a 10 µl reaction scale. The completed reaction mixture was then mixed with 3.5 µl 4× NuPAGE LDS sample loading buffer (cat. no. NP007, Invitrogen) and 0.5 µl beta-mercaptoethanol, heated to 95 °C for 10 min and cooled to room temperature. The samples were then loaded onto 1-mm-thick 10% SDS–polyacrylamide gels or 10% NuPAGE Bis-Tris gels (cat. no. NP0301BOX, Invitrogen) and run at a constant voltage of 90–120 V until the loading dye front reached the bottom of the gel. The gels were stained in InstantBlue Protein Stain (cat. no. ISB01L, Expedeon) overnight at room temperature with gentle shaking. Brightfield gel images were taken with a smartphone camera under ambient white light while fluorescence gel images were taken with a BioRad Gel Doc EZ System with automatic exposure. The obtained gel band intensities were measured using Fiji (ImageJ) with manually defined ROIs; the quantification procedures have been kept constant for all bands in the same set of experiments.

**Functional optimization of SPEAR antigen-binding capacity with an ELISA variant.** The 96-well ELISA plates (Nunc MaxiSorp flat-bottom plates, cat. no. 44-2404-21, ThermoFisher Scientific) were coated with a 10 mg ml⁻¹ stock solution of NeutrAvidin (cat. no. 31050, ThermoFisher Scientific) at room temperature for 24 h. Crosslinked complexes of unconjugated goat anti-rabbit antibodies (cat. no. A16112, Invitrogen) and biotin-conjugated donkey anti-goat antibody Fab fragment (cat. no. 705-067-003, Jackson ImmunoResearch) were prepared as above as 10 µl reaction mixtures and diluted 1:16,000 in PBST. Each well was coated with 100 µl NeutrAvidin solution at 1:100 dilution overnight at 4 °C. The NeutrAvidin-coated wells were washed with PBST for 5 min four times at room temperature, and then aspirated clean. The wells were then blocked with 5% w/v BSA at room temperature for 2 h and washed with PBST for 5 min four times. The wells were then coated with 100 µl diluted crosslinked antibody–Fab complex reaction mixture (diluted to 1:16,000) at room temperature for 2 h. The wells were then aspirated, washed for 5 min four times at room temperature with PBST, and incubated with 100 µl 0.01 mg ml⁻¹ rabbit IgG isotype (cat. no. 02-6102, ThermoFisher Scientific) at room temperature for 2 h. After aspiration and washing with PBST for 5 min four times, 100 µl HRP-conjugated goat anti-rabbit antibodies (diluted to 0.5 µg ml⁻¹ with 1× PBS, cat. no. P0448, Dako) were added and incubated at room temperature for 2 h. After aspiration and washing, 100 µl freshly made substrate solution of 2,2′-azino-bis[3-ethylbenzothiazoline-6-sulfonic acid]-diammonium salt (ABTS; cat. no. 10102946001, Sigma-Aldrich) at 5.71 mM with 0.03% w/w $H_2O_2$ in 1× PBS was added to the wells and incubated for 20 min at room temperature. The colorimetric readout was obtained on a Victor3 spectrophotometer (PerkinElmer) with a 0.1 s exposure at 405 nm. Data and statistical analyses were performed using Prism v8 (GraphPad).

**Functional assessment of SPEAR heat resistance with a hot-start PCR assay.** Mouse anti-Taq antibodies (cat. no. A01849, Genscript) were made into SPEARs as described above without (Taq SPEAR) or with 61.8 mM pyridine (Taq SPEARᵖʸ). The crosslinking duration was 4 h at 13 °C for both groups. The reaction products were purified using Amicon ultracentrifugal filters with a molecular weight cut-off of 30 kDa (cat. no. UFC503096, Millipore) and diluted to 1 unit Taq SPEARs or Taq SPEARsᵖʸ per 10 µl 1× PBS. The purified Taq SPEARs or Taq SPEARsᵖʸ were then split into two groups, one heated at 55 °C for 16 h and another stored at 4 °C until use. To set up the hot-start PCR functional assessment assay, 1 unit anti-Taq antibody, Taq SPEARs or Taq SPEARsᵖʸ (heated or non-heated) was mixed with 0.1 µM forward and reverse primers (forward, GCGTGCACTTTTTAAGGGAGG; reverse, CAGTATTTTTTCCGGTTGTAGCCC), 0.1 ng template (plasmid no. 25361, Addgene), and PCR master mix (cat. no. R004A, TaKaRa) as 30 µl reactions. The PCR thermocycling protocol is as follows: 55 °C for 30 s, 25 cycles of 55 °C for 1 min and 37 °C for 10 min, 3 cycles of 95 °C for 1 min and 60 °C for 1 min and 72 °C for 1 min, 72 °C for 10 min and 4 °C infinity hold. The PCR products were then analyzed on a 1% agarose gel and imaged using a BioRad Gel Doc EZ System with automatic exposure. The obtained gel band intensities were measured using Fiji (ImageJ) with manually defined ROIs, and the quantification procedures have been kept constant for all bands in the same set of experiments. Statistical analysis was performed using Prism v8 (GraphPad).

**Human pons section ThICK staining with SPEARsᵖʸ.** A human postmortem brainstem sample was fixed in 10% neutral buffered formalin for 3 weeks before washing and storage in PBS at 4 °C. A 5-mm-thick transverse section of the pons was then cut. Note that we used archived postmortem samples provided by the Anatomical and Cellular Pathology Department of our institution, and as per local regulations this did not require additional ethics approval. The pons slice was then sectioned sagittally and cut posterior to the medial lemniscus to obtain a subdivision containing the locus coeruleus. The sample was then incubated in 4% w/v SDS in 0.2 M borate buffer, pH 8.5, at 55 °C for 72 h and washed three times in

PBST for 2 h each. Meanwhile, tyrosine hydroxylase SPEARsᵖʸ (with Alexa Fluor 594) were prepared from 30 µl rabbit anti-tyrosine hydroxylase antibody (AB152, Millipore) with 16 h of incubation at 13 °C for 24 h. The washed sample was then placed in 3 ml fresh PBST with 300 µl tyrosine hydroxylase SPEARᵖʸ reaction mixture and ThICK-stained at 55 °C for 48 h. After ThICK staining the sample was cooled to 4 °C overnight, washed in PBST for 1 h at room temperature, and incubated in 20 ml OPTIClear at 37 °C overnight.

The stained and cleared sample was then imaged using a customized two-photon microscope (Scientifica) in tiled z stack mode (total acquisition field of view: 1,773 × 2,754 × 1,084 µm³) using a Thorlabs air objective (TL10X-2P, ×10, numerical aperture (NA) 0.5, working distance (WD) 7.77 mm). The z stacks were then imported into Zen Blue (ZEN 3.3, Carl Zeiss). Gaussian-blurred z stacks were then generated from each tile and used to correct shading inhomogeneity. The adjusted images were then background-subtracted. Stitching was performed using the ImageJ plugin BigStitcher[23]. The stitched image was imported into Imaris (v9.0, Bitplane) and cells were segmented with the local background contrast option and filtered based on volume and sphericity parameters, followed by manual refinements. To quantify cell distance from the nearest tissue surface, a surface was generated that encompassed all of the voxels outside of the tissue. A new channel with a linear gradient of voxel intensity that scales with the distance from the generated surface was created using the DistanceTransformation XTension in Imaris. The mean intensity of the distance transformation channel for each of the segmented cell surfaces was thus their distance from the nearest tissue surface.

For comparison, we used images acquired from a previously immunostained 1.5-mm-thick human pons sample that also contained the locus coeruleus[14]. The sample was fixed in 10% neutral buffered formalin for 3 weeks, permeabilized in 4% w/v SDS in 0.2 M borate buffer, pH 8.5, at 55 °C for 2 weeks and washed three times in PBST (for 2 h each). A total of 10 µl rabbit anti-tyrosine hydroxylase antibody was then added every day to the immunostaining PBST solution to a total of 100 µl, and the tissue was then incubated for an additional 4 days at 37 °C. The sample was then washed in PBST overnight, and Alex Fluor 594-labeled donkey anti-rabbit secondary antibody (cat. no. R37119, Invitrogen) was applied in a similar regimen. The sample was then washed and cleared in OPTIClear overnight. Imaging was performed with a Carl Zeiss LSM 780 confocal microscope using a Carl Zeiss air objective (Plan-Apochromat, ×10, NA 0.45), with an imaging depth of 1,500 µm (that is, full-thickness imaging). Stitching was performed alongside acquisition in Zen Black (ZEN 2.3, Carl Zeiss). Subsequent image analyses and cell segmentation were identical to the tyrosine hydroxylase SPEARᵖʸ-stained sample as described above.

**Whole mouse brain ThICK staining with SHIELD-protected sample.** SHIELD protection and delipidation of a whole adult mouse brain along with its spinal cord was performed as previously described[13]. A total of 100 µl anti-ChAT antibody (AB144P diluted version, Millipore) was then mixed with 10 µl Alexa Fluor 488-conjugated donkey anti-goat Fab fragments for the synthesis of ChAT SPEARs. The SHIELD-processed and delipidated mouse brain was thoroughly washed for 1 day and then ThICK-stained with the prepared ChAT SPEARs diluted in 10 ml PBST at 55 °C for 72 h. The brain was then cooled to and kept at 4 °C for 6 h and washed in PBST for 1 h at room temperature. The stained sample was then cleared with OPTIClear and pre-visualized in a custom-built light sheet microscope to confirm the presence of positive staining. The tissue was then immersed in PBS, stored at 4 °C for 1 week, and imaged using serial two-photon tomography.

**Serial two-photon tomography.** ThICK-stained whole mouse brains were imaged automatically with a custom-built two-photon microscope coupled to a Leica VT1000 vibratome. The excitation laser wavelength was 780 nm. To set up the sample for serial two-photon tomography, the sample was immersed in 5% gelatin at 40 °C for 4 h. The sample was then removed from the solution, dried and crosslinked using 4% paraformaldehyde (PFA) at 4 °C. This process rendered the brain samples suitably rigid for agar embedding and slicing. The microscopy system alternates between tile-scanning the surface of the tissue block using ScanImage 5.6.1 (ref. [24]) and removing a 100 µm section to expose the next area for imaging. The process was coordinated with a user function BakingTray (https://github.com/SainsburyWellcomeCentre/BakingTray) integrated into ScanImage. We imaged the brains at a resolution of 2.5 µm per pixel in the x–y plane and 10 µm in the z direction, the latter achieved by taking 10 optical planes for each 100 µm cut. Tile scans were assembled into stitched planes using StitchIt (https://github.com/SainsburyWellcomeCentre/StitchIt).

**Mouse whole-brain two-photon tomography image processing and tractography.** The mouse whole-brain serial two-photon images were processed in custom scripts written in MATLAB (R2020b, Mathworks), to adjust for the inhomogeneity of staining from the sample surface and equalize the pixel intensity mean and standard deviation across images from different optical sections. The output images were then imported into Imaris (v9.0, Bitplane) and 3D-rendered for the visualization of whole-brain staining patterns. Mesoscopic tractography analysis and visualization were performed by adapting the CLARITY-based Activity and Projection Tracking upon Recombination (CAPTURE) method[25]. In brief, the whole-brain volumetric images were downscaled by 10-fold in the x–y

plane and twofold in the *z* direction to obtain an isotropic volume. To smoothen the artifactual periodic variations of signal intensities for every 10 imaged planes, the image stack was first de-interleaved into odd- and even-numbered slices, separately Gaussian-filtered in the *z* direction with a sigma value of 1 pixel, grouped by *z* projection for every two slices, and interleaved before performing a final Gaussian filtering in the *z* direction with a sigma value of 1 pixel again. For each image voxel, the principal orientation of fiber tracts was determined by calculating the 3D structure tensor, followed by extracting its eigenvector with the smallest corresponding eigenvalue, using the MATLAB script provided in the CAPTURE website (http://capture-clarity.org/), with the derivative of the Gaussian sigma value set at 0.6, the Gaussian sigma value set at 2.3, and the angular threshold for tracking at 25°. Tracking was performed using the Diffusion Toolkit (http://trackvis.org/dtk/). Fiber tracts were visualized using TrackVis (http://trackvis.org/), with tract grouping done using manually annotated anatomical structures as seeds in TrackVis.

**Whole mouse brain ThICK staining with the iDISCO–BABB protocol.** A whole adult mouse brain was collected as described above with perfusion fixation using 4% PFA and post-fixed at 4 °C overnight. The next day the mouse brain was washed thoroughly in PBS for 1 h three times, followed by dehydration in graded methanol (20 ml 50% methanol, 90% methanol, 100% methanol and repeated 100% methanol for 10 min each, with gentle shaking). The dehydrated brain was then immersed in a 2:1 v/v mixture of dichloromethane : methanol for 10 min and transferred to 100% dichloromethane overnight at room temperature. In the meantime, 30 μl parvalbumin SPEARs$^{py}$ (that is, a total reaction volume of 300 μl) was synthesized using Alexa Fluor 647-labeled donkey anti-rabbit Fabs with 61.8 mM pyridine as the catalyst at 13 °C for 24 h. After incubating in dichloromethane, the brain was rehydrated in a series of 50% methanol and water for 10 min each, and finally washed in PBST for 10 min three times. The brain was then ThICK-stained in 10 ml PBST with 300 μl parvalbumin SPEAR$^{py}$ reaction mixture prepared as described, and incubated at 55 °C for 72 h. The brain was then cooled to 4 °C overnight and washed with PBST for 1 h. The brain was then dehydrated in graded methanol as described and cleared in a 1:2 v/v mixture of benzyl alcohol : benzyl benzoate.

**Selective plane illumination microscopy.** The stained and cleared sample was mounted on a custom-built mesoscale selective plane illumination microscope (mesoSPIM v5.1, ref. [16], www.mesospim.org). For the whole-brain images, the tissue was imaged using 647 nm excitation while the emission signal was acquired at ×1.25 zoom (2,048 × 2,048 pixels with 5.26 μm pixel size) at 10 μm optical sections.

**Mouse whole-brain SPIM image processing and spot detection.** The acquired parvalbumin SPEAR-stained image contained bubble artifacts that scattered the light sheets and thus had to be preprocessed. The .raw file output from the mesoSPIM was first imported into Fiji (ImageJ) and re-sliced in the *x*–*z* plane, and the re-sliced stack was processed with custom scripts written in MATLAB (R2020b, Mathworks), to adjust for the inhomogeneity of staining from the sample surface and equalize the pixel intensity mean and standard deviation across images from different optical sections. Stack normalization was then performed in Imaris and imported into Zen Blue. The *z* stacks were then duplicated, and one copy was median-blurred with *x*–*y* kernels of 37, two-dimensionally Gaussian-blurred with a sigma of 2.0, and brightness-adjusted to provide a reference for global multiplicative shading correction. The final adjusted image was again imported into Imaris and vascular precipitates were removed as described. Detection of cells was performed using the Spots function with the region growing option and an expected cell diameter of 10 μm.

**Benchmarking experiments against optimized conventional immunostaining.** Mouse brain slice phospho-S6 staining and hemibrain ChAT staining benchmark experiments were conducted as described above for SHIELD-processed samples, except with 1 μl rabbit anti-phospho-S6 antibodies and various dilutions of goat anti-ChAT AB144 antibodies (equivalent to a 10× concentrate of the AB144P antibody) used as indicated in the figures, and the use of a corresponding amount of Alexa Fluor 647-conjugated donkey anti-rabbit Fab fragments and Alexa Fluor 594-conjugated donkey anti-goat Fab fragments. No image processing was performed except for brightness and contrast adjustments for figure display, which were controlled across all comparison groups. Intensity measurements were performed using Fiji (ImageJ) and macros available from the Imperial College London FILM facility website (https://www.imperial.ac.uk/medicine/facility-for-imaging-by-light-microscopy/software/fiji/). Statistical analysis was performed using Prism v8 (GraphPad).

The mouse hemibrain tyrosine hydroxylase staining benchmark experiment was conducted as described above for the iDISCO–BABB method, except that 10 μl tyrosine hydroxylase SPEAR was used. The control sample was stained for the designated time using 10 μl tyrosine hydroxylase antibodies instead of tyrosine hydroxylase SPEAR. After mesoSPIM imaging, the 561 nm background channel was subtracted from the 647 nm signal channel. The brightness and contrast were enhanced to the same extent for both samples and visually compared without further imaging processing.

**Benchmarking experiments against CUBIC-HistoVIsion and iDISCO.** For benchmarking experiments, mouse brains were perfusion-fixed with 4% PFA as above and hemispheres were obtained. In all pairwise comparisons, tissues were obtained from the same mouse but different hemispheres or regions. The benchmarking experiment proceeded in two stages: the first stage involved whole hemispheric brain staining using iDISCO, CUBIC-HistoVIsion, or SPEARs with ThICK staining, with Alexa Fluor 647-conjugated Fab fragments as the fluorescent probe, and the second stage involved post-sectioning staining in PBST under the same conditions, with Alexa Fluor 488-conjugated Fab fragments as the fluorescent probe. The choice of the fluorophores served to minimize any signal crosstalk between the two stages, while ensuring minimal autofluorescence backgrounds and signal scattering of the first stage staining signals. Excess Fab fragments were used in the first stage staining to ensure complete blockage of the Fab fragment binding site to avoid post-staining by the second stage Fab fragments. Analogously, in the second stage an insufficient amount of Fab fragments was added to ensure their complete complexation by the antibodies. In the second stage staining we also pre-complexed the Fab fragments with the antibodies in a small volume before dilution in the staining buffer to avoid displacement of the first stage Fab fragments by local excesses of second stage Fab fragments .

For iDISCO the hemisphere was dehydrated in graded methanol and incubated in a 2:1 v/v mixture of dichloromethane : methanol overnight at room temperature, then rehydrated with graded methanol the next day. The brain was then washed for 15 min three times with PBST and then incubated with 5 μg anti-parvalbumin antibody and 6 μg Alexa Fluor 647-conjugated Fab fragment at 1:200 dilution in PBST. Staining was performed for 2 weeks at room temperature. The stained brain was then washed in PBST and processed in the second stage staining (see below).

For CUBIC-HistoVIsion the hemisphere was washed once for 15 min in PBS and treated with CUBIC-L (10% w/v *N*-butyldiethanolamine, 10% w/v Triton X-100 in water) at 37 °C for 2 weeks, during which the brain became optically clear to the naked eye. The brain was then washed with PBS and stained in HEPES-TSC buffer at room temperature with 5 μg anti-parvalbumin antibody and 6 μg Alexa Fluor 647-conjugated Fab fragment at 1:200 dilution. Staining was performed for 4 weeks at room temperature and included a post-staining incubation at 4 °C for 5 days to stabilize the signals. After staining, the sample was washed with 10% Triton X-100 in 0.1 M phosphate buffer followed by 0.1 M phosphate buffer for 1 h each, and post-fixed with 1% formaldehyde in 0.1 M phosphate buffer for 24 h at room temperature. The sample was then washed with PBS and processed in the second stage staining.

For SPEAR with ThICK staining, the hemisphere was washed once for 15 min in PBST and incubated in fresh PBST with 5 μg anti-parvalbumin antibody and 6 μg Alexa Fluor 647-conjugated Fab fragment at 1:200 dilution in PBST. Staining was performed for 3 days at 55 °C. The stained brain was then moved to room temperature for an additional 24 h incubation, washed for 15 min three times in PBST and stored at 4 °C until the second stage staining.

For the second stage staining, the samples were cut coronally at 2 mm anterior to the rostral end of the superior colliculus aided with a brain cutting matrix and washed in PBST. For each sample, in a separate tube, 5 μg anti-parvalbumin antibody was mixed with 4 μg Alexa Fluor 488-conjugated Fab fragments without dilution and incubated at room temperature for 10 min. After complexation was completed, they were diluted in 500 μl PBST and the cut brain samples were added to the mixture for staining at room temperature for 3 days. The samples were then washed for 15 min three times in PBST, dehydrated in graded methanol and cleared in a 1:2 v/v mixture of benzyl alcohol : benzyl benzoate. The samples were then imaged using confocal microscopy with dual channels at the cut surface using a ×10 objective. We choose BABB clearing due to the associated volume shrinkage to avoid confounding factors in the quantification of diffusion distances.

The obtained images were analyzed as follows. The tissue contour was segmented using Fiji (ImageJ), and the extra-tissue spaces were annotated to differentiate the source of first stage and second stage probe solution spaces, inclusive of the lateral ventricles. Diffusion distances were then calculated using the DistanceTransform XTension function in Imaris for each staining stage. The parvalbumin-positive cells from each staining stage were then segmented in Fiji (ImageJ) using a Laplacian of Gaussian (LoG) filter with $\sigma_L$ and $\sigma_G$ values of 10 and 5, respectively. The mean cell diffusion distances of the two staining stages as well as the parvalbumin signal intensities in their respective channels were then quantified and correlated (as shown in Fig. 5). Statistical analysis and graph plotting were performed using Prism v8 (GraphPad).

**Expression and purification of untagged nanobodies.** Bacterial expression plasmids pTP943 and pTP955 were a gift from Dirk Görlich (Addgene plasmid no. 104157, http://n2t.net/addgene:104157; RRID, Addgene_104157; and Addgene plasmid no. 104164, http://n2t.net/addgene:104164; RRID, Addgene_104164, respectively). Nanobodies were expressed in the cytoplasm of NEBExpress Competent *Escherichia coli* (New England Biolabs). Expression and purification protocols were adopted and modified from ref. [26]. In brief, a 50 ml preculture (2YT medium containing 50 μg ml$^{-1}$ kanamycin) was grown overnight. The culture was then diluted with fresh medium to 250 ml autoinduction 2YT medium containing 0.05% glucose, 0.2% α-lactose, 0.6% glycerol, 1% glycine, 40 mM Na$_2$HPO$_4$, 20 mM KH$_2$PO$_4$ and 1 mM MgSO$_4$. Bacteria were collected by

centrifugation, resuspended in a buffer (50 mM sodium phosphate, 300 mM NaCl and 10 mM imidazole) and lysed by sonication. The lysate was ultracentrifuged and filtered before being affinity purified via Ni²⁺ chelate affinity chromatography. After washing with a buffer (50 mM sodium phosphate, 300 mM NaCl, 50 mM imidazole), nanobodies were eluted with a buffer containing 500 mM imidazole. Buffer exchange to maleimide-labeling buffer (MLB; 20 mM sodium phosphate, 150 mM NaCl, 250 mM sucrose, 10 mM imidazole) using the PD-10 desalting column was performed, and the amino-terminal His14-bdNEDD8-tag was cleaved with 500 nM His6-NEDP1/SENP8 protease (R&D systems) for 1 h at 4 °C. The His14-bdNEDD8 tag and His6-tagged protease were then removed by reverse nickel chromatography. The unbound fraction containing the untagged nanobodies was stored in MLB with 10 mM dithiothreitol in liquid nitrogen.

**Site-specific fluorescent labeling of nanobodies with maleimide dyes.** Untagged nanobodies were buffer exchanged using the PD-10 desalting column to remove the reducing agent. For the labeling reaction, 5 μM nanobody was mixed at a protein molar ratio of 10–20 dyes (Biotium) per protein and incubated at room temperature for 2 h or 4 °C overnight. Free dye was separated from labeled nanobody using molecular weight cut-off columns. Fluorescently labeled nanobodies were aliquoted, snap-frozen in liquid nitrogen and stored at −80 °C.

**Reaction–diffusion modeling.** We performed reaction–diffusion modeling similar to that previously described[21], by solving the following radially symmetrical reaction–diffusion equation describing the diffusion of free antibodies into a cylindrical structure containing antigen targets:

$$\frac{\partial[\text{Ab}]}{\partial t} = D_{\text{eff}} \cdot \frac{1}{r} \cdot \frac{\partial}{\partial r}\left(r\frac{\partial[\text{Ab}]}{\partial r}\right) + k_{\text{off}}[\text{Ab}-\text{Ag}] - k_{\text{on}}[\text{Ab}][\text{Ag}]$$

$$\frac{\partial[\text{Ag}]}{\partial t} = k_{\text{off}}[\text{Ab}-\text{Ag}] - k_{\text{on}}[\text{Ab}][\text{Ag}]$$

$$\frac{\partial[\text{Ab}-\text{Ag}]}{\partial t} = -k_{\text{off}}[\text{Ab}-\text{Ag}] + k_{\text{on}}[\text{Ab}][\text{Ag}]$$

where [Ab], [Ag] and [Ab–Ag] are the spatial coordinate (r)-dependent and time (t)-dependent free antibody, unbound antigen and antibody–antigen complex concentrations, respectively. $D_{\text{eff}}$ is the free antibody diffusion coefficient, the value of which inside the cylinder is assumed to be one-seventh of that outside[27]. $k_{\text{off}}$ and $k_{\text{on}}$ are the antibody–antigen dissociation and association rate constants, respectively. $D_{\text{eff}}$, $k_{\text{off}}$ and $k_{\text{on}}$ were all taken to be temperature dependent, with values set as follows: at 22 °C, $D_{\text{eff}}$ (outside cylinder) = 1, $k_{\text{off}} = 1$ and $k_{\text{on}} = 1$; at 55 °C, $D_{\text{eff}}$ (outside cylinder) = 1.11 (calculated based on the $D_{\text{eff}}$ value at 22 °C using the Stokes–Einstein equation), $k_{\text{off}} = 60.75$ and $k_{\text{on}} = 2$ (extrapolated based on data from ref. [28]). Simulation was performed with t running from 0 to 1. To incorporate a temperature change during the process at $t = 0.95$, the $D_{\text{eff}}$, $k_{\text{off}}$ and $k_{\text{on}}$ parameter set at 22 °C was used for $t \in [0, 0.95)$ (first phase), and that at 55 °C was used for $t \in [0.95, 1]$ (second phase). For simplicity, we assumed that there is no denaturation during the heating phase (that is, simulating full protection against heat denaturation). To isolate the effect of $k_{\text{off}}$ and $k_{\text{on}}$ changes, $D_{\text{eff}}$ was kept constant at the 22 °C value. Similarly, to isolate the effect of $D_{\text{eff}}$ increase, $k_{\text{off}}$ and $k_{\text{on}}$ were kept constant at their 22 °C values. Initial and boundary conditions were as follows: initial [Ab] outside the cylinder (with radius 1) = 50; initial [Ab] inside the cylinder = 0; initial [Ag] inside the cylinder = 50; initial [Ab–Ag] = 0 both inside and outside the cylinder; [Ab] ≡ 50 at $r = 2$. The partial differential equations were solved using the function pdepe in MATLAB (R2021a, MathWorks) with 1,000 and 500 evenly discretized $r \in [0, 2]$ and $t \in [0, 1]$ values, respectively, and other options in default settings.

**Gel filtration chromatography.** The prepared SPEAR samples were separated by fast protein liquid chromatography (FPLC) on an ÄKTA pure 25 chromatography system (GE Healthcare) fitted with a Superdex 200 Increase 10/300 GL column (GE Healthcare). In brief, protein samples were first centrifuged at 13,000×g for 10 min at 4 °C to remove any precipitants. Then a 500 μl protein sample was loaded onto the column pre-equilibrated with 100 mM carbonate/bicarbonate buffer (pH 10.0) and eluted with the same column buffer at a flow rate of 0.4 mL min⁻¹. The absorbance of the column eluate was continuously monitored at 280 nm. Gaussian fitting on the obtained ultraviolet absorbance tracing over elution volumes was performed manually for peak detection. Molecular weight estimation was performed separately for IgG and Fab due to the non-linear molecular size calibration curve obtained, as well as significant deviations from predicted sizes for proteins reacted with crosslinkers.

**Comparison of standard immunostaining and ThICK staining in human kidney tissues.** Human postmortem kidney tissues were generously donated from a male donor without pathological evidence of renal diseases. The archived sample was provided by the Anatomical and Cellular Pathology Department of our institution, and as per local regulations additional ethics approval was not required. The postmortem delay while storing at −18 °C was 4 days, after which the cadaver was transfemorally perfused with 4% PFA and fixed at 4 °C for 3 days. The whole kidney was collected and further trans-arterially perfused with 4% PFA and fixed at 4 °C for 1 week, after which 1-mm-thick slices of tissues were obtained for immunostaining.

Pairs of adjacent tissue sections were obtained for the immunostaining of each antibody and corresponding SPEAR to avoid confounding of the immunostaining results by locoregional anatomical and pathological variations. Prior to immunostaining, the tissue sections of both groups were permeabilized with PBST for 1 day. For the control group, antibodies were diluted in PBST (using the vendors' highest recommended dilutions) for 16 h at room temperature, along with the same volume of 1 mg ml⁻¹ fluorescently labeled secondary Fab fragments. For the SPEAR group, a threefold higher amount of SPEARs for the corresponding antibodies was prepared (using the same Fab fragments as the control group), followed by ThICK staining with PBST at 55 °C for 16 h. The stained samples were then washed, cleared using the BABB method, and imaged under the same conditions. Tissue autofluorescence was achieved with 488 nm excitation and emission detection at 520–600 nm.

**Statistics and reproducibility.** Microscopy images shown in the Figures are representative images from varying numbers of experimental replicates with similar results. For the number of experiments performed, in Figs. 3a, 4 and 5a, once; Figs. 3e,m and 5e, twice; Fig. 3c,d, three times. No statistical method was used to predetermine sample size. No data were excluded from the analyses. The experiments were not randomized. The Investigators were not blinded to allocation during the experiments or the outcome assessment.

**Reporting summary.** Further information on research design is available in the Nature Research Reporting Summary linked to this article.

## Data availability
The numeric data for applicable plots are available in the Source Data file. The raw imaging data in this paper are too large for public deposit and will be made available upon reasonable request to the corresponding authors (H.M.L. or H.K.). Source data are provided with this paper.

## Code availability
The code for analysis used in this paper is available at GitHub (AkaBurri/ProjSPEARs, https://github.com/AkaBurri/ProjSPEARs).

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

## Acknowledgements
The authors thank B. Yung, F. Yau, A. Miu, R. Chau and D. Ieong for administrative support for the project. The authors also thank S. Dang, Y. Qin and H. Liu from the Hong Kong University of Science and Technology for their infrastructural support in gel filtration chromatography. This work was funded by a Croucher Innovation Award (CIA20CU01) from the Croucher Foundation (H.K.); Faculty Innovation Awards (FIA2020/B/01, FIA2017/B/01) from the Faculty of Medicine, CUHK (H.M.L., H.K.); the Gerald Choa Neuroscience Centre, the Margaret K. L. Cheung Research Centre for Parkinsonism Management, Faculty of Medicine, CUHK (H.M.L., V.C.T.M., H.K.); the Midstream Research Programme for Universities (MRP/048/20) of the Innovation and Technology Council (ITC) of Hong Kong (H.M.L., H.K.); the Innovation and Technology Support Programme (ITS/435/18FX) of the ITC of Hong Kong (L.Y.C.Y., J.Y.K.C.); the General Research Fund (GRF14108818) of the University Grants Committee (UGC) of Hong Kong (J.Y.K.C.); the Collaborative Research Fund (C6027-19GF and C7074-21GF) and the Area of Excellence Scheme (AoE/M-604/16) of the UGC of Hong Kong (H.K.); and the Excellent Young Scientists Fund from the National Natural Science Foundation of China (H.K.).

## Author contributions
H.M.L. conceived and designed the antibody thermostabilization method, thermally controlled immunostaining and other experiments in this study. Y.T., Z.Y.H.L. and J.C.N.Y. contributed to reaction condition optimization, with H.M.L. providing oversight. H.M.L., Y.T., Z.Y.H.L. and H.K.T.W. carried out the immunostaining experiments.

D.C.W.C. and L.Y.C.Y. assisted in tissue imaging. C.C.Y.C. cloned, expressed, purified and synthesized fluorescently labeled nanobodies. T.Y.W. and H.H.N.L. designed and performed gel filtration chromatography analysis. H.-K.N. and J.Y.K.C. contributed tissue samples for protocol testing. W.K.K.W., S.H.W., K.-W.K., Y.-K.W, V.C.T.M and J.Y.K.C. contributed to technical discussions and data interpretation. R.A.A.C. and T.D.M.-F. contributed to the imaging of whole-brain samples. H.M.L. performed image analyses. H.M.L. and H.K. oversaw the project and wrote the manuscript with input from all authors.

## Competing interests

CUHK has filed a patent application based partly on the invention described in this paper. H.M.L., Z.Y.H.L. and H.K. are the inventors of the patent. All other authors have no competing interests.

## Additional information

**Extended data** are available for this paper at https://doi.org/10.1038/s41592-022-01569-1.

**Correspondence and requests for materials** should be addressed to Hei Ming Lai or Ho Ko.

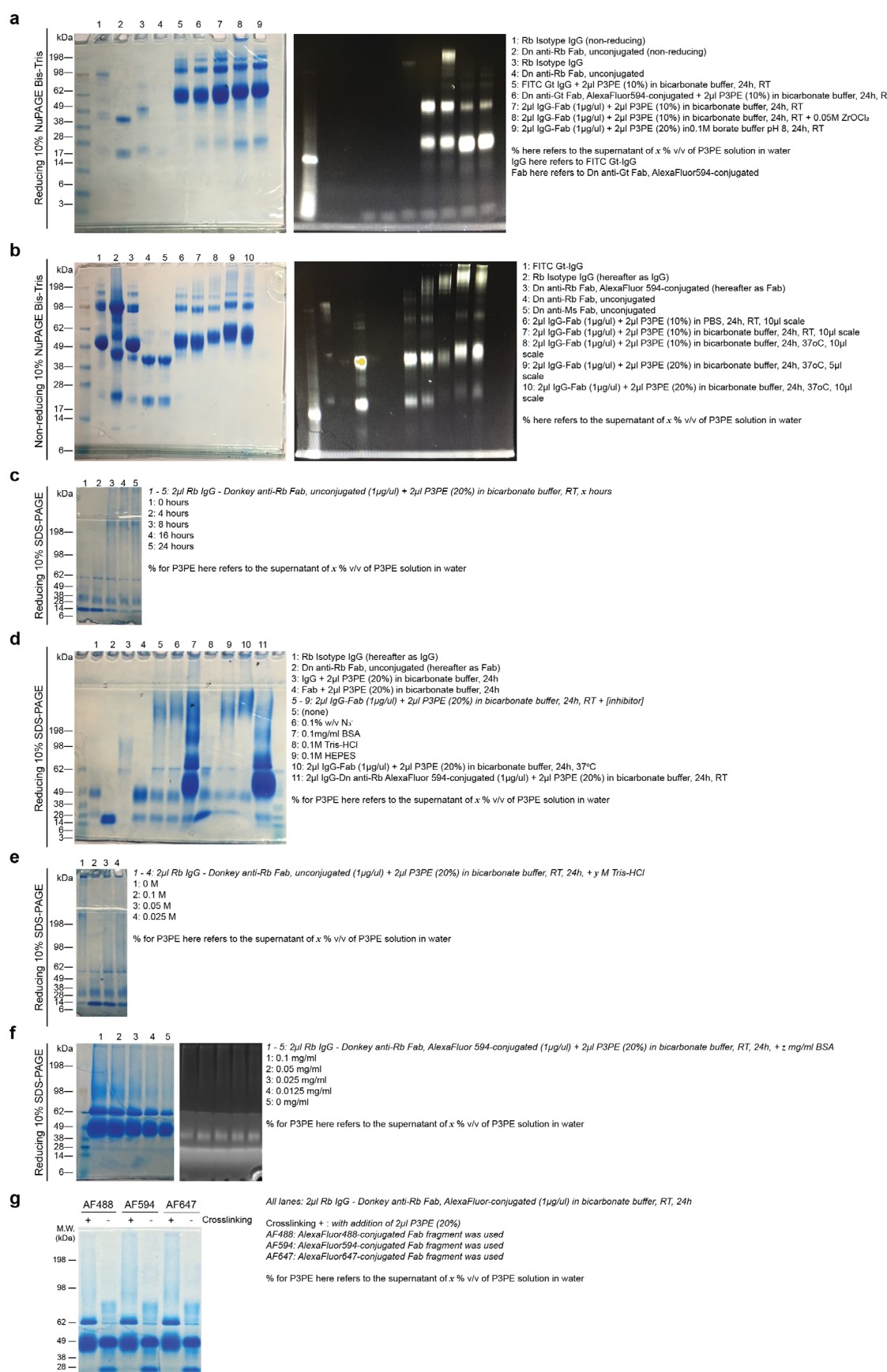

**a**

Reducing 10% NuPAGE Bis-Tris

1: Rb Isotype IgG (non-reducing)
2: Dn anti-Rb Fab, unconjugated (non-reducing)
3: Rb Isotype IgG
4: Dn anti-Rb Fab, unconjugated
5: FITC Gt IgG + 2µl P3PE (10%) in bicarbonate buffer, 24h, RT
6: Dn anti-Gt Fab, AlexaFluor594-conjugated + 2µl P3PE (10%) in bicarbonate buffer, 24h, RT
7: 2µl IgG-Fab (1µg/ul) + 2µl P3PE (10%) in bicarbonate buffer, 24h, RT
8: 2µl IgG-Fab (1µg/ul) + 2µl P3PE (10%) in bicarbonate buffer, 24h, RT + 0.05M ZrOCl₂
9: 2µl IgG-Fab (1µg/ul) + 2µl P3PE (20%) in0.1M borate buffer pH 8, 24h, RT

% here refers to the supernatant of x % v/v of P3PE solution in water
IgG here refers to FITC Gt-IgG
Fab here refers to Dn anti-Gt Fab, AlexaFluor594-conjugated

**b**

Non-reducing 10% NuPAGE Bis-Tris

1: FITC Gt-IgG
2: Rb Isotype IgG (hereafter as IgG)
3: Dn anti-Rb Fab, AlexaFluor 594-conjugated (hereafter as Fab)
4: Dn anti-Rb Fab, unconjugated
5: Dn anti-Ms Fab, unconjugated
6: 2µl IgG-Fab (1µg/ul) + 2µl P3PE (10%) in PBS, 24h, RT, 10µl scale
7: 2µl IgG-Fab (1µg/ul) + 2µl P3PE (10%) in bicarbonate buffer, 24h, RT, 10µl scale
8: 2µl IgG-Fab (1µg/ul) + 2µl P3PE (10%) in bicarbonate buffer, 24h, 37oC, 10µl scale
9: 2µl IgG-Fab (1µg/ul) + 2µl P3PE (20%) in bicarbonate buffer, 24h, 37oC, 5µl scale
10: 2µl IgG-Fab (1µg/ul) + 2µl P3PE (20%) in bicarbonate buffer, 24h, 37oC, 10µl scale

% here refers to the supernatant of x % v/v of P3PE solution in water

**c**

Reducing 10% SDS-PAGE

1 - 5: 2µl Rb IgG - Donkey anti-Rb Fab, unconjugated (1µg/ul) + 2µl P3PE (20%) in bicarbonate buffer, RT, x hours
1: 0 hours
2: 4 hours
3: 8 hours
4: 16 hours
5: 24 hours

% for P3PE here refers to the supernatant of x % v/v of P3PE solution in water

**d**

Reducing 10% SDS-PAGE

1: Rb Isotype IgG (hereafter as IgG)
2: Dn anti-Rb Fab, unconjugated (hereafter as Fab)
3: IgG + 2µl P3PE (20%) in bicarbonate buffer, 24h
4: Fab + 2µl P3PE (20%) in bicarbonate buffer, 24h
5 - 9: 2µl IgG-Fab (1µg/ul) + 2µl P3PE (20%) in bicarbonate buffer, 24h, RT + [inhibitor]
5: (none)
6: 0.1% w/v N₃⁻
7: 0.1mg/ml BSA
8: 0.1M Tris-HCl
9: 0.1M HEPES
10: 2µl IgG-Fab (1µg/ul) + 2µl P3PE (20%) in bicarbonate buffer, 24h, 37ºC
11: 2µl IgG-Dn anti-Rb AlexaFluor 594-conjugated (1µg/ul) + 2µl P3PE (20%) in bicarbonate buffer, 24h, RT

% for P3PE here refers to the supernatant of x % v/v of P3PE solution in water

**e**

Reducing 10% SDS-PAGE

1 - 4: 2µl Rb IgG - Donkey anti-Rb Fab, unconjugated (1µg/ul) + 2µl P3PE (20%) in bicarbonate buffer, RT, 24h, + y M Tris-HCl
1: 0 M
2: 0.1 M
3: 0.05 M
4: 0.025 M

% for P3PE here refers to the supernatant of x % v/v of P3PE solution in water

**f**

Reducing 10% SDS-PAGE

1 - 5: 2µl Rb IgG - Donkey anti-Rb Fab, AlexaFluor 594-conjugated (1µg/ul) + 2µl P3PE (20%) in bicarbonate buffer, RT, 24h, + z mg/ml BSA
1: 0.1 mg/ml
2: 0.05 mg/ml
3: 0.025 mg/ml
4: 0.0125 mg/ml
5: 0 mg/ml

% for P3PE here refers to the supernatant of x % v/v of P3PE solution in water

**g**

AF488 AF594 AF647

All lanes: 2µl Rb IgG - Donkey anti-Rb Fab, AlexaFluor-conjugated (1µg/ul) in bicarbonate buffer, RT, 24h

Crosslinking + : with addition of 2µl P3PE (20%)
AF488: AlexaFluor488-conjugated Fab fragment was used
AF594: AlexaFluor594-conjugated Fab fragment was used
AF647: AlexaFluor647-conjugated Fab fragment was used

% for P3PE here refers to the supernatant of x % v/v of P3PE solution in water

**Extended Data Fig. 1 | See next page for caption.**

**Extended Data Fig. 1 | SDS–PAGE analysis on SPEAR formation. a–c**, Establishment of P3PE-crosslinked IgG–Fab complex electrophoretic patterns and initial optimization of reaction conditions for yield. **a**, Reducing and **b**, non-reducing SDS-NuPAGE analysis of P3PE-crosslinked IgGs, Fabs and their complexes under various conditions and their electrophoretic patterns. **c**, Time course of P3PE crosslinking of IgG–Fab complexes. The tested reaction conditions are listed on the right. All experiments were repeated twice with similar results. **d–g**, Testing the tolerance of P3PE-crosslinking reaction towards common additives in commercially supplied antibodies using reducing SDS–PAGE. **d**, Screening for additives that inhibit P3PE crosslinking of IgG–Fab complexes. **e**, and **f**, Titration of Tris (**e**) and BSA (**f**) and their effects on P3PE-crosslinking reaction. The tested reaction conditions are listed on the right. **g**, Testing the effects of various fluorophores on the P3PE-crosslinking reaction. All experiments were repeated twice with similar results.

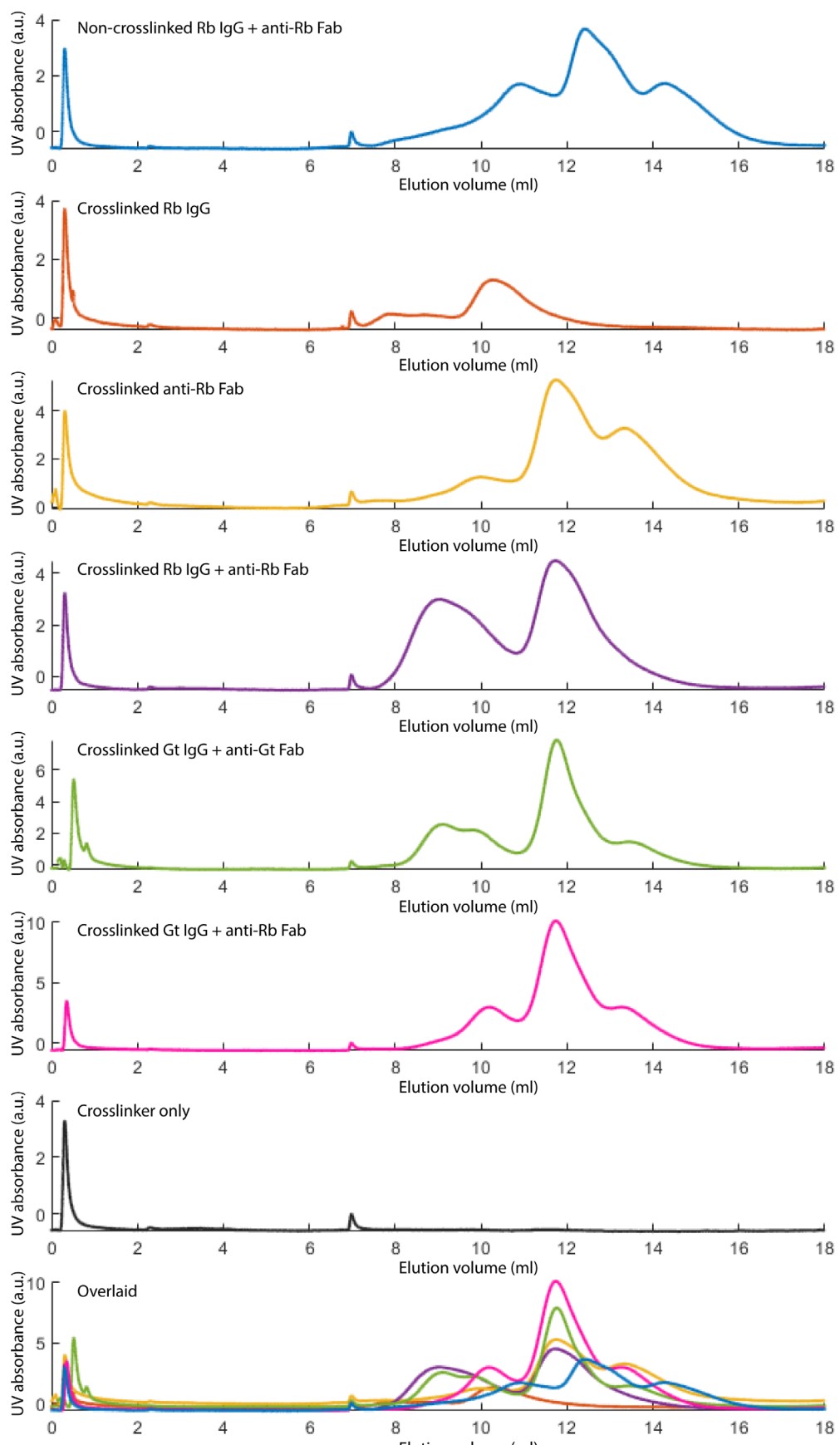

**Extended Data Fig. 2 | Gel filtration chromatography of products formed under optimized crosslinking conditions.** Estimated molecular weights (in kDa) are shown for all major detected peaks.

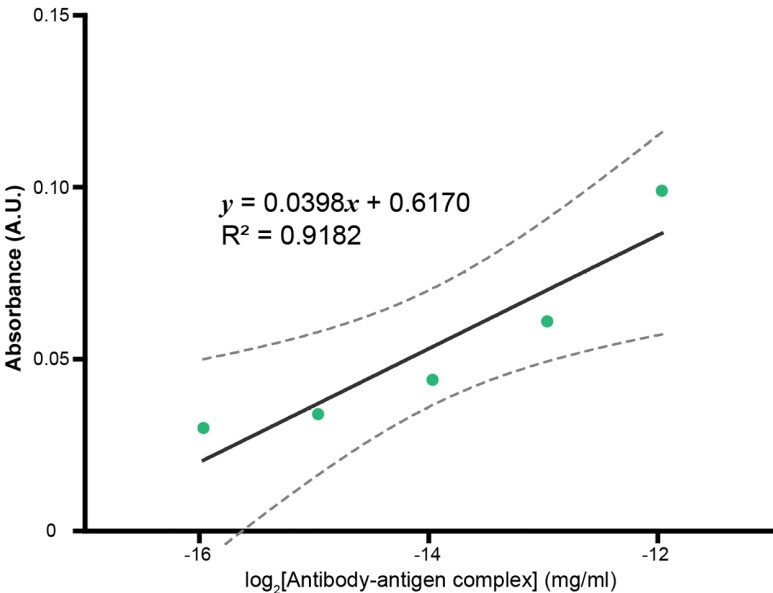

**Extended Data Fig. 3 | Performance of the ELISA variant for functional optimization of SPEARs.** The absorbance response of ABTS is linear over four orders of antigen dilution. The line of best fit on linear regression (black) and its equation are shown. Dashed lines: 95% confidence interval of regression.

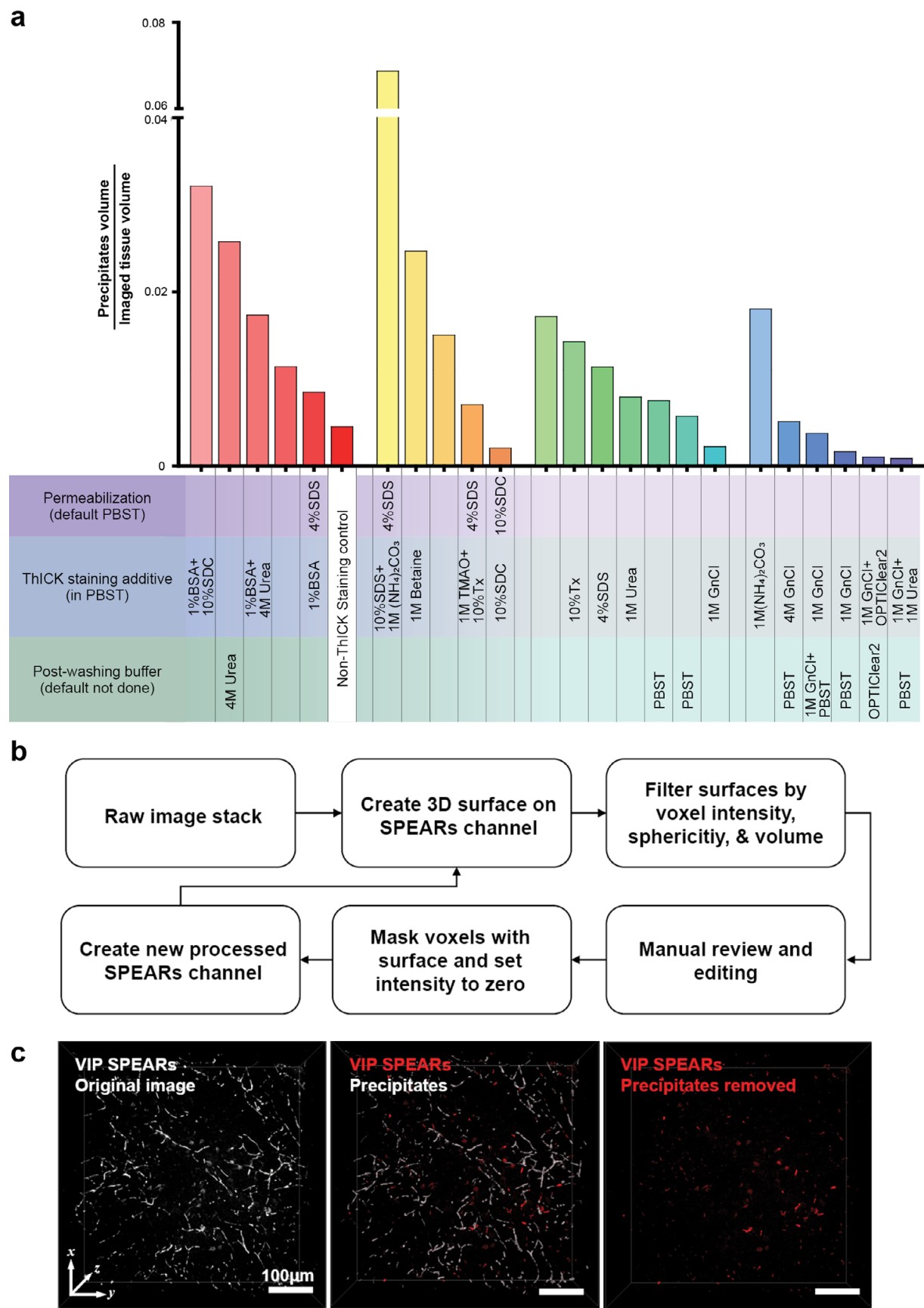

**Extended Data Fig. 4 | See next page for caption.**

**Extended Data Fig. 4 | Strategies for reducing intravascular precipitates during ThICK staining. a**, Optimization of ThICK-staining protocol. Additives were added in various incubation steps while vascular precipitation of SPEARs was globally quantified for each imaged tissue stack and normalized against the imaged tissue volume (see Methods). The optimization was performed iteratively for four rounds (grouped in colors). For all experiments, permeabilization was performed at 37 °C for 1 day, ThICK staining was performed at 55 °C for 16 hours, and post-washing was performed at room temperature for 1 day. Abbreviations: BSA, bovine serum albumin; GnCl, guanidinium chloride; PBST, 1× phosphate-buffered saline with 0.3% v/v Triton X-100; TMAO, trimethylamine oxide; Tx, Triton X-100; SDC, sodium deoxycholate; SDS, sodium dodecyl sulfate. **b**, and **c**, Post-imaging removal of intravascular SPEAR precipitates. **b**, Approach for segmenting and removing intravascular SPEAR precipitates using a commercial software (Imaris, see Methods). **c**, Removal of vasoactive intestinal peptide SPEAR intravascular precipitates after one round of image processing.

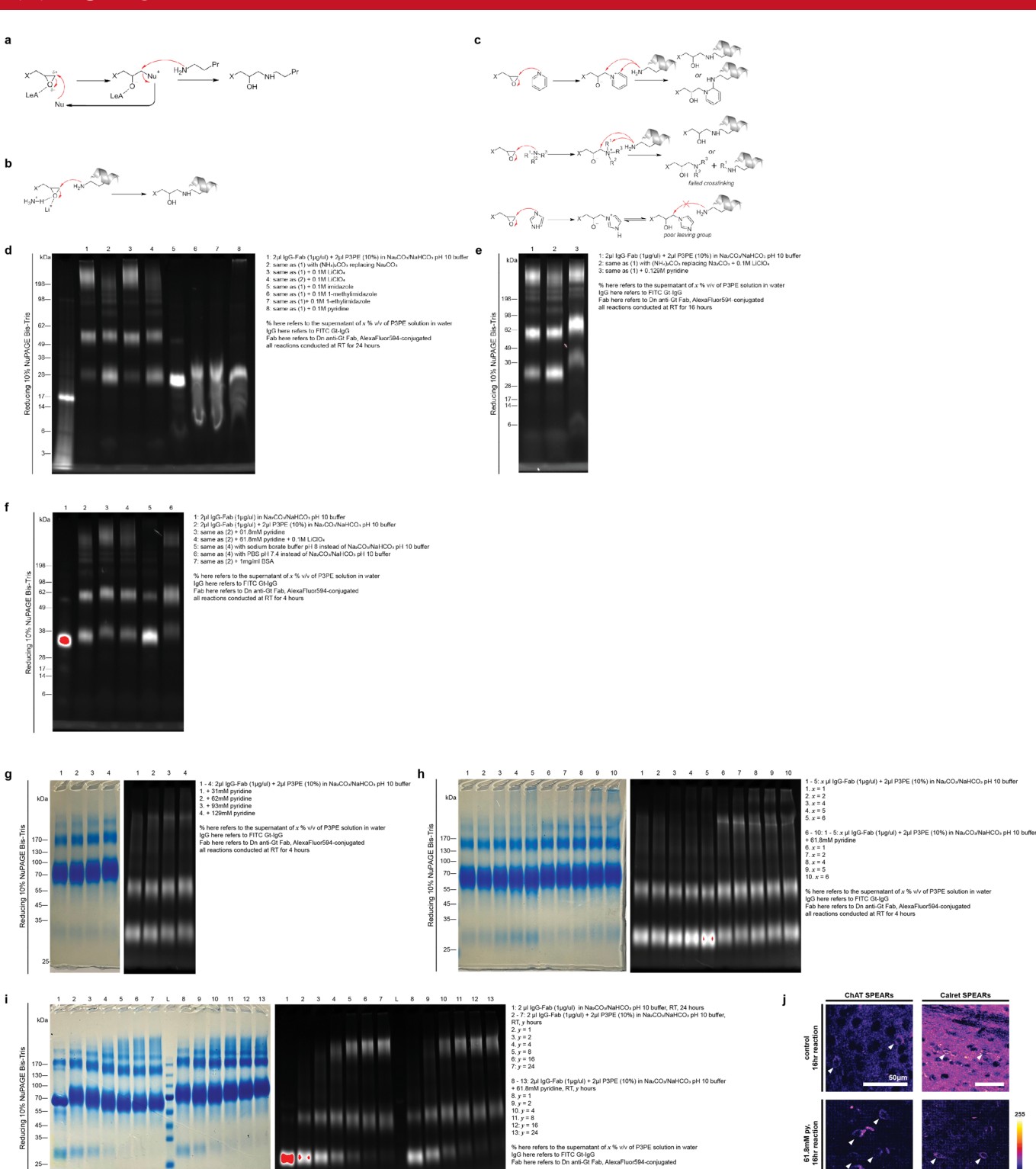

**Extended Data Fig. 5 | See next page for caption.**

**Extended Data Fig. 5 | Development and optimization of a catalyst for P3PE crosslinking of amine-containing proteins. a**, Catalyst conception based on the use of a nucleophile (Nu) that can result in an intermediate with a good leaving group, and/or the use of Lewis acids (LeA) to facilitate the nucleophilic ring opening. **b**, Lewis acids compatible with our reaction condition are lithium and ammonium ions. **c**, Nucleophiles compatible with our reaction condition are those with high nucleophilicities and low basicity in a protic solvent environment, including pyridines, sterically hindered trisubstituted amines, and imidazoles. However, aliphatic amines and imidazoles can lead to failed crosslinking due to side reactions. **d–f**, Reducing SDS–PAGE for screening and confirming catalytic activities. The tested reaction conditions are listed on the right. **d**, Screened catalyst candidates chosen based on rationales described in a–c using reducing SDS–PAGE. **e**, Confirmation of catalytic and non-catalytic effect of pyridine and lithium on SPEARs formation, respectively. **f**, Exploration of pyridine's catalytic effect under various conditions. All experiments were repeated three to six times with similar results. **g–j**, Optimization and characterization of pyridine-catalyzed formation of SPEARs. **g–i**, Reducing SDS–PAGE for optimization and characterization of pyridine-catalyzed P3PE-crosslinking reaction. The tested reaction conditions are listed on the right. **g**, Titration of pyridine concentration in the reaction mixture. **h**, Effect of antibody–Fab complex concentration and P3PE concentration on the overall yield of SPEAR formation. **i**, Time course of pyridine-catalyzed P3PE-crosslinking reaction. All experiments were repeated thrice with similar results. **j**, Comparison of ThICK-staining quality with ChAT SPEARs and calretinin (Calret) SPEARs with and without the use of pyridine. Color scale bar: pixel intensity. Experiments were repeated twice, representative image shown.

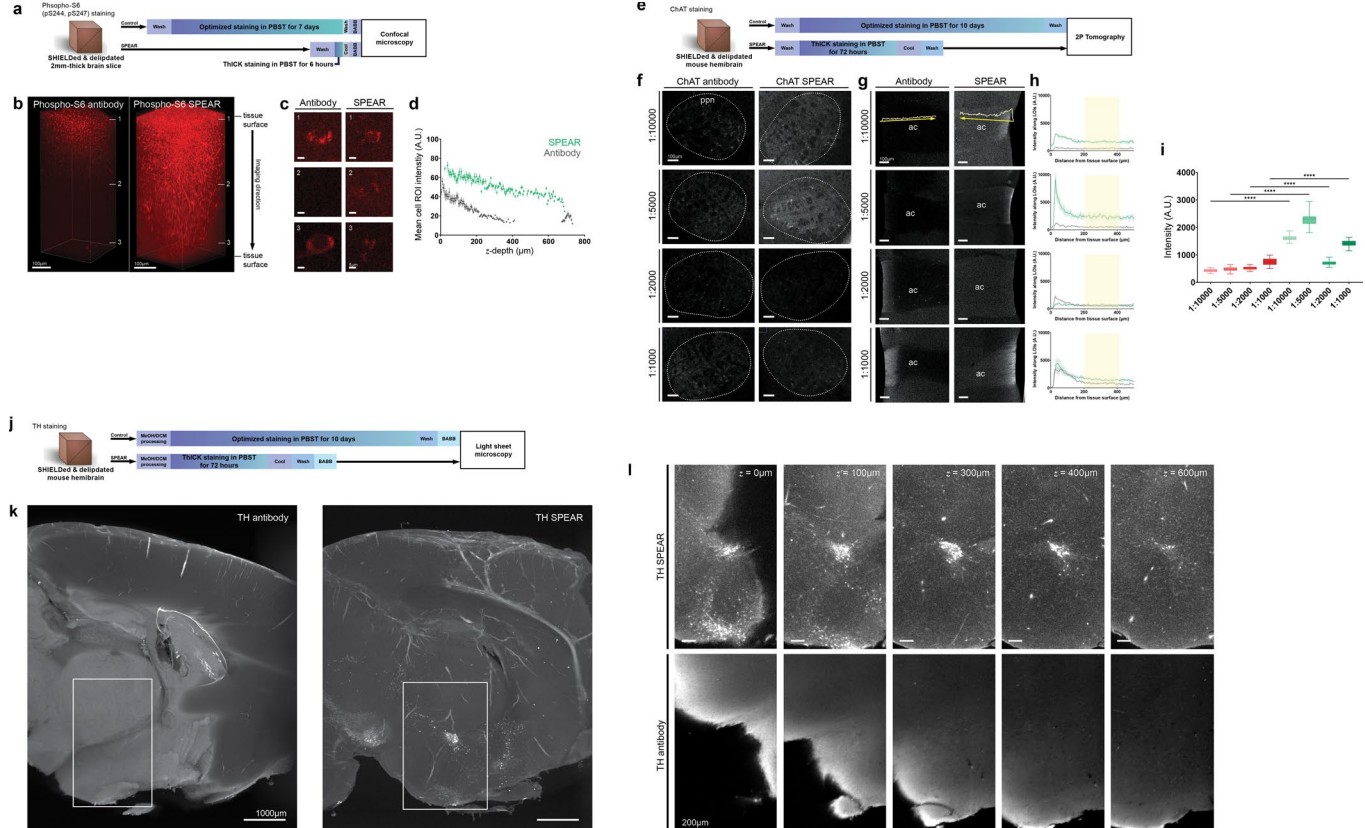

**Extended Data Fig. 6 | Benchmarking optimized conventional antibody staining with or without tissue clearing versus ThICK staining with SPEARs in SHIELD-protected samples. a–d,** 2 mm-thick SHIELD-protected brain slice immunostained with antibodies or SPEARs targeting phospho-S6 ribosomal protein (pSer244, pSer247) (Phospho-S6), a neuronal activity marker. **a,** Schematic of benchmarking experiment. The imaging and tissue processing parameters were kept identical to ensure comparability. **b,** 3D volume rendering of full-thickness images. Note that due to BABB clearing the sample thickness was reduced by ~60%. Experiment performed once. **c,** Excerpted neurons from the 3 indicated levels in **b. d,** Quantified cell intensities across the full thickness of the samples. Data are presented as mean values ± S.D. **e–i,** SHIELD-protected mouse hemibrains immunostained with antibodies or SPEARs targeting choline acetyltransferase (ChAT) at various dilutions. **e,** Schematic of benchmarking experiment. The imaging and tissue processing parameters were kept identical to ensure comparability. **f,** Appearances of the pedunculopontine nuclei (ppn) in various samples, which is located ~300 $\mu$m from the tissue surface in a hemi-sectioned brain with low local ChAT fiber density. Grouped comparative experiment performed once. **g,** Appearances of the anterior commissure (ac) in various samples, which is in contact with the tissue surface. Note that the presence of positive stainings along the tissue surface suggest specific staining of ChAT-positive fibers in all cases but with varying penetration depths. Yellow arrows indicate a line of interest (LOI) along which the direction of signal intensity was repeatedly measured and shown in h. **h,** Averaged signals measured along 10 pixels-wide LOIs and were repeated 5 times per sample. Data are presented as mean values ± S.D. **i,** Signal intensities at the deeper portion (200–400 m $\mu$m) of the visualized anterior commissures, as sampled from the depth range in the yellow boxes in h. Whiskers: minima and maxima, bounds of box: 75th and 25th percentiles, centers: medians. P < 0.0001, two-sided Brown–Forsythe ANOVA test. ****: P < 0.0001, Tamhane's T2 multiple comparisons test with 4 comparisons per family. **j–l,** iDISCO-processed mouse hemibrains immunostained with antibodies or SPEARs targeting tyrosine hydroxylase (TH). **j,** Schematic of benchmarking procedure. The imaging and tissue processing parameters were kept identical to ensure comparability. **k,** 600-$\mu$m-thick maximum intensity projections along the sagittal direction of the stained mouse hemibrains. Somas of the tyrosine hydroxylase (TH)-positive neurons are clearly visible in the TH SPEAR ThICK-stained sample due to homogeneous staining along the z-direction, whereas TH antibodies in the control sample are mostly deposited as bright areas along the mid-sagittal surface of the hemibrain, impeding the visualization of deeper TH-expressing neurons. **l,** Enlarged view in the boxed areas in k, showing TH-positive neurons located in the hypothalamus (group A15). z-depths were sampled starting from the tissue surface. Experiment performed once.

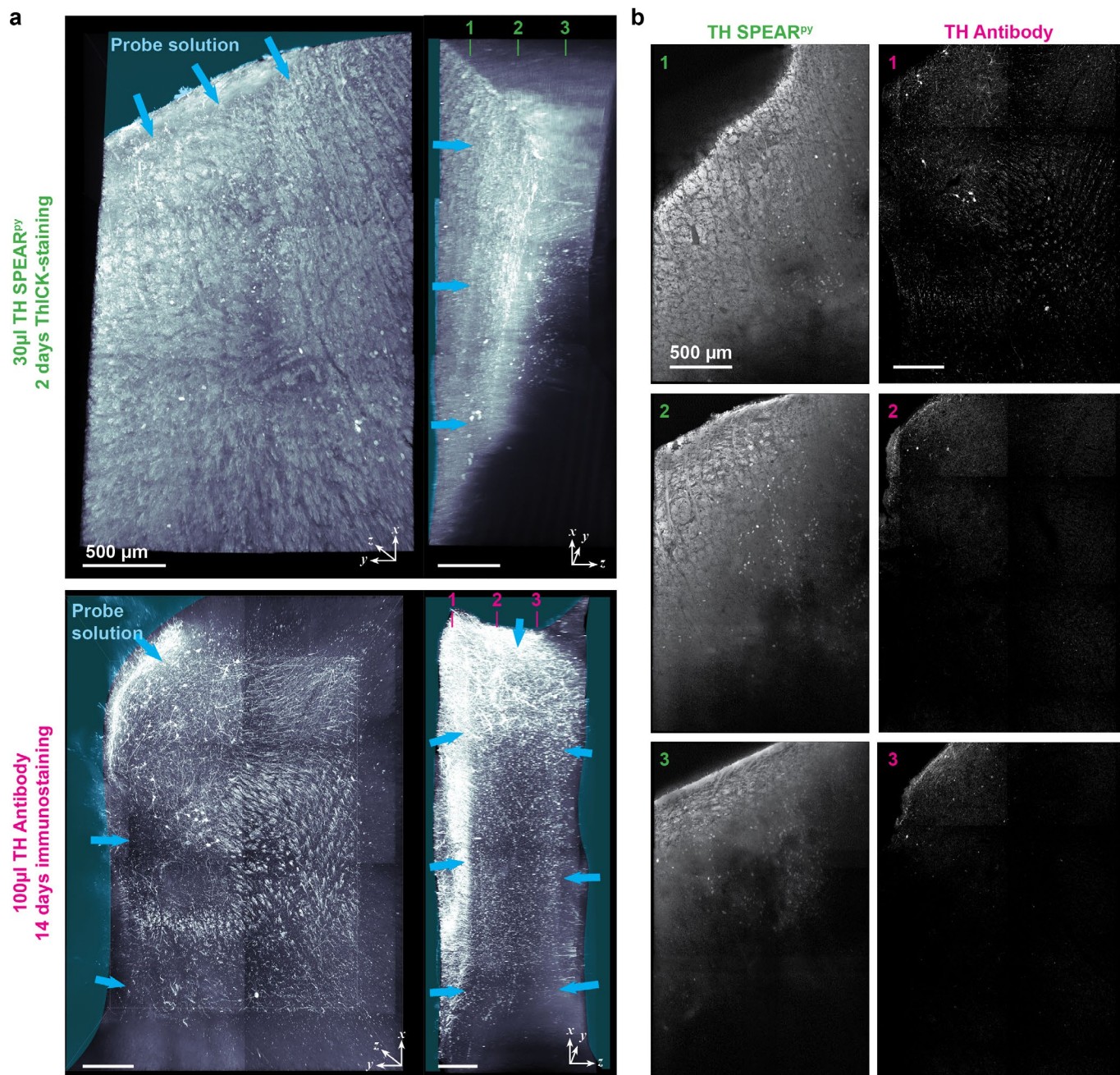

**Extended Data Fig. 7 | Human pons tyrosine hydroxylase (TH) staining with TH SPEAR^py. a**, Rendered views of the ThICK-stained (with TH SPEARpy) and conventional immunostained (with anti-TH antibody) human pons samples. Probe solutions containing TH SPEARpy or anti-TH antibody are outlined in light blue shadows, with arrows depicting the directions of diffusion (into the tissues). Note that the apparent fuzziness over the large imaged volume in the rendered view of the ThICK-stained sample is due to the denser staining of TH-positive fiber near the tissue surface, obscuring the deep-located TH-positive neurons. **b**, The images from three different depths in the ThICK- and conventionally stained samples are shown. Experiment performed once.

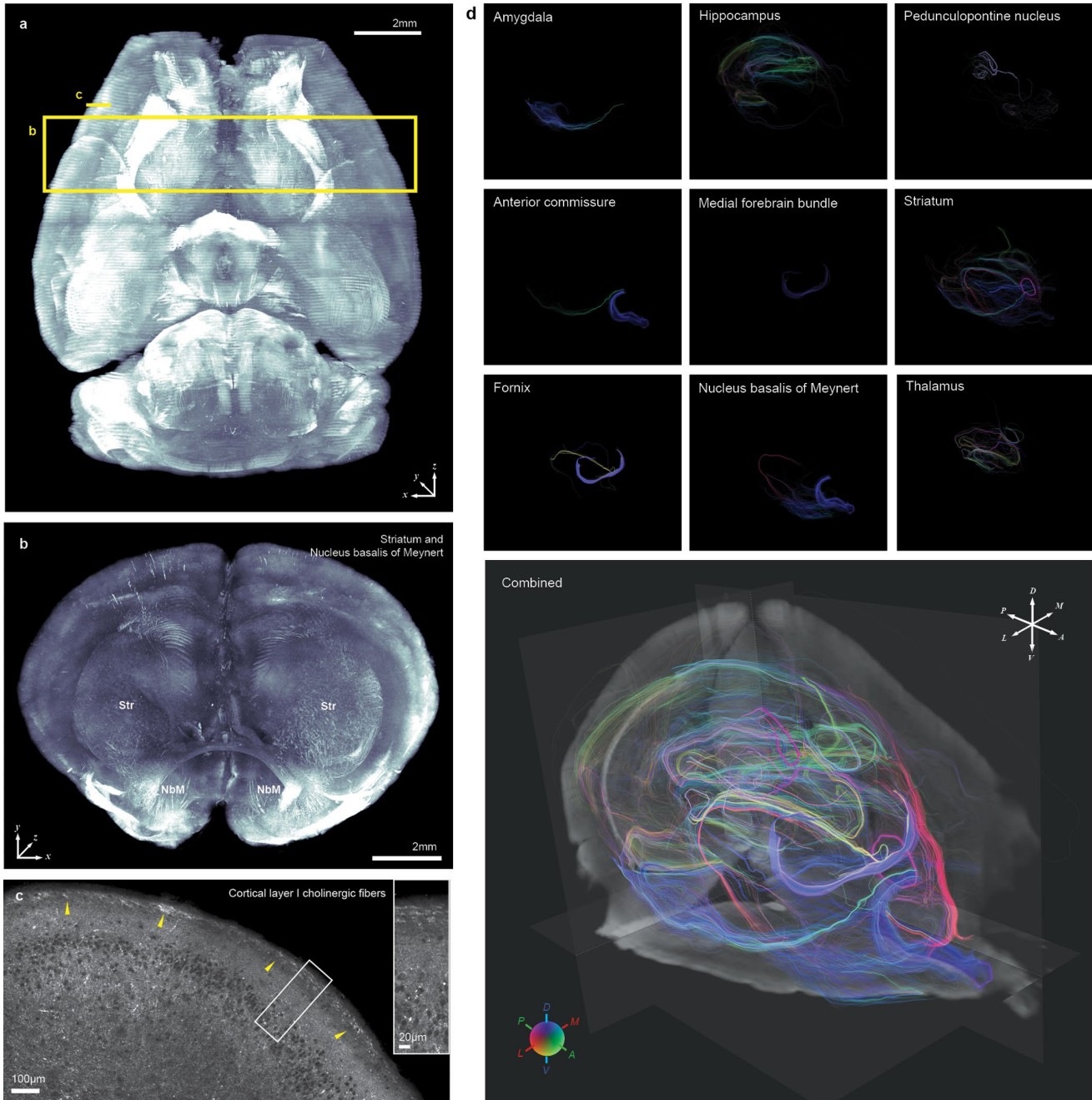

**Extended Data Fig. 8 | Additional excerpted views from a ChAT SPEAR-stained SHIELD-protected whole mouse brain. a**, Enlarged view of the whole mouse brain shown in Fig. 4g. **b**, Volume-rendered coronal plane view of the striatum (Str) and nucleus basalis of Meynert (NbM), demonstrating cholinergic fibers in the anterior commissure, basal forebrain and striatum. **c**, Cholinergic fibers (arrowheads) in layer 1 of inferior frontal cortex. Inset shows an enlarged view. **d**, Cholinergic tractographies grouped according to anatomical landmarks. Selected anatomical regions were manually segmented and ChAT-positive fibers that traverse the segmented region are displayed individually (upper panels) or in combination (lower panel). Fibers are color coded according to their middle segment directionality. Experiment performed once.

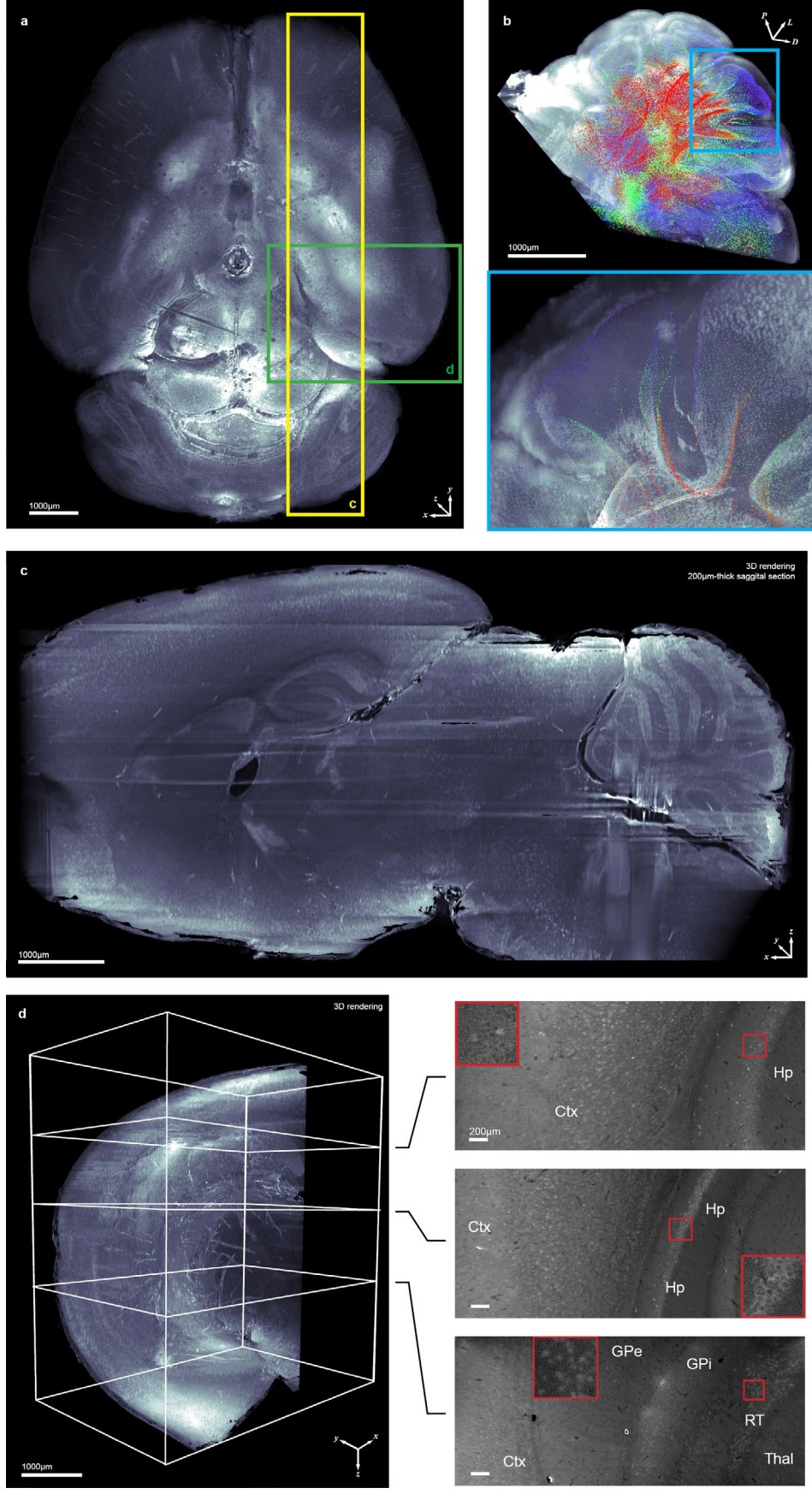

**Extended Data Fig. 9 | See next page for caption.**

**Extended Data Fig. 9 | Additional excerpted views from a PV SPEAR<sup>py</sup>-stained whole mouse brain processed with the iDISCO protocol. a**, Enlarged whole-brain 3D-rendered view. **b**, Another view of the cerebellar hemisphere shown in Fig. 4r, overlaid with identified PV-positive cells. Inset shows a further excerpted view demonstrating accurate localization of spots. **c**, Sagittal slice 3D rendering as indicated by the yellow boxed area in a. Individual PV-positive cells are visible from this volume rendering. **d**, Block 3D rendering as indicated by the green boxed area in a. Enlarged views from different brain regions from different z-slices were excerpted and displayed in the right panels, with further enlarged views for red boxed areas showing cells in insets. Experiment performed once.

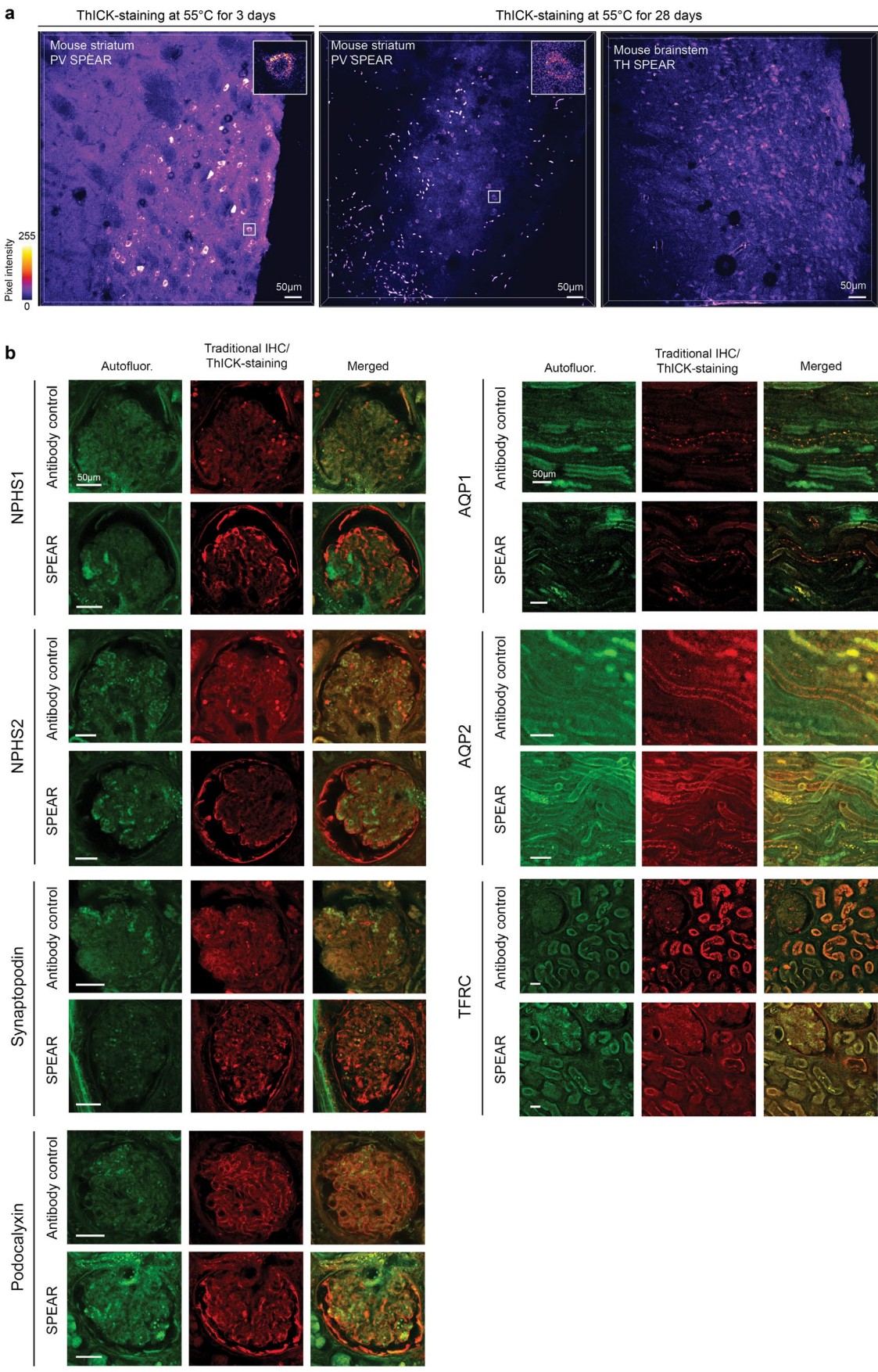

**Extended Data Fig. 10 | See next page for caption.**

**Extended Data Fig. 10 | Scalability and generalizability of SPEARs for ThICK staining. a**, SPEARs targeting parvalbumin (PV) and tyrosine hydroxylase (TH) were prepared and used to stain mouse brain regions as indicated. The right images were obtained with ThICK staining at 55oC for 28 days, demonstrating the heat tolerance of the applied SPEARs over extended periods. The experimental and imaging parameters were kept constant for the PV SPEAR experiment set. Experiment performed once. **b**, ThICK staining with SPEARs is in principle applicable to non-neural tissues. Examples shown are human kidney tissues ThICK-stained for various antigens compared to their traditional antibody counterparts, showing similar staining patterns, and verifying the same antigen-binding specificity of the SPEARs as the original antibody counterparts. Notably, for NPHS1, NPHS2, synaptopodin and TFRC, ThICK staining with SPEARs appeared to provide higher staining quality (that is, achieving staining patterns more closely resembling that expected by comparison to chromogenic immunostaining images from the Human Protein Atlas (https://www.proteinatlas.org/humanproteome/tissue). Experiment performed twice with similar results, representative images shown.

# nature research

# Reporting Summary

Nature Research wishes to improve the reproducibility of the work that we publish. This form provides structure for consistency and transparency in reporting. For further information on Nature Research policies, see our Editorial Policies and the Editorial Policy Checklist.

## Statistics

For all statistical analyses, confirm that the following items are present in the figure legend, table legend, main text, or Methods section.

| n/a | Confirmed | |
|---|---|---|
| ☐ | ☒ | The exact sample size (*n*) for each experimental group/condition, given as a discrete number and unit of measurement |
| ☐ | ☒ | A statement on whether measurements were taken from distinct samples or whether the same sample was measured repeatedly |
| ☐ | ☒ | The statistical test(s) used AND whether they are one- or two-sided *Only common tests should be described solely by name; describe more complex techniques in the Methods section.* |
| ☐ | ☒ | A description of all covariates tested |
| ☐ | ☒ | A description of any assumptions or corrections, such as tests of normality and adjustment for multiple comparisons |
| ☐ | ☒ | A full description of the statistical parameters including central tendency (e.g. means) or other basic estimates (e.g. regression coefficient) AND variation (e.g. standard deviation) or associated estimates of uncertainty (e.g. confidence intervals) |
| ☐ | ☒ | For null hypothesis testing, the test statistic (e.g. *F*, *t*, *r*) with confidence intervals, effect sizes, degrees of freedom and *P* value noted *Give P values as exact values whenever suitable.* |
| ☒ | ☐ | For Bayesian analysis, information on the choice of priors and Markov chain Monte Carlo settings |
| ☒ | ☐ | For hierarchical and complex designs, identification of the appropriate level for tests and full reporting of outcomes |
| ☒ | ☐ | Estimates of effect sizes (e.g. Cohen's *d*, Pearson's *r*), indicating how they were calculated |

*Our web collection on statistics for biologists contains articles on many of the points above.*

## Software and code

Policy information about availability of computer code

| Data collection | The selective plane illumination microscopy was performed using the mesoSPIM-control (https://github.com/mesoSPIM/mesoSPIM-control) software (v0.1.5). The two-photon serial tomography was performed using the ScanImage software (v5.6.1). |
|---|---|
| Data analysis | Image processing and analyses were performed using Imaris (v9), Zen Blue (v3), Fiji (ImageJ) and custom-written codes in MATLAB (R2018b, R2020b & R2021a). The tractography analysis was performed using the Diffusion Toolkit (http://trackvis.org/dtk/) and the TrackVis (http://trackvis.org/) softwares. |

For manuscripts utilizing custom algorithms or software that are central to the research but not yet described in published literature, software must be made available to editors and reviewers. We strongly encourage code deposition in a community repository (e.g. GitHub). See the Nature Research guidelines for submitting code & software for further information.

## Data

Policy information about availability of data

All manuscripts must include a data availability statement. This statement should provide the following information, where applicable:
- Accession codes, unique identifiers, or web links for publicly available datasets
- A list of figures that have associated raw data
- A description of any restrictions on data availability

The numeric data for applicable plots are available in the Source Data file. The raw imaging data presented in this paper are too large for public deposit and will be made available upon reasonable request to the corresponding authors (H.M.L. or H.K.).

# Field-specific reporting

Please select the one below that is the best fit for your research. If you are not sure, read the appropriate sections before making your selection.

☒ Life sciences    ☐ Behavioural & social sciences    ☐ Ecological, evolutionary & environmental sciences

For a reference copy of the document with all sections, see nature.com/documents/nr-reporting-summary-flat.pdf

# Life sciences study design

All studies must disclose on these points even when the disclosure is negative.

| | |
|---|---|
| Sample size | No statistical method was used to predetermine sample size. We decided on the sample sizes and number of experiments based on usual requirements for the specific types of comparisons. These were sufficient as evident from the respective plots in the figures and the associated statistics. |
| Data exclusions | No data were excluded from the analyses. |
| Replication | For the appropriate experiments, all replications are detaily outlined at the respective places in the manuscript. The reproducibility of experiments are evident from the respective plots showing summary statistics and error bars. |
| Randomization | The experiments were not randomized. Wherever applicable, we ensured identical tissue type, source, processing, and/or imaging conditions to ensure comparability. |
| Blinding | The Investigators were not blinded to allocation during experiments and outcome assessment. Wherever applicable, we ensured identical tissue type, source, processing, and/or imaging conditions to ensure comparability. |

# Reporting for specific materials, systems and methods

We require information from authors about some types of materials, experimental systems and methods used in many studies. Here, indicate whether each material, system or method listed is relevant to your study. If you are not sure if a list item applies to your research, read the appropriate section before selecting a response.

## Materials & experimental systems

| n/a | Involved in the study |
|---|---|
| ☐ | ☒ Antibodies |
| ☒ | ☐ Eukaryotic cell lines |
| ☒ | ☐ Palaeontology and archaeology |
| ☐ | ☒ Animals and other organisms |
| ☒ | ☐ Human research participants |
| ☒ | ☐ Clinical data |
| ☒ | ☐ Dual use research of concern |

## Methods

| n/a | Involved in the study |
|---|---|
| ☒ | ☐ ChIP-seq |
| ☒ | ☐ Flow cytometry |
| ☒ | ☐ MRI-based neuroimaging |

# Antibodies

| | |
|---|---|
| Antibodies used | Antibody  (Supplier and cat. no.)<br>Alexa Fluor 594-conjugated donkey anti-goat IgG Fab fragments (Jackson ImmunoResearch 705-547-003)<br>Unconjugated donkey anti-mouse IgG Fab fragments (Jackson ImmunoResearch 715-007-003)<br>Alexa Fluor 488-conjugated donkey anti-mouse IgG Fab fragments (Jackson ImmunoResearch 715-547-003)<br>Alexa Fluor 594-conjugated donkey anti-mouse IgG Fab fragments (Jackson ImmunoResearch 715-587-003)<br>Alexa Fluor 647-conjugated donkey anti-mouse IgG Fab fragments (Jackson ImmunoResearch 715-607-003)<br>Unconjugated donkey anti-rabbit IgG Fab fragments (Jackson ImmunoResearch 711-007-003)<br>Alexa Fluor 488-conjugated donkey anti-rabbit IgG Fab fragments (Jackson ImmunoResearch 711-547-003)<br>Alexa Fluor 594-conjugated donkey anti-rabbit IgG Fab fragments (Jackson ImmunoResearch 711-587-003)<br>Alexa Fluor 647-conjugated donkey anti-rabbit IgG Fab fragments (Jackson ImmunoResearch 711-607-003)<br>Alexa Fluor 488-conjugated goat anti-rat IgG Fab fragments (Jackson ImmunoResearch 112-547-003)<br>AQP1 (ABclonal A4195)<br>AQP2 (ABclonal A16209)<br>Arc (Santa Cruz Biotechnology sc-17839)<br>α-SMA (Progen 690001)<br>CR (Abcam ab702)<br>ChAT (Millipore AB144P)<br>DBH (Sigma HPA002130)<br>DDC (Sigma HPA017742)<br>DLG3 (Sigma HPA001733) |

EGR1 (Santa Cruz Biotechnology sc-515830)
c-Fos (Santa Cruz Biotechnology sc-166940)
Gephyrin (Santa Cruz Biotechnology sc-25311)
GFAP (Santa Cruz Biotechnology sc-58766)
GFAP (Invitrogen 13-0030)
IBA1 (Wako 019-19741)
MAP2 (Abcam ab11267)
Na+/K+-ATPase (ABclonal A11683)
NeuN (Abcam ab104224)
NPAS4 (Invitrogen PA5-39300)
NPHS1 (ABclonal A3048)
NPHS2 (ABclonal A17337)
NT5E (ABclonal A2029)
OLIG2 (Sigma HPA003254)
PDGFRA (ABclonal A2103)
Phospho-S6 (pSer244, pSer247) (Invitrogen 44-923G)
PODXL (ABclonal A10200)
PSD95 (NeuroMab K28/43)
PV (Abcam ab11427)
PV (Invitrogen PA1-933)
S100b (Enzo LifeSciences ENZ-ABS307-0100)
Synapsin I (Novus Biologicals NB300-104)
SYNPO (ABclonal A8484)
SOM (Millipore MAB354)
TFRC (ABclonal A5865)
TH (Millipore AB152)
TPH2 (Sigma AMAb91108)
VGLUT2 (Sigma AMAb91081)
VIP (Bioss bs-0077R)
VGLUT2 (Sigma AMAb91081)
VIP (Bioss bs-0077R)

Validation | As this is an imaging method development paper, we validated all antibodies without modifications ourselves in tissue. These were based on the expected patterns of immunolabeling revealed based on biological knowledge (all figures / supplementary figures), as well as with additional comparisons to the Human Protein Atlas (Supplementary Fig. 10).

# Animals and other organisms

Policy information about studies involving animals; ARRIVE guidelines recommended for reporting animal research

Laboratory animals | Male C57BL/6 and Thy1-GCaMP6f transgenic adult mice of at least 2 months old were used. The mice were provided by the Laboratory Animal Service Center of CUHK and maintained at controlled temperature (22–23°C) with an alternating 12h light/dark cycle with free access to standard mouse diet and water. The ambient humidity was maintained at <70% relative humidity.

Wild animals | This study did not involve wild animal.

Field-collected samples | This study did not involve field-collected samples.

Ethics oversight | All experimental procedures were approved in advance by the Animal Research Ethical Committee of the Chinese University of Hong Kong and were carried out in accordance with the Guide for the Care and Use of Laboratory Animals.

Note that full information on the approval of the study protocol must also be provided in the manuscript.

