## [Peer Review File · Nature Methods]

Peer Review Information

Manuscript Title: Antibody stabilization for thermally accelerated deep immunostaining

Corresponding author name(s): Hei-Ming Lai, Ho Ko

Reviewer Comments & Decisions:

Decision Letter, initial version:

Dear Dr Ko,

Thank you for submitting your manuscript entitled "Antibody stabilization for thermally accelerated deep immunostaining". We have given the paper our careful consideration but we regret that we cannot publish it in Nature Methods.

It is Nature Methods' policy to decline a substantial proportion of manuscripts without peer-review, so that they may be sent elsewhere without delay. Decisions of this kind are made by the editorial staff when it appears that papers are unlikely to succeed in the competition for limited space.

Among the considerations that arise at this stage are a manuscript's probable interest, level of methodological development and immediate practical relevance to a general readership. We do not doubt the technical quality of your work or that it will be of interest to others working in this area of research. We read your work with interest, however I am sorry that without further benchmarking against existing methods, including a clear demonstration of the advantage of using SPEAR/THICK over SHIELD alone, we do not think that the technical advances presented will have a sufficiently significant and immediate impact on a broader readership to justify publication in Nature Methods.

Nevertheless, thank you very much for giving us the opportunity to consider your manuscript. I am sorry that we cannot be more positive on this occasion and hope that you will promptly find a more appropriate forum for presenting your work.

Sincerely,
Madhura

Madhura Mukhopadhyay, PhD
Associate Editor
Nature Methods

Although we cannot offer to publish your paper in Nature Methods, the work may be appropriate for another journal in the Nature Research portfolio. If you wish to explore suitable journals and transfer your manuscript to a journal of your choice, please use our [manuscript transfer portal](https://mts-nmeth.nature.com/cgi-bin/main.plex?el=A6M2ULf1A1BaRo3X3A9ftdvqF6Pvky1ouhnR1jwzRqgQZ). If you transfer to Nature-branded journals or to the Communications journals, you will not have to re-supply manuscript metadata and files. This link can only be used once and remains active until used.

All Nature Research journals are editorially independent, and the decision to consider your manuscript will be taken by their own editorial staff. For more information, please see our [manuscript transfer FAQ](http://www.nature.com/authors/author_resources/transfer_manuscripts.html?WT.mc_id=EMI_NPG_1511_AUTHORTRANSF&WT.ec_id=AUTHOR) page. Note that any decision to opt in to In Review at the original journal is not sent to the receiving journal on transfer. You can opt in to *In Review* at receiving journals that support this service by choosing to modify your manuscript on transfer. In Review is available for primary research manuscript types only.

** For Nature Research Group general information and news for authors, see <http://npg.nature.com/authors>

Author Rebuttal to initial comments

Dear Madhura,

We thank you and the reviewers for detailed evaluations and constructive criticisms. We are grateful that Reviewer 2 evaluated our work positively, and that Reviewer 3 also expressed that it is an interesting work, despite some concerns raised which we are confident in addressing. We felt that a couple of major concerns Reviewer 1 had may have arisen from our inadequate elaboration in the manuscript, perhaps unclear on our most important motivation and innovation – the synergistic preservation of antibody functions with a dual chemical approach that confers enhanced heat-stability and broadened applicability.

In the previous version of the manuscript, as data was presented only in figures, we definitely agree that videos will help us greatly in clarifying the quality of staining. The raised concerns on benchmarking against the original iDISCO method are also reasonable and addressable. We are confident that these will provide the readers with more information for evaluating ThICK-staining with SPEARs and its advantages – principally achieving deep immunostaining with a more homogenous intensity profile across depth.

Our revision plan is summarized as follows, for your kind consideration:

1. We will revise the proposed model of heat-accelerated strategy (original Fig. 1b) as suggested by Reviewer 2 (Comment 1), to more accurately reflect the relative contributions of antibody-antigen binding kinetics (or equilibrium) change vs. diffusivity increase at a higher temperature in attaining deeper staining. In response to Reviewer 1's concerns (Comments 1 and 4), this will be supplemented by reaction-diffusion modeling (similar to Susaki et al., 2020 but additionally incorporating a temperature change in the process) to (i) substantiate the claims in a more quantitative context, and (ii) better explain the advantages of our approach over immunostaining at a lower temperature.
2. Regarding benchmarking experiments, we have already performed those suggested by the reviewers in Supplementary 8 and 12, which we believe could alleviate some of their concerns with better explanations and provision of videos. We hope after their initial re-evaluation we can more effectively liaise on additional benchmarking experiments necessary for the readers. We propose to carry out quantification of signal homogeneity vs. staining depth (to mm-scale) and assessment of staining specificity and quality by sectioning after ThICK-staining, counter-staining with conventional antibody, and imaging as suggested by Reviewer 2 (Comments 3 & 4) and Reviewer 3 (Comment 2), in a head-to-head comparison between ThICK-staining with SPEARs vs. original iDISCO. This should be sufficient for detailed assessment of staining penetrations and quality, as well as bypasses any signal disturbances due to volumetric imaging constraints. As suggested by both Reviewer 1 (Comments 1 & 2) and Reviewer 2 (Comments 3 & 4), we will stain more widely expressed protein markers (e.g., NeuN, MAP2) for this purpose.
3. We plan to perform ThICK-staining for an extended period of time (2 weeks) and compare against a 3-day staining protocol with SPEARs, to test the heat-stability of SPEARs in immunostaining. This shall address Reviewer 2 (Comment 3) and Reviewer 3's (Comment 3) concerns over the stability of SPEARs and the scalability of our method.
4. We will repeat ThICK-staining experiments with GFAP, S100b, OLIG2 and c-Fos SPEARs, to revise data presented in Fig. 2d. This shall address the related concerns of Reviewer 1 (Comment 4), Reviewer 2 (comment 3) and Reviewer 3 (Comment 1) raised on the signal-to-noise ratio of staining. However, we wish to clarify the original images shown in Fig. 2d were by no means optimized immunostaining, as we used very low SPEAR concentrations just to show the compatibility of our approach with the antibodies.
5. Prompted by Reviewer 1 (Comment 5), we will additionally test the applicability of ThICK-staining with SPEARs in mouse kidney tissues, to demonstrate the applications in a wider context.

6. We will attempt gel filtration chromatography experiments on SPEARs, to test whether they form monomers or concatemers – an interesting and important question asked by Reviewer 2 (Comment 2).
7. While there have not been reports of fluorophore heat sensitivity, as Reviewer 1 (Comment 4) asked if a higher temperature may quench the signals of antibody-Fab complexes, we will perform heat stability tests on commonly used fluorophores.
8. We would gladly provide detailed images (e.g., of the TH SPEAR-stained human brain samples) and videos to better illustrate whole- or hemi-mouse brain staining quality (e.g., TH, PV), as Reviewer 1 (Comments 6, 14) suggested and to address Reviewer 2 (Comment 3) and Reviewer 3's concerns (Comment 1).
9. At appropriate places, we will revise the manuscript to ensure better clarity of writing and provide additional information (e.g., the study's motivations, advantages and drawbacks, enlisting all tested compatible/incompatible antibodies with our method, further discussions on the limitations of SPEARs and ThICK-staining).
10. We are also keen to clarify some misunderstandings. For example, we have to emphasize that (i) the Maillard reaction would only occur significantly above 110°C – way above the 55°C heating temperature in ThICK staining (Reviewer 1, Comment 4), (ii) fluorophores can be quenched by light but would generally not suffer from heat instability (Reviewer 1, Comment 4, which we are however willing to additionally show experimentally as per point 7 above), (iii) our image processing workflow for large-volume images mainly relied on custom-written codes, and Imaris was only used for some visualization purposes (Reviewer 1, Comment 9), and (iv) the use of P3PE alone did not lead to sufficient stabilization of antibodies for ThICK staining and hence our design of synergistic protection by Fab-antibody complexation plus P3PE crosslinking (Fig. 1d; Reviewer 1, Comments 3 and 4).

Thank you very much and we look forward to your further advice.

Decision Letter, first revision:

Dear Dr Ko,

Your Brief Communication entitled "Antibody stabilization for thermally accelerated deep immunostaining" has now been seen by 3 reviewers, whose comments are attached. While they find your work of potential interest, they have raised serious concerns which in our view are sufficiently important that they preclude publication of the work in Nature Methods, at least in its present form.

As you will see, the reviewers raise significant concerns about the stability of the SPEARs and an overall lack of benchmarking.

Should further experimental data allow you to fully address these criticisms we would be willing to look at a revised manuscript (unless, of course, something similar has by then been accepted at Nature

Methods or appeared elsewhere). This includes submission or publication of a portion of this work somewhere else. We hope you understand that until we have read the revised paper in its entirety we cannot promise that it will be sent back for peer-review.

If you are interested in revising this manuscript for submission to Nature Methods in the future, please contact me to discuss your appeal before making any revisions or adding experiments.

Otherwise, we hope that you find the reviewers' comments helpful when preparing your paper for submission elsewhere.

Sincerely,
Madhura

Madhura Mukhopadhyay, PhD
Associate Editor
Nature Methods

Reviewers' Comments:

Reviewer #1:

Remarks to the Author:

The authors developed an approach for stabilizing antibodies against thermal and chemical denaturation during deep tissue staining. They highlighted that the P3PE-stabilized-antibodies could withstand 72 hrs of continuous heating at 55 °C. It is unclear what is the advantage of this new method over prior methods if there are any. To judge the quality of deep-tissue labeling, authors would need to generate 3D representative videos with orthoslicing that readers can dive into whole tissues and see the staining quality end-to-end. Comparative quantifications to prior methods are also required. Most of the authors' demonstrations and quantifications are done up to a few hundred micrometers, while whole mouse brain depth could be 5-9 mm. The manuscript should clarify the following issues.

1. As shown in Figure 1m, authors use the primary anti-GFP Ab-Fab fragment complex without crosslinking as negative control and crosslinking after complex formation as a positive control, which explains the crosslinker is beneficial on the immunostaining. However, one paper from Nature Communication already published different opinion using similar antibody-Fab fragment complex [1], which they use primary antibody-Fab fragment complex at 37 °C to show the immunostaining of the brain and even the whole body. This and some other methods show that a lower temperature can already stain the entire mouse brain. Why the authors improve the temperature to 55°C to show the brain staining? If it is faster, better (yields contrast and specific staining throughout the tissue), and/or more versatile (compatible with more diverse antibodies compared to prior methods), this has to be comparatively shown over other whole mouse brain staining methods.

2. A convincing demonstration would be the staining of whole mouse brain with a prominent antibody, e.g., the microglia using Iba1+ antibody (which clearly stains microglia and their processes) compared to other methods such as iDISCO, SHIELD and CUBIC.
3. The linker P3PE is not the first time reported for protein stabilizers in the tissue clearing area [2]. This reference systematically evaluated different denaturation conditions and showed even higher heat exposure up to 70°C tissue reacted with P3PE retained GFP signal in brain slices.
4. Authors stabilize the antibodies for continuous heating. However, the correlation between the elevated temperature and antibody penetration is not fully clear. Firstly, what is the difference between 55 and 37 C used in other methods? Secondly, the higher temperature will lead to the Maillard reaction, which has a terrible impact on the transparency of labeling tissue samples. Thirdly, higher temperatures might quench the fluorescent signal of the antibody-Fab complex. How to balance the antibody penetration and antibody fluorescence quenching during the temperature increase? For example, Fig. 1d shows relatively poor GFAP labeling, which typically works very well in tissue slices with normal histology. After Spears, the quality of the staining seems to go down.
5. The authors showed a limited application of their method in the mouse brain, mouse spinal cord, and five mm-thick human brain slices. How about immunostaining on other mouse organs (liver, spleen, lung, and muscle)?
6. To truly assess the utility of method in large tissue labeling, unbiased whole organs 3D labeling results should be shown. In Figure 2d, a minimal view is shown. In Figure 3, the part of whole mouse brain in detail shown. It is impossible to see details (and assess if the signal is real) from just a whole 3D brain projection. Therefore, the authors should show the whole brain staining using ortho-slice view end-to-end.
7. In Figures 2c and 2d, the authors observed Ab-Fab complex precipitation in vessels and explained that the vessels are low-resistance diffusion channels. One concern is that this phenomenon may be attributed to the non-specific binding of the P3PE linker. And in Figure S12 b, it is tough to say TH stained the neurons, and it seems like TH also stained the vessels.
8. In Figure 3p-q and Figure S11, many unspecific binding staining are observed. Many dots are similar to autofluorescence. Can authors quantify how the autofluorescence is affected with this new protocol?
9. Please don't use Imaris to process the imaging. Filter-based methods are notoriously poor-performers when it comes to large tissue analysis.
10. How many animals are used in this study and how much repetition for each antibody staining experiment?
11. Please also report which antibodies being tested and did not work that other researchers could save resources and time.

Minor comments:

12. In Figures 1a and 1b, the authors try to use a model to explain antibody diffusion to reach a deep tissue antigen target. However, this model might be unrealistic because the natural combination of the antigen with antibody is influenced by many parameters, such as the chemical steric hindrance and specific binding sites. Before the antibody meets the antigen, the antibody should also overcome the barrier of endothelial cells/intercellular substances to enter into deeper tissue.

13. In Figure 1c, triepoxide P3PE combines with protein not only by the amino group but also by the sulfhydryl group.

14. I love the colored images in Figure 3l and Sup. Fig 10. However, I don't see the original data fitting. Can authors show similar axonal tracks in original Th stained brains by orthoslicing end-to-end?

15. Figure 3c-e compared the penetration depth difference to demonstrate the ThICK staining is better. But the conventional immunostaining method as a negative control group is not reliable for deep tissue immunostaining.

[1] Etsuo A. Susaki, et al, Versatile whole-organ/body staining and imaging based on electrolyte-gel properties of biological tissues, Nature Communications, 2020.

[2] Young-Gyun Park, et al, Protection of tissue physicochemical properties using polyfunctional crosslinkers, Nature Biotechnology, 2019.

Reviewer #2:

Remarks to the Author:

This manuscript by Lai and colleagues proposes a new high-throughput 3D staining method in combination with antibody complex stabilization (SPEARs) and staining protocol for thick tissue samples (ThICK). They claimed that crosslinking by an epoxy chemical P3PE stabilizes the antibody complex so that it can resist incubation at a high temperature (55°C). The given thermostability leads to reduced antibody binding to the antigen (decreased K_d) and thus contributes to promoting antibody penetration. The authors showed SPEARs to apply to up to 24 antibodies and the ThICK protocol to stain whole human tissue blocks and whole adult mouse brains.

Considering that 3D tissue staining methods for large tissue samples are still in their infancy, various advanced methods must be proposed and discussed in the research community. Therefore, this reviewer positively evaluated the author's timely idea and efforts to shorten the 3D staining period, a significant bottleneck of the current methods, by applying the recent SHIELD-related crosslinking. On the other hand, this reviewer found several concerns in the current manuscript as follows, which is expected to be revised for finalization.

Major concerns

#1: Diffusion-reaction model (Figure 1a, b)

-Could the authors further clarify the temperature dependence of K_d on T ? In the current manuscript, the thermal denaturation (that can cause the decrease of $[Ab]$ and $[Ab+Ag]$. In contrast, increase $[Ag]$) and exothermic reaction (that means, as long as the reviewer understood, a higher environmental temperature reduced reaction in terms of thermodynamics). However, the reviewer does not think these mechanisms are self-evidence. Is it possible for the authors to add any data (or reference with data) to show the K_d of Ab-Ag reaction as a function of T and its underlying mechanisms? In addition, the change in K_d is related to $[Ab]$ value, so in Figure 1b, the line shape of K_d and the "antibody % active" line should be coordinated.

-The diffusion constant is proportional to the first order of T if the Stokes-Einstein type diffusion ($D \propto T$) is assumed. On the other hand, D is drawn to be proportional to T's second-order or nonlinear function in Figure 1b. What kind of penetration model is assumed here?

-This reviewer thinks that Figure 1b should be re-considered to reflect the real impact of each parameter on diffusion efficiency. For example, assuming D as a function of the first order of T, the change of reaction temperature from 37°C (310.15 K) to 55°C (328.15 K) only gives a 5.8% increase of D value ($328.15/310.15 \approx 1.058$). The impact of Kd change due to T should also be expressed as such an actual value change.

#2: Crosslinking and complex formation (Figure 1, 2)

-The documentation of SPEARs formation efficiency is still confusing. The antibody bands that are not contained in the complex are detected in Figure 1d, while Figure 2i states that they almost completely moved to the complex band (even in the "control" samples). Is it possible to show how much formation efficiency the SPEARs used in the benchmark test in Figure 3 produced?

-Is this complex monomeric or a huge concatenated one? The band is smeared in Figures 1e and 1f, suggesting concatenated complex formation. If that is the case, the diffusion efficiency and signal intensity could be affected by the molecular weight and reduced active sites for antigen binding.

#3: Staining and imaging quality (Figure 3, Supplementary Fig 9,11,12)

-Compared to previously reported mouse whole-brain imaging images such as iDISCO, eFLASH, and CUBIC-HistoVision, the data presented in Figure 3 and related supplementary figures are not comparable. Compared to these previous reports, the whole-brain image of this manuscript has a significantly worse signal-to-noise ratio and shows non-uniform staining (Supplementary Figure 9). If there is a trade-off between speed and data quality, it would not be accepted in many researchers. The shortcomings and limitations of current SPEARs in the comparison of previously reported methods should be accurately discussed.

For example, one of the causes of the low signal-to-noise ratio may be the persistence of thermal stability of SPEARs. As the authors state, "15.9% of crosslinked SPEARs still remained functional after heating at 55 °C for 16 h" (page 3, paragraph 3), SPEAR can be largely denatured and cease to function as an active antibody during the 72-hour staining incubation (Figure 3). These inactive antibody complexes can cause high non-specific background.

-Further investigation should be made on the staining specificity of SPEARs. For example, some of the data shown in Figure 2d may not be specific stained images. This reviewer could not judge that a specific staining signal was obtained, at least for the images of GFAP, S100b, OLIG2, c-Fos. As for the whole brain images, the uniform staining signal of ChAT and TH in the striatum was not obtained (Supplementary Figures 9, 12, also see the results in Susaki et al. 2020). A comparison data with conventional staining should be presented, at least for these cases.

#4: Benchmarking test

-Immunostaining against an antigen uniformly expressed throughout the sample helps show the 3D staining efficiency in a large tissue sample. As demonstrated in Scales [Hama et al. Nat Neurosci 2015] and CUBIC-HistoVision [Susaki et al. Nat Commun 2020], this reviewer suggests that mouse brains should be stained with NeuN antibody for the benchmark of whole mouse brain immunostaining. This reviewer also suggests the comparison of NeuN immunostaining efficiency and quality by using THICK and the original iDISCO (<https://idisco.info/idisco-protocol/>) protocol, not the modified one in the current manuscript, related to Supplementary Fig. 12.

Minor concerns

- #1: Is there a specific reason for using Affipure Fab rather than FabuLight Fab?
- #2: CUBIC-HistoVision -> designated as CUBIC-HistoVIsion in the original paper.

Reviewer #3:

Remarks to the Author:

The authors report a chemical approach for antibody stabilization, termed SPEARs, to allow their usage at higher temperature (up to 55C) for faster diffusion into thick tissues, for up to 72 hours. To achieve this, the authors build upon previously published P3PE approach to crosslink Antibodies with fluorescently labelled Fab fragments (from secondary antibodies), yielding stabilized SPEARs. The author first utilized a variant of ELISA method to systematically optimize various parameters and the reaction stoichiometry, for enhanced stabilization of antibodies. Next, the authors used such SPEAR reagents to perform stainings in human brain tissue (up to 800 microns; TH antibody) and whole mouse brain preparations (ChAT and PV). In addition, the authors report testing of 24 antibodies for compatibility with the SPEAR approach.

Overall, its an interesting study with a lot of experiments to address a very challenging problem in the field - namely of faster and deeper staining of large blocks of tissues. However, there are a few significant concerns, as follows:

1. Generally, the SNR of images shown looks to be relatively poor in most cases (even for highly expressed target proteins). For examples, in Figure 2d many of the images don't look to be good and high SNR lables as would be expected. Figure 3b is not showing raw image quality but instead focusing on "segmented" labels (whereas for the regularly antibody staining raw images are shown). Figure S12b, reporting TH staining, is not convincing. Why there is no staining at all in regular TH antibody, not even in hypothalamus which is pretty near to surface? The signal in the SPEAR stained TH is not impressive. Generally TH antibody labels tons of projections also. PV stainings in Figure 3 appears to be quite bad, generally PV is highly expressed and gives fantastic signals. Also, surprised by a lack of videos (unless I missed it), to better demonstrate the staining qualities, specially with depth. Overall, low SNR images,

and reported existence of intravascular precipitates, might suggest that the SPEARs have significantly altered and/or reduced affinity.

2. The demonstration of whole mouse brain staining is poor to fully assess the performance of this approach. Ideally, the author should take a physical cross-section of the stained brain, counter-stain (with different fluorophore) the section with regular antibody, and image it to assess to uniformity and quality of signal. The images shown are not impressive.

3. If the SPEARs are only stable for 72 hours, then that does put hard limits on their utility for deeper staining in human brain, beyond 800 microns. How is this approach "scalable" then?

Overall, its an extensive study, however it remains unclear how usable this approach is generally. Maybe for a few highly expressed targets.

Although we cannot publish your paper, it may be appropriate for another journal in the Nature Portfolio. If you wish to explore the journals and transfer your manuscript please use our <https://mts-nmeth.nature.com/cgi-bin/main.plex?el=A3M6ULf7C5BaRo6X5A9ftdvqF6Pvky1ouhnR1jwzRqgQZ> manuscript transfer portal. If you transfer to Nature journals or the Communications journals, you will not have to re-supply manuscript metadata and files. This link can only be used once and remains active until used.

All Nature Portfolio journals are editorially independent, and the decision on your manuscript will be taken by their editors. For more information, please see our http://www.nature.com/authors/author_resources/transfer_manuscripts.html?WT.mc_id=EMI_NPG_1511_AUTHORTRANSF&WT.ec_id=AUTHOR manuscript transfer FAQ page.

Note that any decision to opt in to In Review at the original journal is not sent to the receiving journal on transfer. You can opt in to *[In Review](https://www.nature.com/nature-research/for-authors/in-review)* at receiving journals that support this service by choosing to modify your manuscript on transfer. In Review is available for primary research manuscript types only.

** For Nature Research Group general information and news for authors, see <http://npg.nature.com/authors>.

Author Rebuttal, First revision:

We thank the reviewers for their constructive suggestions and criticisms that helped us significantly improve the paper. We have included more data to address the concerns raised, and made changes to

parts of the manuscript where we or the reviewers felt more clarity or detail was required. Major additions and changes include:

- We revised the proposed model of heat-accelerated strategy to more accurately reflect the relative contributions of antibody-antigen binding equilibrium changes vs. diffusivity increase at a higher temperature in attaining deeper staining (Fig. 1b). This is also supplemented by reaction-diffusion modeling (Fig. 1c, d) (similar to Susaki et al., 2020 but additionally incorporating a temperature change in the process) to (i) substantiate the claims in a more quantitative context, and (ii) better explain the advantages of our approach over immunostaining at a lower temperature (e.g., 37 °C). The results match what we observed experimentally (Supplementary Fig. 9).
- For benchmarking experiments, apart from the original data presented in Supplementary Fig. 9 & 13, we now provide results from additional experiments comparing ThICK vs. iDISCO and CUBIC-HV (presented in new Fig. 4). These were carried out in head-to-head comparisons between ThICK-staining with SPEARs vs. iDISCO and CUBIC with stringent experimental designs (please see Methods), using two widely expressed markers in the brain (PV and MAP2), as well as the quantification of signals in relation to the penetration distance from the nearest tissue surface.
- To show the heat stability of SPEARs in immunostaining, we performed ThICK-staining (with PV SPEARs) for an extended period of time (4 weeks) (new Supplementary Fig. 14). This shows that if necessary (e.g., for staining centimeter- or decimeter-scale samples), SPEARs possess sufficient stability for prolonged large scale ThICK-staining.
- We repeated ThICK-staining experiments with selected SPEARs (including GFAP, OLIG2 and c-Fos SPEARs), and the revised data are now presented in Fig. 2d. We also revised the main text and Supplementary Table 1, to clearly enlist antibodies that can be used to make functional SPEARs (i.e., compatible with ThICK-staining), as well as those that failed to work.
- While there have not been reports of fluorophore heat sensitivity, to address the inquiry that whether a higher temperature may quench the signals of fluorophores on antibody-Fab complexes, we demonstrated the heat stability of several commonly used fluorophores (Rebuttal Fig. 1).
- We provide more detailed images for the TH SPEAR-stained human brain sample (Supplementary Fig. 10), as well as videos for the ChAT, PV and TH SPEARs-stained mouse brain samples (Supplementary Videos 1–3), to better illustrate the staining quality.
- We performed ThICK-staining in mouse kidney tissues (new Supplementary Fig. 15), to demonstrate that our method is in principle applicable to other organs.
- We have revised the manuscript to improve the clarity of writing and provide additional necessary information. These include better explanations for the study's motivations, further discussions on the limitations of SPEARs and ThICK-staining, and clarifications at suitable places to avoid misunderstandings.

Please see below the detailed point-by-point responses to address each of the concerns.

Reviewer #1:

Remarks to the Author:

The authors developed an approach for stabilizing antibodies against thermal and chemical denaturation during deep tissue staining. They highlighted that the P3PE-stabilized-antibodies could withstand 72 hrs of continuous heating at 55 °C. It is unclear what is the advantage of this new method

over prior methods if there are any. To judge the quality of deep-tissue labeling, authors would need to generate 3D representative videos with orthoslicing that readers can dive into whole tissues and see the staining quality end-to-end. Comparative quantifications to prior methods are also required. Most of the authors' demonstrations and quantifications are done up to a few hundred micrometers, while whole mouse brain depth could be 5-9 mm. The manuscript should clarify the following issues.

General response:

We thank the reviewer for the constructive criticisms and comments. Our manuscript intended to focus not only on the novelty of thermostabilizing antibodies using a synergistic combination of complexation with molecular chaperone and multifunctional crosslinker, we also proposed a novel concept in 3D immunostaining where the intentional application of high temperatures in immunostaining – previously deemed impossible – can provide a simple mean in overcoming reaction-diffusion obstacles in deep immunostaining. This is supported by more sophisticated discussions on the putative mechanisms (page 2, paragraph 3 & page 3, paragraph 1), as well as quantitative modeling (Fig. 1c, d). The result is a fast (within days, Fig. 4c) and scalable 3D immunostaining method without the use of sophisticated equipment. We apologize if our writings were not sufficiently clear that led to misunderstandings, and are very grateful the reviewer's comments prompted us to clarify our technique's novelty.

In response to the other comments of the reviewer, we have (i) generated videos with end-to-end slicing to demonstrate the staining quality more clearly (Supplementary Videos 1–3), (ii) performed new experiments to benchmark ThICK-staining against iDISCO and CUBIC-HV, with quantifications of staining intensity and homogeneity up to ~1 mm distance from tissue surface (Fig. 4). We also wish to clarify that our quantification of penetration distance is different from previous studies – rather than approximating the penetration using the z-depth of imaging, we defined it as the distance of a given voxel from the nearest tissue surface, inclusive of the ventricles or other non-vascular spaces. We believe that while our method of quantification may give smaller estimates of penetration distance compared to z-depth-based measurements, it is more accurate in reflecting the true penetration depths.

1. & 2. As shown in Figure 1m, authors use the primary anti-GFP Ab-Fab fragment complex without crosslinking as negative control and crosslinking after complex

formation as a positive control, which explains the crosslinker is beneficial on the immunostaining. However, one paper from Nature Communication already published different opinion using similar antibody-Fab fragment complex [1], which they use primary antibody-Fab fragment complex at 37 °C to show the immunostaining of the brain and even the whole body. This and some other methods show that a lower temperature can already stain the entire mouse brain. Why the authors improve the temperature to 55°C to show the brain staining? If it is faster, better (yields contrast and specific staining throughout the tissue), and/or more versatile (compatible with more diverse antibodies compared to prior methods), this has to be comparatively shown over other whole mouse brain staining methods. A convincing demonstration would be the staining of whole mouse brain with a prominent antibody, e.g., the microglia using Iba1+ antibody (which clearly stains microglia and their processes) compared to other methods such as iDISCO, SHIELD and CUBIC.

Response:

Thanks for these helpful comments, that prompted us to better explain several points on the main novelty and advantages of our method, supported by new benchmarking experimental results, in the revised manuscript:

- We chose Fab fragment for its monovalency on complexing antibodies, which can synergistically stabilize antibodies when coupled with the crosslinker P3PE. The synergism in stabilization was shown by comparing the heat stability provided by antibody-Fab complexation without or with P3PE-crosslinking (Fig. 1n). Such strategy was not obvious nor explored in previous studies, and represents a key novelty in the development of SPEARs – the prerequisite for immunostaining at high temperatures.
- The use of a higher temperature (i.e., 55°C) in immunostaining stemmed from the theoretical consideration that antibody-antigen binding is the key obstacle impeding antibody penetration into the tissue, which can be hindered by raising the temperature but is only viable with the heat stability of SPEARs. Such a theoretical basis has been described (Fig. 1a, b), now more quantitatively demonstrated via reaction-diffusion modeling (Fig. 1c, d) and discussed in detail in the main text (page 2, paragraph 3 & page 3, paragraph 1). The related experimental results on the key role of reaction barrier has also been demonstrated (Supplementary Fig. 9), where simply adding more antibodies and prolonging the incubation (as illustrated in the associated timeline) would not achieve staining at distances far from the tissue surface.
- We agree that the comparisons could have been illustrated more clearly. We now added a set of stringent benchmarking experiments (presented in new Fig. 4), which shows that despite the much shorter incubation period (3 days), our method achieved similar or better results than iDISCO or CUBIC-HV (which require longer incubation times (weeks)), by staining against two widely expressed markers (e.g. MAP2 and PV). Unfortunately we failed to synthesize functional SPEARs for immunostaining for the tested IBA1 and NeuN antibodies, which has now been documented (Supplementary Table 1).

3. The linker P3PE is not the first time reported for protein stabilizers in the tissue clearing area [2]. This reference systematically evaluated different denaturation conditions and showed even higher heat exposure up to 70°C tissue reacted with P3PE retained GFP signal in brain slices.

Response:

As discussed above, we wish to clarify that another novel strategy in our study is the choice of chemically engineering antibodies themselves – rather than the buffers or tissues – which has never been considered in the tissue clearing area. We acknowledged our choice of P3PE for antibody stabilization was based on good data by the suggested work by Park et al., 2018 (please see page 3, paragraph 1), though we also note that this work applied P3PE to tissues and was not related to deep immunostaining techniques. In addition, our results presented in Fig. 1n demonstrated that P3PE alone cannot lead to reliable formation of heat-resistant SPEARs.

4. Authors stabilize the antibodies for continuous heating. However, the correlation between the elevated temperature and antibody penetration is not fully clear.

Firstly, what is the difference between 55 and 37 C used in other methods?

Response:

At 37°C, most antibodies will bind to their antigens as soon as they encounter one upon diffusing into the tissue. Since antigens have been fixed in position within the tissue, the slow dissociation of antibodies from their antigens effectively locks the antibodies among the superficial antigens (i.e., this creates an antibody-antigen binding reaction barrier to deep probe penetration). At 55°C, most native antibodies will progressively become partially or permanently denatured, therefore unable to bind to the antigens and free to diffuse deep into the tissue. In other words, a higher temperature biases the equilibrium of antibody-antigen binding towards their dissociation, thus increasing the proportion of mobile antibodies. However, under such a scenario, since the antibodies become irreversibly denatured, they would cease to be functional.

To avoid permanent heat denaturation, we thus devised the following strategy: (i) antibodies are first chemically modified to render them heat-stable with preserved antigen-binding capability (i.e., become SPEARs), (ii) during the incubation phase of immunostaining, antibody-antigen binding is minimized by heating to 55°C; as SPEARs remain functional and mostly unbound to the tissue antigens, they are free to diffuse deeper into tissue, and (iii) temperature is then lowered, to favor SPEARs binding to antigens and become “locked” to the deeply located antigens. To provide a more comprehensive quantitative theoretical framework for the strategy, in the revised manuscript we have provided (1) more formal discussions on the above-stated design concept (page 2, paragraph 3 & page 3, paragraph 1), which are supplemented by (2) dimensionless reaction-diffusion process simulation (similar to that by Susaki et al., 2020), which demonstrated that overcoming the antibody-antigen binding reaction barrier is the main contributor to heat-facilitated deep probe penetration (see Fig. 1c, d, and Methods for details).

On a separate note, in our initial stages of method development, we did try 65°C, 75°C, 85°C and 95°C incubation, which could in theory inhibit antibody-antigen binding more strongly, but these temperatures were too harsh even for SPEARs and thus we chose the optimally balanced 55°C.

Secondly, the higher temperature will lead to the Maillard reaction, which has a terrible impact on the transparency of labeling tissue samples.

Response:

We wish to clarify that we are not concerned about the Maillard reaction as it only occurs at a significant rate at temperatures > 100°C. Based on our experience, we never observed any tissue opacification after refractive index homogenization using various clearing agents even with prolonged heating at 55°C.

Thirdly, higher temperatures might quench the fluorescent signal of the antibody-Fab complex. How to balance the antibody penetration and antibody fluorescence quenching during the temperature increase?

Response:

Thanks for bringing up this issue. We believe that the Alex Fluor fluorophores do not suffer from heat instability. To verify this, we tested various Alex Fluor dye and showed that there is no detectable fluorescence quenching after heating at 55°C (Rebuttal Fig. 1).

Rebuttal Fig. 1. Effect of heating fluorophores in PBST at 55°C for 24 hours. Left: digital photograph taken under UV light illumination for various Alexa Fluor dyes without (upper row, control) or with (lower row, heated) heating. Right: quantified fluorescence intensity of the two groups, the experiment was repeated 3 times and the means are plotted. Error bars: S.D.

For example, Fig. 1d shows relatively poor GFAP labeling, which typically works very well in tissue slices with normal histology. After Spears, the quality of the staining seems to go down.

Response:

We wish to clarify that with the images in the original Fig. 1d (now Fig. 1n), we meant to show the synergistic effect of stabilizing antibodies against heat denaturation in immunostaining. In Fig. 1n, as illustrated in the upper panel schematics of the middle column, when the antibody was crosslinked by P3PE followed by complexation with P3PE-crosslinked Fab fragments, we found the immunostaining prone to degradation after

heating at 55°C for 1 hour in the presence of SDS (from the top image to the bottom image, the same tissue was re-imaged). However, in the right column, when the antibody was first complexed by Fab fragments, and the whole complex was subsequently crosslinked, the immunostaining signal can now withstand heating in the presence of denaturants. This demonstrates the synergism in using a molecular chaperone (Fab fragment) in stabilizing antibodies, along with P3PE. Note that the staining conditions and heating conditions (1 hour heating with SDS) were different from that employed in the subsequent ThICK-staining ones (> 16 hours heating without SDS but with Triton X-100). In other words, staining conditions had not been optimized and in Fig. 1n we meant to show the synergistic protection effect only. Nonetheless, staining with the GFAP SPEARs revealed typical astrocyte morphologies (Fig 1n, inset, also see Fig. 2d, sixth row for an image from another sample stained with GFAP SPEAR made from a different anti-GFAP antibody) with good signal-to-background contrast.

5. The authors showed a limited application of their method in the mouse brain, mouse spinal cord, and five mm-thick human brain slices. How about immunostaining on other mouse organs (liver, spleen, lung, and muscle)?

Response:

Given the Brief Communications manuscript, we limited our study to neural tissues as we are more familiar with this organ. We fully agree with the reviewer that in principle this method is generalizable to the immunostaining of other tissues as the same reaction-diffusion barrier applies, which we have now demonstrated with kidney samples (Supplementary Fig. 15).

6. To truly assess the utility of method in large tissue labeling, unbiased whole organs 3D labeling results should be shown. In Figure 2d, a minimal view is shown. In Figure 3, the part of whole mouse brain in detail shown. It is impossible to see details (and assess if the signal is real) from just a whole 3D brain projection. Therefore, the authors should show the whole brain staining using ortho-slice view end-to-end.

Response:

Thanks for the comments in helping to clarify our contents to the readers. We wish to clarify that data shown in Fig. 2d intended to test whether SPEARs can be synthesised from various antibodies, the staining concentrations and durations have not been optimised, and due to resource constraints it would be prohibitively laborious and impractical to perform large-scale, whole-brain imaging for each of the compatible antibodies (25 in total, see Supplementary Table 1). Nonetheless, we believe this should be sufficient for proving the stability of SPEARs made from these antibodies after 16 hours of heating, akin to that shown in the SWITCH method paper (Murray et al., 2015).

For the data shown in Fig. 3, indeed our projection images were not sufficiently clear for showing the staining quality. As advised by the reviewer, we have now provided videos (Supplementary Videos 1–3) for the whole- and hemi-brain images with end-to-end orthoslicing and enlarged views where appropriate.

7. In Figures 2c and 2d, the authors observed Ab-Fab complex precipitation in vessels and explained that the vessels are low-resistance diffusion channels. One concern is that

this phenomenon may be attributed to the non-specific binding of the P3PE linker. And in Figure S12 b, it is tough to say TH stained the neurons, and it seems like TH also stained the vessels.

Response:

We thank the reviewer for the reminder that P3PE can exacerbate the formation of intravascular precipitates. We have added this point in the main text (page 4, paragraph 3). We also wish to note that we also observed these precipitates across many techniques but they are more prominent in staining conditions that are denaturing (i.e., high temperatures, presence of denaturants). We do not yet have a precise mechanism for this and indeed it would be important for subsequent studies to investigate and improve the techniques.

For the mouse hemibrain THICK-stained with TH SPEARs, we wish to point out that vessel precipitates or nonspecific staining were also observed in the control sample without the use of SPEARs or P3PE (compare the left and right images of Supplementary Fig. 13b), while the staining of TH+ve neurons is highly definite given the typical neuronal morphology revealed with typical regional distributions as expected (see Supplementary Fig. 13b, right panel, and Supplementary Fig. 13c, upper panels, also see Supplementary Video 3).

8. In Figure 3p-q and Figure S11, many unspecific binding staining are observed. Many dots are similar to autofluorescence. Can authors quantify how the autofluorescence is affected with this new protocol?

Response:

Thanks for raising this point. We found the quantification of autofluorescence difficult as it lacks defining signal characteristics. In the revised manuscript, we admit that despite achieving a much faster deep probe penetration compared to other methods (see Fig. 4c, Supplementary Table 2), we did observe more background with ThICK-staining (page 7, paragraph 1): “we noticed the overall background of bulk-staining was higher for ThICK-staining with SPEARs than with other staining techniques, which may explain the weaker signal-to-background ratios for ThICK-staining (Fig. 4e).”. We also acknowledge that (page 7, paragraph 2 & page 8, paragraph 1): “Although other state-of-the-art methods take longer times to achieve whole-organ immunolabeling (see Fig. 4c, Supplementary Table 2), ThICK still requires further optimization given the comparatively lower signal-to-noise ratio of staining.”, to more appropriately acknowledge such trade off of SPEARs/ThICK-staining for a 4- to 20-times faster immunostaining speed.

9. Please don't use Imaris to process the imaging. Filter-based methods are notoriously poor-performers when it comes to large tissue analysis.

Response:

We are uncertain which image processing the reviewer was referring to. If the reviewer was referring to that shown in Supplementary Fig. 6 on the removal of vascular precipitates via image processing, we wish to clarify that our point was to show that since these vascular precipitates have distinctive morphologies, they can be easily distinguished from other specific or nonspecific staining via a semi-automated manner using commercial software, if the buffer adjustment approach (Fig. 2f) was not used by the end user for some reason. For

the other analyses, please note that most of the image processing was carried out with flexible, combined uses of MATLAB (including BakingTray, StitchIt and other custom-written scripts), Fiji (ImageJ), Diffusion Toolkit, TrackVis, and Imaris – depending on whether the specific image processing algorithms or functions required were well-implemented in the respective softwares used or needed custom coding.

10. How many animals are used in this study and how much repetition for each antibody staining experiment?

Response:

For each antibody, we repeated the staining at least 3 times until we declared failure of staining, some antibodies (e.g. anti-IBA1 antibody) had gone through more trials due to attempts in buffer exchange, adjustment of staining buffer conditions etc. 12 animals were used in the study to provide tissues for the experimental results presented in Fig. 1 & 2. We now provide this information in the revised manuscript (see page 4, paragraph 2 and page 31, paragraph 2), whereas the number of brain samples used for whole-brain immunostaining experiments are self-evident.

11. Please also report which antibodies being tested and did not work that other researchers could save resources and time.

Response:

Thanks for the important advice, we have now added the information, enlisting all 25 tested antibodies from which we obtained functional SPEARs, as well as 4 that we found unsuccessful for SPEAR synthesis in our hands (Supplementary Table 1).

Minor comments:

12. In Figures 1a and 1b, the authors try to use a model to explain antibody diffusion to reach a deep tissue antigen target. However, this model might be unrealistic because the natural combination of the antigen with antibody is influenced by many parameters, such as the chemical steric hindrance and specific binding sites. Before the antibody meets the antigen, the antibody should also overcome the barrier of endothelial cells/intercellular substances to enter into deeper tissue.

Response:

Thanks to the reviewer for providing the critical comments and the insightful discussion. We agree that all models are at best approximations to real world phenomena and the level of details should suit the question we seek to address. In our case, since we used this model to formulate our novel strategy in deeply penetrating immunostaining (namely, immunostaining at denaturing temperatures to switch off antibody-antigen reaction on-demand), and that the experimental results were compatible with what we modelled (e.g., as shown in Fig. 3 & 4, Supplementary Fig. 9, 10 & 13), we are confident that the model (now revised based on the comments of Reviewer 2, and more quantitative with additional simulation results shown in Fig. 1c, d) captured the essential aspects of the antibody-antigen binding reaction-diffusion process for our study's purpose.

For discussion sake, at initial stages of SPEAR development, we also considered the possibility that P3PE crosslinking might slow antibody-antigen interaction via a mechanism

by which the rotational dynamics of the Fab regions (cf. Galanti et al., 2016; doi:10.1038/srep18976) would be slowed. This may be due to an increased intramolecular viscosity, especially with the increase in hydrodynamic radii imparted by the addition of hydroxyl-rich crosslinker on the antibody surfaces, leading to the "searching" of suitable antigen binding sites becoming less efficient. The changes in the physicochemical properties of the antibodies could also differentially affect how the antibody moves across hydrophilic and hydrophobic barriers in the tissue. Nonetheless these are too complex to model and deviates from the scope of the current study.

13. In Figure 1c, triepoxide P3PE combines with protein not only by the amino group but also by the sulfhydryl group.

Response:

Thanks for the careful observation and reminder, we have now corrected the corresponding figure as advised (revised Fig. 1e).

14. I love the colored images in Figure 3I and Sup. Fig 10. However, I don't see the original data fitting. Can authors show similar axonal tracks in original Th stained brains by orthoslicing end-to-end?

Response:

We are glad the reviewer appreciated the colored images. We have now provided a detailed video for the ChAT SPEAR-stained whole mouse brain to better illustrate the staining quality of fibers tracts in the original images (Supplementary Video 1), which also includes 3D rotating views of the tractography. The resolution of the TH-stained brain is too low for performing tractography as it was imaged using mesoSPIM (whereas the ChAT-stained brain was imaged with two-photon tomography). The TH+ve axons of the nigrostriatal projection can be seen in Supplementary Video 3 via ortho-slicing.

15. Figure 3c-e compared the penetration depth difference to demonstrate the ThICK staining is better. But the conventional immunostaining method as a negative control group is not reliable for deep tissue immunostaining.

Response:

We fully agree with the reviewer that conventional immunostaining should indeed not be regarded as negative control for ThICK-staining, but to illustrate the point that ThICK-staining helps to overcome the reaction barrier that is prominent in conventional immunostaining (Fig. 3c–e and Supplementary Fig. 10). These have been made clearer in the revised manuscript (page 5, paragraph 2), along with the addition of further stringent benchmarking experiments against iDISCO and CUBIC-HV (Fig. 4).

Reviewer #2:

Remarks to the Author:

This manuscript by Lai and colleagues proposes a new high-throughput 3D staining method in combination with antibody complex stabilization (SPEARs) and staining protocol for thick tissue samples (ThICK). They claimed that crosslinking by an epoxy chemical P3PE stabilizes the antibody complex so that it can resist incubation at a high

temperature (55°C). The given thermostability leads to reduced antibody binding to the antigen (decreased K_d) and thus contributes to promoting antibody penetration. The authors showed SPEARs to apply to up to 24 antibodies and the ThICK protocol to stain whole human tissue blocks and whole adult mouse brains.

Considering that 3D tissue staining methods for large tissue samples are still in their infancy, various advanced methods must be proposed and discussed in the research community. Therefore, this reviewer positively evaluated the author's timely idea and efforts to shorten the 3D staining period, a significant bottleneck of the current methods, by applying the recent SHIELD-related crosslinking. On the other hand, this reviewer found several concerns in the current manuscript as follows, which is expected to be revised for finalization.

General response:

We thank the reviewer for the positive evaluation that our idea is timely and recognition of our effort. Based on the reviewer's comments, we have now (i) improved the mechanistic model proposed with better conceptual explanations and reaction-diffusion simulation, (ii) carried out gel filtration chromatography to study the complex formation, (iii) included new benchmarking experiments against the original iDISCO and CUBIC-HV methods, (iv) repeated immunostaining for selected SPEARs, and (v) thoroughly revised the manuscript accordingly, including acknowledgement of our method's limitations more appropriately.

Major concerns

#1: Diffusion-reaction model (Figure 1a, b)

- Could the authors further clarify the temperature dependence of K_d on T ? In the current manuscript, the thermal denaturation (that can cause the decrease of $[Ab]$ and $[Ab+Ag]$. In contrast, increase $[Ag]$) and exothermic reaction (that means, as long as the reviewer understood, a higher environmental temperature reduced reaction in terms of thermodynamics). However, the reviewer does not think these mechanisms are self-evidence. Is it possible for the authors to add any

data (or reference with data) to show the K_d of Ab-Ag reaction as a function of T and its underlying mechanisms?

- In addition, the change in K_d is related to [Ab] value, so in Figure 1b, the line shape of K_d and the "antibody % active" line should be coordinated.
- The diffusion constant is proportional to the first order of T if the Stokes-Einstein type diffusion ($D \propto T$) is assumed. On the other hand, D is drawn to be proportional to T's second-order or nonlinear function in Figure 1b. What kind of penetration model is assumed here?
- This reviewer thinks that Figure 1b should be re-considered to reflect the real impact of each parameter on diffusion efficiency. For example, assuming D as a function of the first order of T, the change of reaction temperature from 37°C (310.15 K) to 55°C (328.15 K) only gives a 5.8% increase of D value ($328.15/310.15 \approx 1.058$). The impact of K_d change due to T should also be expressed as such an actual value change.

Response:

We thank the reviewer for the important comments to help us improve the theoretical framework. In the revised manuscript, we have (i) provided more detailed mechanistic explanations for how temperature increase helps overcoming the antibody-antigen binding reaction barrier during diffusion (note: binding reaction depletes antibodies, effectively creating a barrier that hinders their deep penetration), (ii) reviewed the literature and retrieved measurements regarding the temperature dependences of antibody-antigen binding kinetic (k_{on} and k_{off}) and association constants (i.e., $K_a = k_{on} / k_{off}$, where k_{off} and k_{on} denote the antibody-antigen dissociation and association rate constants, respectively), (iii) revised the diffusivity (D_{eff}) change to follow the Stokes-Einstein equation, (iv) incorporated the information from (ii) & (iii) into a reaction-diffusion process simulation that includes a temperature change, and (v) revised the main text and figures accordingly, as detailed below. By the principles of chemical reaction kinetics, we propose that the interactions between antibodies and antigens can be broken down into two equilibrium reactions: (1) an intramolecular reaction where a functional antibody is in equilibrium with its inactive form and denatured form (permanently inactive), assuming only one form exist for each species, and (2) a reversible intermolecular reaction kinetics of an antibody reacting with its antigen, based on intermolecular collisions, whereby there is a larger increase in the rate of bond breakage than rate of bond formation when T is increased (i.e., a larger increase in k_{off} than k_{on}). The process of how SPEARs work under heat-accelerated immunostaining can then be explained in terms of (1) favouring the equilibrium between antibody forms towards the maintenance of a functional form by preventing heat-denaturation, or encouraging reversible transition from the temporarily inactive form to the functional form (upon lowering the incubation temperature), while (2) allowing the larger k_{off} increase to overcome the antibody-antigen binding reaction barrier that depletes antibodies during diffusion into tissue. We have added more formal discussions on the above design concept in the main text (page 2, paragraph 3 & page 3, paragraph 1), along with reaction-diffusion modeling.

In the original manuscript, we intended to just conceptually illustrate how the different parameters might change with temperature, and we apologize that the schematic was not sufficiently sophisticated. We have made the following revisions:

- For diffusivity (D_{eff}): we fully agree with the reviewer that adopting the Stokes-Einstein relation is much more appropriate, and we have adjusted the line to be linear instead, and with a flatter slope to more realistically reflect the true magnitude of change (Fig. 2b).
- For antibody-antigen association constant ($K_a = k_{on} / k_{off}$): we reviewed the literature (cf. Ref 27: Johnstone et al., 1990) and extrapolated the data to obtain the estimates that at 55°C, there is ~60.75 times increase in k_{off} , and only ~2 times increase in k_{on} , compared to ambient temperature (22°C), resulting in a ~30.38 times smaller K_a (i.e., favoring antibody-antigen dissociation, as would be expected given the exothermic nature of the reaction). The magnitude of change in K_a is therefore much larger than that in D_{eff} (as we show in the revised Fig. 1b). K_a vs. temperature (T) follows van 't Hoff's equation:

where ΔH , ΔS , and R are standard reaction enthalpy change, standard reaction entropy change, and the ideal gas constant, respectively. With $\Delta H < 0$ for exothermic reactions, K_a has an exponential dependence on $1/T$ (or equivalently $-T$), which is the basis for the shape of the line shown in the schematic (Fig. 1b). If the reviewer has other models in mind, please by all means kindly feel free to further advise us.

- To dissect the contributions of lowering K_a vs. increasing D_{eff} , we performed dimensionless reaction-diffusion process modeling similar to previously described (Susaki et al., 2020), but additionally incorporating a temperature change during the process (see Methods for details). We noted that the contribution by overcoming the reaction barrier (i.e., lowering K_a with an elevated T , which is only viable with thermostabilized antibodies) is theoretically much larger than the diffusivity increase, to the increased antibody penetration and eventual antibody-antigen complex concentration homogeneity across tissue depth (Fig. 1c, d).

Collectively, thanks to the enlightening comments, we believe these revisions provide a more solid theoretical foundation for our method design concept.

#2: Crosslinking and complex formation (Figure 1, 2)

-The documentation of SPEARs formation efficiency is still confusing. The antibody bands that are not contained in the complex are detected in Figure 1d, while Figure 2i states that they almost completely moved to the complex band (even in the "control" samples). Is it possible to show how much formation efficiency the SPEARs used in the benchmark test in Figure 3 produced?

Response:

We apologise for the confusion and we thank the reviewer for carefully noting the discrepancies. Gel imaging conditions were kept the same within each experimental set for comparability, but the two figures (original Fig. 1d, now Fig. 1f and Fig. 2i) came from two sets of experiments and therefore had different imaging conditions (e.g., exposure and contrast). Assessing the efficiency of SPEAR formation is in theory possible with gel filtration chromatography, but challenging due to substantial interference by albumin additives in commercial samples. In this regard, we developed our ELISA variant for the functional assessment and optimisation of SPEARs, which is more relevant for immunostaining.

-Is this complex monomeric or a huge concatenated one? The band is smeared in Figures 1e and 1f, suggesting concatenated complex formation. If that is the case, the diffusion efficiency and signal intensity could be affected by the molecular weight and reduced active sites for antigen binding.

Response:

Thanks for bringing up this question, which we also have been curious about since it may impact staining performance, as the reviewer pointed out. We thus performed gel filtration chromatography experiments on SPEARs. To our pleasant surprise, we found minimal concatenated complex formation when the IgGs are complexed by the correct Fabs, and concatamers of IgGs are present only when incorrect Fabs are used (see Fig. 1g and

Supplementary Fig. 3). In the revised manuscript, we thus concluded that “Gel filtration chromatography analysis showed that most complexes are bound by 1 – 2 Fab fragments, and multi-complex crosslinking rarely occurs under our optimized reaction conditions (Fig. 1g, Supplementary Fig. 3).”

#3: Staining and imaging quality (Figure 3, Supplementary Fig 9,11,12)

-Compared to previously reported mouse whole-brain imaging images such as iDISCO, eFLASH, and CUBIC-HistoVIsion, the data presented in Figure 3 and related supplementary figures are not comparable. Compared to these previous reports, the whole-brain image of this manuscript has a significantly worse signal-to-noise ratio and shows non-uniform staining (Supplementary Figure 9). If there is a trade-off between speed and data quality, it would not be accepted in many researchers. The shortcomings and limitations of current SPEARs in the comparison of previously reported methods should be accurately discussed.

For example, one of the causes of the low signal-to-noise ratio may be the persistence of thermal stability of SPEARs. As the authors state, "15.9% of crosslinked SPEARs still remained functional after heating at 55 °C for 16 h" (page 3, paragraph 3), SPEAR can be largely denatured and cease to function as an active antibody during the 72-hour staining incubation (Figure 3). These inactive antibody complexes can cause high non-specific background.

Response:

We thank the reviewer for sharing the insights and advice on appropriately discussing the limitations of our method. Indeed the lower signal-to-noise ratio (SNR) as well as the presence of intravascular precipitates are the main disadvantages of ThICK-staining with SPEARs, in exchange for speed. We do wish to share that these can be partially mitigated with staining protocol optimisations (Fig. 2f), and the staining is more clearly shown via image stack end-to-end slicing (please see new Supplementary Videos 1–3).

In the revised manuscript, we added supplementary notes in the Methods for end-users' information (pages 32–33, sections SPEARs synthesis from commercially available primary antibodies and Thermo-immunohistochemistry with optimized kinetics (ThICK) staining protocol), and reassessed our performance with the new benchmarking experiments against iDISCO and CUBIC-HV (presented in the new Fig. 4, please see responses to Major Comment 4 for details). Indeed, we observed higher backgrounds in the SPEAR group, that may be attributed to non-specific binding of denatured SPEARs, and we admit that we have yet to completely prevent this from happening. In the revised main text, we acknowledge that while ThICK-staining offers much improved tissue staining speed (Fig. 4c and Supplementary Table 2), there is a trade-off on SNR as exemplified by the overall higher background signals seen on the mouse hemibrain sample ThICK-stained with PV SPEARs (page 7, paragraph 1, Fig. 4e), and that ThICK will require further optimizations to improve the SNR of staining (page 7, paragraph 2 & page 8, paragraph 1), to more appropriately reflect the current limitations of our method.

On a separate note, inspired by this comment (and Major Comment 3 by Reviewer 3), we additionally tested and show that SPEARs exhibit extended thermostability with

ThICK-staining up to 4 weeks (page 7, paragraph 1, Supplementary Fig. 14), which should be sufficient for staining centimeter-scale tissues, compared with millimeter-scale tissues taking 3 days.

-Further investigation should be made on the staining specificity of SPEARs. For example, some of the data shown in Figure 2d may not be specific stained images. This reviewer could not judge that a specific staining signal was obtained, at least for the images of GFAP, S100b, OLIG2, c-Fos. As for the whole brain images, the uniform staining signal of ChAT and TH in the striatum was not obtained (Supplementary Figures 9, 12, also see the results in Susaki et al. 2020). A comparison data with conventional staining should be presented, at least for these cases.

Response:

We repeated the immunostaining for the specified antibodies and updated Fig. 2d accordingly. Indeed we found that for the anti-S100b antibody tested, the signals were too weak for it to be deemed workable as SPEAR. We thus removed the corresponding image from the figure and enlisted the tested anti-S100b antibody as unsuccessful in our hands (Supplementary Table 1). For the other targets, the improved immunostaining with the respective SPEARs showed typical staining patterns (e.g., revealing morphologies typical of astrocytes for GFAP, neuronal soma with c-Fos, round cells in white matter tract for OLIG2, see updated Fig. 2d). The staining specificities of other SPEARs are also evidence from the respective images (e.g., in Fig. 2c, showing that TH or VIP SPEARs stained a subset of Thy1 promoter-driven GCAMP6f-expressing neurons).

The volumetric images for the ChAT and TH SPEARs mouse whole-brain ThICK-staining are now shown in videos (Supplementary Videos 1 & 2). The signals of ChAT whole-brain staining for striatal cholinergic neurons were indeed weak compared to those in other areas (e.g. hippocampus, pedunculo-pontine nuclei) and to those by Susaki et al., 2020, though we were able to observe finer structures such as layer I cholinergic axons and those around the basal forebrain. Note that for the TH SPEAR hemibrain staining, there was striatal staining as visualized by low-resolution imaging using mesoSPIM, with a better signal-to-background contrast than optimized conventional staining (see comparisons shown in Supplementary Video 3). Overall, we agree that the signal homogeneity with ThICK-staining has room for improvement, although based on additional stringent benchmarking experiments, we showed that it is comparable to CUBIC-HV and superior to iDISCO (Fig. 4) (please see below responses to Major Comment 4 for details).

#4: Benchmarking test

- Immunostaining against an antigen uniformly expressed throughout the sample helps show the 3D staining efficiency in a large tissue sample. As demonstrated in Scales [Hama et al. Nat Neurosci 2015] and CUBIC-HistoVision [Susaki et al. Nat Commun 2020], this reviewer suggests that mouse brains should be stained with NeuN antibody for the benchmark of whole mouse brain immunostaining. This reviewer also suggests the comparison of NeuN immunostaining efficiency and quality by using ThICK and the original iDISCO (<https://idisco.info/idisco-protocol/>) protocol, not the modified one in the current manuscript, related to Supplementary Fig. 12.

Response:

Thanks for this advice. We have now performed mouse thick brain slice and hemi-brain ThICK-staining against two uniformly expressed targets (MAP2 and PV, but not NeuN due to inability in making NeuN SPEARs, please see Supplementary Table 1), for our benchmarking experiments comparing against (i) the original iDISCO protocol except omitting the DMSO (which in our hands did not work well with our antibodies), and (ii) CUBIC-HV. The use of BABB clearing was based on our experience that it produces the greatest tissue transparency for imaging with SPIM, as well as for fairness in comparing across different experiments.

As shown in the new Fig. 4, we had the following main findings:

- In the first set of benchmarking experiments, we stained 2-mm thick mouse brain tissues with ThICK vs. iDISCO, and found that ThICK-staining resulted in strong signal intensity and homogeneity across tissue depths (Fig. 4a, b).
- In the second set of benchmarking experiments, we ThICK-stained mouse hemibrain samples (with PV SPEARs), and compared the image obtained against that by iDISCO or CUBIC-HV (with standard anti-PV antibodies) after counter-staining the cut-surface near sample centroids (using anti-PV antibodies conjugated to a different fluorophore) (Fig. 4c–e). By quantifying PV+ve cell pixel intensity vs. distance from nearest tissue surface, we noted that CUBIC-HV achieved the best penetration depth, whereas ThICK-staining resulted in better signal homogeneity and less “rimming” effect across depth (Fig. 4e, f).

To ensure comparability, we ensured that all tissue pre-processing steps and imaging parameters in the benchmarking / counter-staining experiments were kept identical. We tried to design our benchmarking as stringent as possible (please see Methods).

Minor concerns

#1: Is there a specific reason for using Affipure Fab rather than FabuLight Fab?

Response:

This is an interesting question, we in fact did not have a specific reason for using Affinipure Fab – to be honest, we just bought Fab based on the first answers we got from Google search.

#2: CUBIC-HistoVision -> designated as CUBIC-HistoVision in the original paper.

Response:

Thanks for the reminder, we have corrected this throughout the manuscript.

Reviewer #3:

Remarks to the Author:

The authors report a chemical approach for antibody stabilization, termed SPEARs, to allow their usage at higher temperature (up to 55C) for faster diffusion into thick tissues, for up to 72 hours. To achieve this, the authors build upon previously published P3PE approach to crosslink Antibodies with fluorescently labelled Fab fragments (from secondary antibodies), yielding stabilized SPEARs. The author first utilized a variant of

ELISA method to systematically optimize various parameters and the reaction stoichiometry, for enhanced stabilization of antibodies. Next, the authors used such SPEAR reagents to perform stainings in human brain tissue (up to 800 microns; TH antibody) and whole mouse brain preparations (ChAT and PV). In addition, the authors report testing of 24 antibodies for compatibility with the SPEAR approach.

Overall, it's an interesting study with a lot of experiments to address a very challenging problem in the field - namely of faster and deeper staining of large blocks of tissues. However, there are a few significant concerns, as follows:

General response:

We thank the reviewer for the positive remark that our study is an interesting attempt to address a very challenging problem in the field. We understand that the reviewer had reservations regarding the image quality and scalability of the proposed ThICK-staining method, in response to which we have (i) performed additional stringent benchmarking experiments, (ii) included more example images to illustrate the image quality, (iii) demonstrated the extended thermostability of SPEARs for up to 4 weeks when used in ThICK-staining, (iv) thoroughly revised the manuscript to emphasize the main advancements made, while also acknowledging more appropriately our method's limitations.

1. Generally, the SNR of images shown looks to be relatively poor in most cases (even for highly expressed target proteins). For examples, in Figure 2d many of the images don't look to be good and high SNR labels as would be expected. Figure 3b is not showing raw image quality but instead focusing on "segmented" labels (whereas for the regularly antibody staining raw images are shown). Figure S12b, reporting TH staining, is not convincing. Why there is no staining at all in regular TH antibody, not even in hypothalamus which is pretty near to surface? The signal in the SPEAR stained TH is not impressive. Generally TH antibody labels tons of projections also. PV stainings in Figure 3 appears to be quite bad, generally PV is highly expressed and gives fantastic signals. Also, surprised by a lack of videos (unless I missed it), to better demonstrate the staining qualities, specially with depth. Overall, low SNR images, and reported existence of intravascular precipitates, might suggest that the SPEARs have significantly altered and/or reduced affinity.

Response:

We thank the reviewer for reminding us the importance of showing convincing raw images and the illustration of staining quality with videos, which we now have included in the revised manuscript as detailed below:

- For the example previously shown in Fig. 2d, we have repeated the experiments for the SPEARs that did not give clear images on the initial test (including GFAP, c-Fos and OLIG2 SPEARs, as also suggested by Reviewer 2), and made sure we have sufficient insets in other figures to better showcase the staining quality using enlarged views (e.g., Fig. 1n, 2c, m, 3i-k, 4a, e, and Supplementary Fig. 12d, 13b, c, 14). Please note that the data presented in Fig. 2d were meant to demonstrate the compatibility of SPEAR synthesis method with a wide range of commercially available antibodies, as well as ability to withstand 16 hours of 55°C heating in ThICK-staining. They were not

optimized for large-scale deep immunostaining for each of the antibodies, which would be prohibitively laborious and deviates from the scope of the current study. In the revised manuscript, we enlist all 25 tested antibodies compatible with SPEAR synthesis, and 4 that were not (Supplementary Table 1).

- For the image volumes of the human pons sample ThICK-stained with TH SPEARs shown in Fig. 3b, we have now included further detailed images without segmented cells overlaid for the readers' reference (new Supplementary Fig. 10a, b). We also outline the tissue surfaces (Supplementary Fig. 10a), to better illustrate the improved probe penetration by SPEARs compared to standard antibodies (Supplementary Fig. 10b), from which the quantifications were based upon (Fig. 3e).

- Concerning TH staining in the mouse hemibrains shown (original Supplementary Fig. 12, now as Supplementary Fig. 13), the 600 μ m-thick maximum intensity projection (MIP) images obtained by optimized conventional immunostaining with iDISCO-tissue preprocessing (Supplementary Fig. 13b, left panel) did not show TH+ve neurons because: (i) the standard anti-TH antibodies deposited mostly over the tissue surface, (ii) a low concentration of antibodies was used (but equimolar to TH SPEARs used for fair comparison), and (iii) the imaging was done using a low-magnification objective, which collectively resulted in a bright rim that obscured the deep TH+ve neurons on MIP (also see Supplementary Fig. 13c, lower panels). Some TH-positive neurons can in fact still be vaguely seen in the hypothalamus for the conventionally stained hemibrain (Supplementary Fig. 13c, lower panels). For the mouse hemibrain ThICK-stained with TH SPEARs which also underwent iDISCO-tissue preprocessing, we have now included a video with side-by-side comparison for better demonstration of the improved staining quality (Supplementary Video 3). With ThICK-staining, note that there are visible nigrostriatal projections labeled by TH SPEARs (containing the thickest TH+ve axons, see Supplementary Video 3). For the mouse hemibrain ThICK-stained with TH SPEARs (Supplementary Fig. 13), we emphasize that the imaging resolution and magnification with mesoSPIM were insufficient to observe individual TH+ve fibers, but only a general distribution pattern.
- To elaborate further: Due to the poor penetration and low imaging resolution, the superficially distributed TH antibodies on the iDISCO-processed, conventionally stained brains would appear like a smear of bright signal near the tissue surface. With better penetration by SPEARs yet still imaged at low resolution, the images would appear as more uniform smears of signals across depth. These would be in contrast to the case of the TH SPEAR-stained human pons (presented in Fig. 3), where the human sample was imaged at much higher resolution and magnification and subsequently stitching the tiled images together. Thus, the superficial stained axons can be seen more easily as there is little signal overlap over depth when imaged under high resolution (Supplementary Fig. 10), while the image volume of a thoroughly stained sample imaged at low resolution will appear quite uniform on projected view (Supplementary Fig. 13b, c). To better illustrate such phenomena, please refer to the schematics shown below (Rebuttal Fig. 2).

Case of Supp Fig 13, mouse hemibrain with TH staining

Case of Fig 3, human pons with TH staining

Rebuttal Fig. 2. Illustration of how sample signal distribution (uniform vs. some local features more prominent than others), imaging resolution, and antibody penetration affect the displayed images in Supplementary Fig. 10 & 13 and Fig. 3.

- We provide additional videos (Supplementary Videos 1 & 2) to show that ThICK-staining of whole mouse brain with ChAT or PV SPEARs allowed unambiguous brain-wide identifications of ChAT+ve neurons and cholinergic fibers (Supplementary Videos 1), or PV+ve neurons (Supplementary Videos 2).
- Finally, in the revised manuscript (page 7, paragraph 2 & page 8, paragraph 1), we also state that: “Although other state-of-the-art methods take longer times to achieve whole-organ immunolabeling (see Fig. 4c, Supplementary Table 2), ThICK still requires further optimization given the comparatively lower signal-to-noise ratio of staining.”, to more appropriately acknowledge the limitation of SPEARs and ThICK-staining.

2. The demonstration of whole mouse brain staining is poor to fully assess the performance of this approach. Ideally, the author should take a physical cross-section of the stained brain, counter-stain (with different fluorophore) the section with regular antibody, and image it to assess to uniformity and quality of signal. The images shown are not impressive.

Response:

Thanks to the reviewer for this important advice on the necessity of counter-staining experiments. As described in the revised manuscript (page 6, paragraph 3 & page 7, paragraph 1, with details in the Methods), we have now performed two stringent benchmarking experiments using highly expressed targets (i.e., MAP2 and PV) with the appropriate imaging resolutions for benchmarking purposes (Fig. 4):

- In the first set of benchmarking experiments, we stained 2-mm thick mouse brain tissues with ThICK vs. iDISCO, and showed that ThICK-staining resulted in strong signal intensity and homogeneity across tissue depths (Fig. 4a, b).
- In the second set of benchmarking experiments, we ThICK-stained mouse hemibrain samples (with PV SPEARs), and compared the image obtained against that by iDISCO or CUBIC-HV (with standard anti-PV antibodies) after counter-staining the cut-surface near sample centroids (using anti-PV antibodies conjugated to a different fluorophore) (Fig. 4c–e). By quantifying PV+ve cell pixel intensity vs. distance from nearest tissue surface, we noted that CUBIC-HV and ThICK-staining achieved the best penetration depths, whereas ThICK-staining resulted in better signal homogeneity and less “rimming” effect across depth (Fig. 4e, f).

For comparability, we ensured that all tissue pre-processing steps and imaging parameters in the benchmarking / counter-staining experiments were kept identical. Apart from demonstrating the advantages of ThICK-staining, we believe such stringent benchmarking experimental design would be helpful for future deep immunostaining method development too (for details please see Methods).

3. If the SPEARs are only stable for 72 hours, then that does put hard limits on their utility for deeper staining in human brain, beyond 800 microns. How is this approach "scalable" then?

Response:

Thanks for this comment on the potential limit of SPEAR heat stability and the implication on scalability. To address this concern, we have now tested and shown the extended heat stability of SPEARs with ThICK-staining up to 4 weeks (page 7, paragraph 1, Supplementary Fig. 14), which should be sufficient for staining centimeter-scale tissues, compared with millimeter-scale tissues taking 3 days.

We also wish to clarify the rationale of our experimental design and data presented, and we apologize if these were not sufficiently clear in the previous manuscript. Based on others' works and ours, the penetration of immunostaining depends on the amount of antibody used, the volumetric shape of the tissue, and buffer compositions. Notice that during antibody diffusion into tissue, antibody-antigen binding reaction depletes antibodies, effectively creating a barrier that hinders their deep penetration. For the data presented in Fig. 2, we meant to demonstrate that (i) heating can in principle be continued for at least 72 hours by showing the preservation of signals (Fig. 2a, please also see page 4, paragraph 2), and (ii) that SPEAR penetration depth increases with ThICK-staining duration (Fig. 2b). For the human brain samples, please note that the amount of antibodies used differed (30 ul SPEARs for ThICK-staining vs. 100 ul standard antibodies for conventional staining), with which we

intended to show that the penetration depth depended more on overcoming the reaction barrier with heating (enabled only by the heat stability of SPEARs) than simply increasing the amount of antibodies used. Admittedly, even the SPEARs got depleted, as the antigen density appeared greater than expected, resulting in the limited penetration of ~800 μm after ThICK-staining for 48 hours. Nonetheless, we believe this can be mitigated by adding more SPEARs, as with the case for mouse whole- or hemi-brain immunostaining with millimeter-scale penetration depths of SPEARs (i.e., ThICK-staining with ChAT, TH, PV and MAP2 SPEARs as shown in Fig. 3, 4, and Supplementary Fig. 11–13). We also wish to point out that (1) factoring in tissue shrinkage due to BABB clearing (performed after staining, see Methods and Supplementary Fig. 1), the actual attained penetration depths in tissue were even greater than that shown in acquired volumetric images (page 6, paragraph 2); and (2) our quantification of penetration depths were based on 3D calculations from tissue surfaces to each voxel/ROIs, which will be smaller but more accurate than estimating from the z-depth of images, as done in previous studies. Overall, it's an extensive study, however it remains unclear how usable this approach is generally. Maybe for a few highly expressed targets.

Response:

We thank the reviewer again for the constructive evaluation. In the revised manuscript, we do admit our weakness stems from the lowered SNR (partly due to the overall higher background, see page 7, paragraph 1), and that ThICK will still require further optimization (page 7, paragraph 2 & page 8, paragraph 1). Nonetheless, this is a trade-off for the much faster tissue staining speed (from weeks to 3 days for whole brains), scalability and simplicity in operation offered by ThICK. In addition, we believe our development on synthesizing SPEARs from conventional antibodies can also benefit other methods – which when used in combination can speed up, simplify and enhance deep immunostaining, representing an approach distinct from other approaches, such as modifying buffer compositions, tissue matrix and staining conditions.

Decision Letter, second revision:

Dear Dr Ko,

Thank you for your patience. Your Brief Communication entitled "Antibody stabilization for thermally accelerated deep immunostaining" has now been seen by 2 reviewers, whose comments are attached. In the light of their advice we have decided that we cannot offer to publish your manuscript in Nature Methods.

You will see that, while they find your work of some potential interest, the reviewers remain unconvinced about the advance your methodological approach represents over available methods and about its broad applicability at this stage. We think that these criticisms are sufficiently important as to prevent publication of your work in Nature Methods.

Although we regret that we cannot offer to publish your paper in Nature Methods given these reviews, I have discussed your manuscript and the reviewers' comments with our colleagues at Nature Communications. If you are able to address all of the remaining reviewer concerns, and you are able to show that your method outperforms iDISCO, they would send the appropriately revised version out for further review if you transfer the revised manuscript to Nature Communications. Should you wish to have your revised paper considered by Nature Communications, please use the link to the Springer Nature manuscript transfer service in the footnote once the revision is ready, and include a point-by-point response to the reviewers' concerns.

Your handling editor at Nature Communications would be Dr. Cara Eldridge (cara.eldridge@nature.com). If there is anything you would like to discuss before transferring the paper and its reviews, please don't hesitate to contact her by e-mail.

I am sorry that we cannot be more positive on this occasion but hope that you find the reviewers' comments helpful when preparing your paper for submission elsewhere.

Sincerely,
Madhura

Madhura Mukhopadhyay, PhD
Associate Editor
Nature Methods

Please note that Nature Communications is a fully open access journal. For information about article processing charges, open access funding, and advice and support from Springer Nature, please consult the Nature Communications Open Access page (www.nature.com/ncomms/open_access/index.html).

For journal metrics, please visit our [Nature journals metrics page](http://www.nature.com/npg_/company_info/journal_metrics.html). Our [open access pages](http://www.nature.com/ncomms/open_access/index.html) contain information about article processing charges, open access funding, and advice and support from Springer Nature.

** To transfer your manuscript to Nature Communications, or another Nature Portfolio journal, please use our [manuscript transfer portal](https://mts-nmeth.nature.com/cgi-bin/main.plex?el=A2M3ULf1E6BaRo5X7A9ftdvqF6Pvky1ouhnR1jwzRqgQZ). If you transfer to Nature journals or to the Communications journals, you will not have to re-supply manuscript metadata and files, unless you wish to make modifications. This link can only be used once and remains active until used.

All Nature Portfolio journals are editorially independent, and the decision on your manuscript will be taken by their editors. For more information, please see our [manuscript transfer FAQ](http://www.nature.com/authors/author_resources/transfer_manuscripts.html?WT.mc_id=EMI_NPG_1511_AUTHORTRANSF&WT.ec_id=AUTHOR) page.

Note that any decision to opt in to In Review at the original journal is not sent to the receiving journal on transfer. You can opt in to [In Review](https://www.nature.com/nature-portfolio/for-authors/in-review) at receiving journals that support this service by choosing to modify your manuscript on transfer. In Review is available for primary research manuscript types only.

Reviewer Comments:

Reviewer #1 (Remarks to the Author):

I appreciate the addition of new data that improved the manuscript. Below are some further thoughts that could improve the manuscript.

Overall, the study is based on the P3PE-stabilized-antibodies that could accelerate the antibody process with heating at 55°C. The authors mentioned that raising the temperature would denature the protein and, therefore, use the P3PE linker to stabilize antibodies; it will also present how to solve the higher temperature impact of tissue antigen. Although the author uses the SHIELD protocol for cross-linking the protein before their THICK method, it might limit the broad application in the traditional PFA fixation tissue, which authors mentioned the PFA fixation could only stand for 1 hour for the THICK method (Figure 2c). Secondly, the author said another novel strategy in their study is the choice of chemically engineering antibodies and show the new data of validated antibody list in this updated manuscript (Supplementary Table 1). However, this chemically engineering antibodies method required the antibody from a compatible buffer without serum and BSA, which significantly reduces the choice of antibody for the user in the future.

Here are some further considerations:

rebuttal letter, reviewer 1

Comment 1&2:

It is still difficult to conclude that the new method is better than the iDISCO as the author selected an unvalidated antibody (MAP2) for comparison (Figure 4). Here is the validated antibodies list for iDISCO. <https://idisco.info/validated-antibodies/>

Comment 4.1:

It lacks reference or experimental evidence from the author to support their explanation which is the essential theoretical foundation for their study:

At 37°C, most antibodies will bind to their antigens as soon as they encounter one upon diffusing into the tissue. Since antigens have been fixed in position within the tissue, the slow dissociation of antibodies from their antigens effectively locks the antibodies among the superficial antigens.

Comment 4.2:

Regarding the Rebuttal Fig. 1, the author's new experiment only shows the Alexa Fluor dye heating at 55 °C. However, this experiment contained a far higher concentration of the Alexa Fluor dye than the actual concentration used in the tissue staining. Also, the higher temperature might damage the conjugation between the antibodies and Alexa Fluor dye. Therefore, this experiment is challenging to represent the actual situation of

the raising temperature impact to staining tissue and the conjugation between the antibodies and Alexa Fluor dye.

Comment 5:

The authors only provide kidney tissue with the THICK method, in which it is hard to say the successful labeling of the collecting ducts in the mouse kidney (AQP1 labeling in Supplementary Figure 15).

Comment 15:

Even though the authors want to illustrate that the THICK-staining has better antibody penetration than conventional immunostaining, it is difficult to conclude because the THICK-staining procedure contains the permeabilized SDS. This strong detergent will increase the penetration of antibodies anyway. This has to be compared similarly and/or discussed.

Reviewer #2 (Remarks to the Author):

This reviewer appreciated much effort made by the authors in the revision. All of the reviewer's concerns in the first round of review were correctly answered.

There are additional concerns for the revised manuscript as follows:

#1: This reviewer did not recognize the difference in Figure 2i, showing faster crosslinking by adding pyridine. Error bars and statistical analysis should be added to the WB panel's dot plot.

#2: The legend should be added in Supplementary Fig. 3.

#3: While the reviewer appreciated this is one of the current state-of-the-art in the 3D staining field, this reviewer still felt that overall image quality, particularly whole-brain images, is not relatively high compared with other similar studies. In addition, the advantages of this technology, i.e., the reasons users select this method among several similar techniques, are relatively ambiguous due to the authors' sincere and neutral way of describing the results. This point can be further improved in the revised manuscript.

Response to Editor, third revision:

Dear Dr Ko,

We are in the process of discussing your appeal on your manuscript - NMETH-BC45759E-Z.

Would you please provide a response to the following reviewer comments (from the previous round of review) to help us make a more informed decision on your manuscript?

As you will see referee 1 points out that controls are missing from the kidney tissue staining. They are also concerned that SDS alone might improve the permeability of the tissue without the need to increased temperature. Referee 2 is not completely convinced of the improvement in image quality.

1. From ref 1 - "Comment 5: The authors only provide kidney tissue with the THICK method, in which it is hard to say the successful labeling of the collecting ducts in the mouse kidney (AQP1 labeling in Supplementary Figure 15)."

2. From ref 1 - "Comment 15: Even though the authors want to illustrate that the THICK-staining has better antibody penetration than conventional immunostaining, it is difficult to conclude because the THICK-staining procedure contains the permeabilized SDS. This strong detergent will increase the penetration of antibodies anyway. This has to be compared similarly and/or discussed."

3. From ref 2 - "#3: While the reviewer appreciated this is one of the current state-of-the-art in the 3D staining field, this reviewer still felt that overall image quality, particularly whole-brain images, is not relatively high compared with other similar studies. In addition, the advantages of this technology, i.e., the reasons users select this method among several similar techniques, are relatively ambiguous due to the authors' sincere and neutral way of describing the results. This point can be further improved in the revised manuscript."

This information is intended for the editors but be sure to provide sufficient detail for us to adequately assess it. If we decide we would like to send any portion of the response to a reviewer we will first request permission and allow you to rephrase your response.

We would appreciate hearing from you as soon as possible so that we can take your comments into consideration when making a decision. If you expect this to take more than a few days please contact me. Please do not actually revise the paper at this time.

Thank you for your prompt attention to this matter.

Dear Madhura,

Thanks very much for promptly handling our appeal! Please kindly find below our more detailed responses:

1. From ref 1 - "Comment 5: The authors only provide kidney tissue with the THICK method, in which it is hard to say the successful labeling of the collecting ducts in the mouse kidney (AQP1 labeling in Supplementary Figure 15)."

Response

We wish to clarify that the main focuses of our study include (i) demonstrating the principle of thermostabilizing antibodies while retaining their antigen binding properties, which has more general implications on antibody engineering beyond immunostaining, and (ii) showing that our chemical stabilization method can be applied to attain deep and fast immunostaining with off-the-shelf antibodies. Arguably, studying the brain naturally benefits the most from intact organ deep immunostaining, since it enables the identification of neuronal subtypes with much higher throughput, and the tracing of intact neuronal fiber tracts over long distances. It is also the system we are ourselves most familiar with, hence we developed most of the immunostaining protocols and performed the benchmarking using mouse and human brain tissues.

In the previous round of review, Reviewer 1 asked "How about immunostaining on other mouse organs (liver, spleen, lung, and muscle)?" Our understanding was that the reviewer was asking whether THICK staining can be applied to the other organs. While we explained that our study focuses on the above-stated aspects, we did our best to address the inquiry by showing that ThICK-staining also works on kidney tissues. In the example shown in Supplementary Fig. 15, the collecting duct structures are already clearly demonstrated with the AQP1 staining. Note that in Supplementary Fig. 15, the image shown is a volumetric rendered view without any post hoc processing (e.g., contrast adjustment) and may hence appear slightly dim. We can provide individual image planes showing more clearly the AQP1 staining pattern and morphology of renal tubules if necessary, along with controls (which as we understood was not requested by Reviewer 1 in the previous round). We believe that further benchmarking experiments using kidney tissue, however, deviates from the scope and focus of our study, and has very limited added scientific values to the study.

2. From ref 1 - "Comment 15: Even though the authors want to illustrate that the THICK-staining has better antibody penetration than conventional immunostaining, it is difficult to conclude because the THICK-staining procedure contains the permeabilized SDS. This strong detergent will increase the penetration of antibodies anyway. This has to be compared similarly and/or discussed."

Response

We wish to point out that achieving better antibody penetration is a substantial challenge, therefore the Ueda's group, Chung's group, as well as several other reputed groups and us all have had to go to great lengths in developing CUBIC-HV, eFLASH, ELAST, and ThICK-staining with SPEARs etc. In our previous attempts before the development of ThICK/SPEARs, we tried 3 months of delipidation and still achieved only ~200 um penetration.

Our study was first inspired by qualitative observation by Renier et al., (Cell 159: 896-910) where antibodies are limited in amount, thereby subject to reaction rather than diffusion barrier. This was further substantiated by our modeling (Fig. 1c, d, also see Methods). Regarding the repeated request to compare similar cases, this has already been done in Fig. 3a–e and Supplementary Fig. 10, where both the control human sample was equally SDS-delipidated and their penetration directly compared. To

further generalize, delipidation by dichloromethane in iDISCO or 10% v/v Triton X-100 for 2 weeks also did not result in further penetration in our benchmarking experiment in Fig. 4. These were further illustrated by other groups' works in the development of CUBIC-HV and eFLASH.

Intriguingly, since other's and our modeling and experimental data showed substantial proofs that reaction barrier prevail over diffusion in antibody penetration, Reviewer 1's intuitive prediction could be opposite of what will be observed experimentally: since SDS-permeabilization opens up more intracellular antigens for access by antibodies, they will subject to more reaction barrier, and hence penetrate even less than non-permeabilized tissues.

3. From ref 2 - "#3: While the reviewer appreciated this is one of the current state-of-the-art in the 3D staining field, this reviewer still felt that overall image quality, particularly whole-brain images, is not relatively high compared with other similar studies. In addition, the advantages of this technology, i.e., the reasons users select this method among several similar techniques, are relatively ambiguous due to the authors' sincere and neutral way of describing the results. This point can be further improved in the revised manuscript."

Response

We agree with Reviewer 2 that perhaps due to our modest writing, other advantages (e.g. much faster method (days versus weeks), no requirement for specialized equipment, compatible with multiple clearing and staining methods, low cost and simple operation) were insufficiently emphasized. We can definitely further elaborate on these points in the revised manuscript, as Reviewer 2 kindly suggested. The field of 3D histology has been flooded with eye-catching images, but very few had performed carefully designed benchmarking experiments as we had done (e.g., in Fig. 4) – the selling point which we wish to draw the readers' attention to, even if they are visually less appealing – as we believe this is the most important aspect when comparing our method against others. Our benchmarking method was designed based on the reviewers' important suggestions (dual labeling in staged experiments), as well as other considerations, such as (i) minimal amount of antibodies with calculated stoichiometries, (ii) 3D voxel-wise penetration field quantification for ground-truth penetration assessment, and (iii) carefully chosen fluorophores and imaging strategies. These experiments had already benchmarked the "quality" of our method as thoroughly and fairly as possible, including THICK/SPEAR against optimized standard immunostaining (Supplementary Fig. 9, 10), iDISCO and CUBIC-HV (Fig. 4, Supplementary Fig. 13), all of which showed in

quantitative terms the relative performance of THICK/SPEAR over other methods. If additional visually appealing images are required, we will be more than happy to provide them.

Thanks a lot again!! Best regards,

Ho Ko

Decision Letter, third revision:

Dear Dr. Ko,

Thank you for submitting your revised manuscript "Antibody stabilization for thermally accelerated deep immunostaining" (NMETH-BC45759F). I am delighted to inform you that we'll be happy in principle to publish it in Nature Methods, pending minor revisions and to comply with our editorial and formatting guidelines.

TRANSPARENT PEER REVIEW

Nature Methods offers a transparent peer review option for new original research manuscripts submitted from 17th February 2021. We encourage increased transparency in peer review by publishing the reviewer comments, author rebuttal letters and editorial decision letters if the authors agree. Such peer review material is made available as a supplementary peer review file. Please state in the cover letter 'I wish to participate in transparent peer review' if you want to opt in, or 'I do not wish to participate in transparent peer review' if you don't. Failure to state your preference will result in delays in accepting your manuscript for publication.

Thank you again for your interest in Nature Methods. Please do not hesitate to contact me if you have any questions.

Sincerely,
Madhura

Madhura Mukhopadhyay, PhD
Associate Editor
Nature Methods

ORCID

Final Decision Letter:

Dear Dr Ko,

I am pleased to inform you that your Article, "Antibody stabilization for thermally accelerated deep immunostaining", has now been accepted for publication in Nature Methods. Your paper is tentatively scheduled for publication in our August print issue, and will be published online prior to that. The received and accepted dates will be May 2, 2021 and Jun 27, 2022. This note is intended to let you know what to expect from us over the next month or so, and to let you know where to address any further questions.

Please note that *Nature Methods* is a Transformative Journal (TJ). Authors may publish their research with us through the traditional subscription access route or make their paper immediately open access through payment of an article-processing charge (APC). Authors will not be required to make a final decision about access to their article until it has been accepted. [Find out more about Transformative Journals](https://www.springernature.com/gp/open-research/transformative-journals)

Authors may need to take specific actions to achieve [compliance](https://www.springernature.com/gp/open-research/funding/policy-compliance-faqs) with funder and institutional open access mandates. If your research is supported by a funder that requires immediate open access (e.g. according to [Plan S principles](https://www.springernature.com/gp/open-research/plan-s-compliance))

then you should select the gold OA route, and we will direct you to the compliant route where possible. For authors selecting the subscription publication route, the journal's standard licensing terms will need to be accepted, including [self-archiving policies](https://www.springernature.com/gp/open-research/policies/journal-policies). Those licensing terms will supersede any other terms that the author or any third party may assert apply to any version of the manuscript.

Your paper will now be copyedited to ensure that it conforms to Nature Methods style. Once proofs are generated, they will be sent to you electronically and you will be asked to send a corrected version within 24 hours. It is extremely important that you let us know now whether you will be difficult to contact over the next month. If this is the case, we ask that you send us the contact information (email, phone and fax) of someone who will be able to check the proofs and deal with any last-minute problems.

If, when you receive your proof, you cannot meet the deadline, please inform us at rjsproduction@springernature.com immediately.

Once your manuscript is typeset and you have completed the appropriate grant of rights, you will receive a link to your electronic proof via email with a request to make any corrections within 48 hours. If, when you receive your proof, you cannot meet this deadline, please inform us at rjsproduction@springernature.com immediately.

Once your paper has been scheduled for online publication, the Nature press office will be in touch to confirm the details.

Once your paper has been scheduled for online publication, the Nature press office will be in touch to confirm the details.

Content is published online weekly on Mondays and Thursdays, and the embargo is set at 16:00 London time (GMT)/11:00 am US Eastern time (EST) on the day of publication. If you need to know the exact publication date or when the news embargo will be lifted, please contact our press office after you have submitted your proof corrections. Now is the time to inform your Public Relations or Press Office about your paper, as they might be interested in promoting its publication. This will allow them time to

prepare an accurate and satisfactory press release. Include your manuscript tracking number NMETH-A45759G and the name of the journal, which they will need when they contact our office.

About one week before your paper is published online, we shall be distributing a press release to news organizations worldwide, which may include details of your work. We are happy for your institution or funding agency to prepare its own press release, but it must mention the embargo date and Nature Methods. Our Press Office will contact you closer to the time of publication, but if you or your Press Office have any inquiries in the meantime, please contact press@nature.com.

Nature Research journals [encourage authors to share their step-by-step experimental protocols](https://www.nature.com/nature-research/editorial-policies/reporting-standards#protocols) on a protocol sharing platform of their choice. Nature Research's Protocol Exchange is a free-to-use and open resource for protocols; protocols deposited in Protocol Exchange are citable and can be linked from the published article. More details can found at www.nature.com/protocolexchange/about.

Please note that you and any of your coauthors will be able to order reprints and single copies of the issue containing your article through Nature Research Group's reprint website, which is located at <http://www.nature.com/reprints/author-reprints.html>. If there are any questions about reprints please send an email to author-reprints@nature.com and someone will assist you.

Best regards,
Madhura

Madhura Mukhopadhyay, PhD
Senior Editor
Nature Methods

** Visit the Springer Nature Editorial and Publishing website at http://editorial-jobs.springernature.com?utm_source=ejp_NMeth_email&utm_medium=ejp_NMeth_email&utm_campaign=ejp_Nmeth for more information about our career opportunities. If you have any questions please click [here](mailto:editorial.publishing.jobs@springernature.com). **